# Intragenic viral silencer element regulates HTLV-1 latency via RUNX complex recruitment

Kenji Sugata[1,15], Akhinur Rahman[1,15], Koki Niimura[1,2], Kazuaki Monde [3], Takaharu Ueno [4], Samiul Alam Rajib[1], Mitsuyoshi Takatori[1], Wajihah Sakhor[1], Md Belal Hossain [1], Sharmin Nahar Sithi [1], M. Ishrat Jahan[1], Kouki Matsuda[5], Mitsuharu Ueda [6], Yoshihisa Yamano[7,8], Terumasa Ikeda [9], Takamasa Ueno [10], Kiyoto Tsuchiya [11], Yuetsu Tanaka[12], Masahito Tokunaga [13], Kenji Maeda[5], Atae Utsunomiya [13], Kazu Okuma[4], Masahiro Ono [14] & Yorifumi Satou [1]✉

Retroviruses integrate their genetic material into the host genome, enabling persistent infection. Human T cell leukaemia virus type 1 (HTLV-1) and human immunodeficiency virus type 1 (HIV-1) share similarities in genome structure and target cells, yet their infection dynamics differ drastically. While HIV-1 leads to high viral replication and immune system collapse, HTLV-1 establishes latency, promoting the survival of infected cells and, in some cases, leading to leukaemia. The mechanisms underlying this latency preference remain unclear. Here we analyse blood samples from people with HTLV-1 and identify an open chromatin region within the HTLV-1 provirus that functions as a transcriptional silencer and regulates transcriptional burst. The host transcription factor RUNX1 binds to this open chromatin region, repressing viral expression. Mutation of this silencer enhances HTLV-1 replication and immunogenicity, while its insertion into HIV-1 suppresses viral production. These findings reveal a strategy by which HTLV-1 ensures long-term persistence, offering potential insights into retroviral evolution and therapeutic targets.

There are two types of retroviruses, endogenous and exogenous retroviruses. The host cellular genome encodes the genetic code that is essential for the development and maintenance of a healthy life. The host cell has evolved with strategies to maintain homeostasis of the genome by silencing mobile DNA elements, both the endogenous and exogenous retroviruses. Endogenous retroviruses are subject to silencing that is established in early development, which can be stably maintained, whereas exogenous retrovirus silencing needs de novo establishment in differentiated host cells. Previous studies have revealed the silencing mechanisms of human endogenous retroviruses and the murine leukaemia virus[1–3]; however, those of human exogenous retroviruses, such as human T cell leukaemia virus type 1 (HTLV-1) and human immunodeficiency virus type 1 (HIV-1), have not been fully understood yet.

HTLV-1, a leukaemogenic retrovirus, has a geographically limited presence, hinting at its limited propagating capabilities[4,5]. HTLV-1

[1]Division of Genomics and Transcriptomics, Joint Research Center for Human Retrovirus Infection, Kumamoto University, Kumamoto, Japan. [2]School of Medicine, Kumamoto University, Kumamoto, Japan. [3]Department of Microbiology, Faculty of Life Sciences, Kumamoto University, Kumamoto, Japan. [4]Department of Microbiology, Kansai Medical University, Hirakata, Japan. [5]Division of Antiviral Therapy, Joint Research Center for Human Retrovirus Infection, Kagoshima University, Kagoshima, Japan. [6]Department of Neurology, Graduate School of Medical Sciences, Kumamoto University, Kumamoto, Japan. [7]Department of Neurology, St. Marianna University School of Medicine, Kawasaki, Japan. [8]Department of Rare Diseases Research, Institute of Medical Science, St. Marianna University School of Medicine, Kawasaki, Japan. [9]Division of Molecular Virology and Genetics, Joint Research Center for Human Retrovirus Infection, Kumamoto University, Kumamoto, Japan. [10]Division of Infection and Immunity, Joint Research Center for Human Retrovirus Infection, Kumamoto University, Kumamoto, Japan. [11]AIDS Clinical Center, National Center for Global Health and Medicine, Tokyo, Japan. [12]School of Medicine, University of the Ryukyus, Okinawa, Japan. [13]Department of Hematology, Imamura General Hospital, Kagoshima, Japan. [14]Department of Life Sciences, Imperial College London, London, UK. [15]These authors contributed equally: Kenji Sugata, Akhinur Rahman. ✉e-mail: y-satou@kumamoto-u.ac.jp

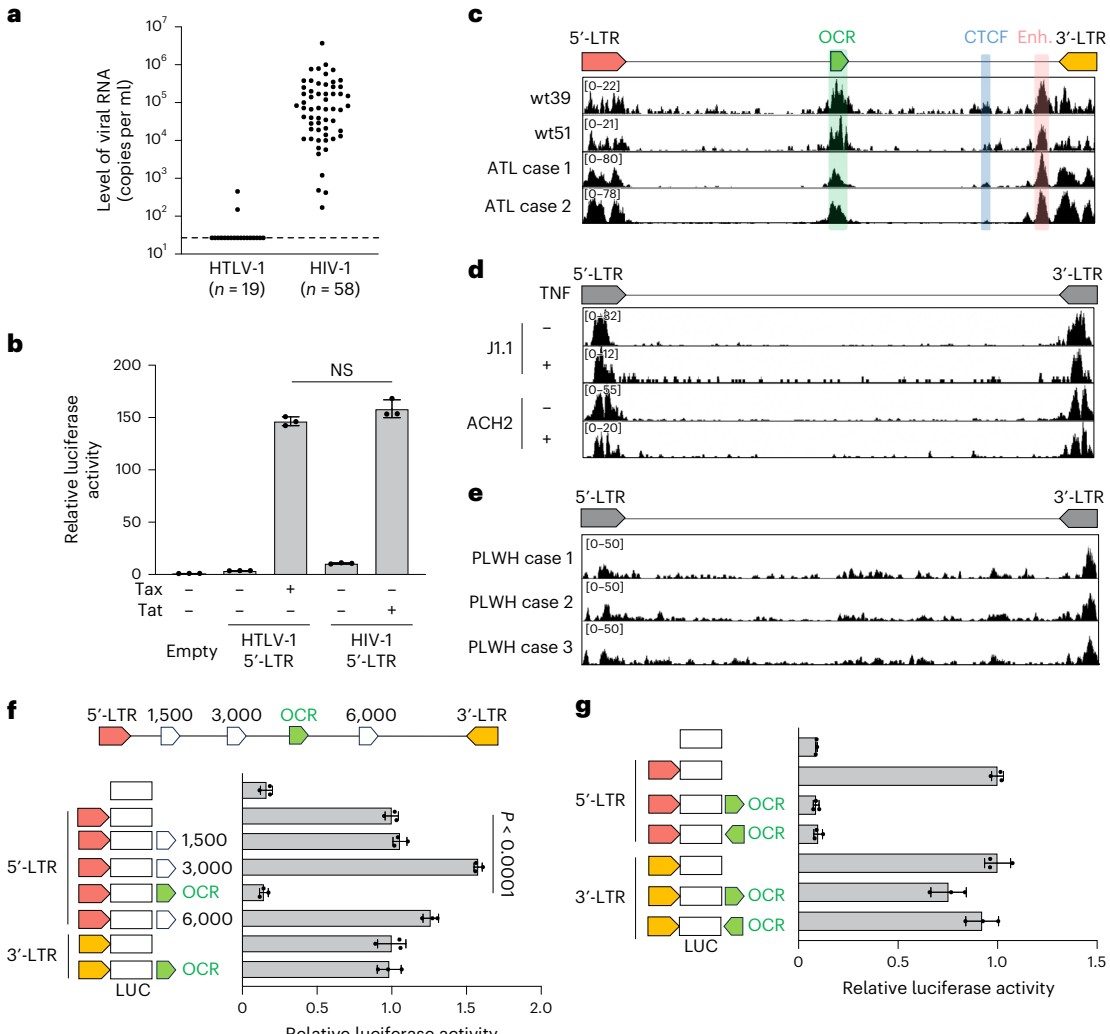

**Fig. 1 | Identification of an OCR in the middle of the HTLV-1 proviral genome with a suppressive function on the promoter activity of the HTLV-1 5′-LTR. a**, Plasma viral RNA levels in HTLV-1- or HIV-1-infected individuals. Viral RNA copy numbers were evaluated by reverse-transcription quantitative PCR (RT-qPCR) and ddPCR for HTLV-1 and HIV-1, respectively. **b**, The transcriptional activity of HTLV-1 or HIV-1 5′-LTRs was assessed by luciferase (LUC) reporter assay in Jurkat T cells with their respective *trans*-activators (Tax and Tat). **c**–**e**, ATAC-seq signals in the provirus region (5′-LTR, OCR, CTCF, enhancer (Enh.) and 3′-LTR) of HTLV-1-infected Jurkat T cell clones (wt39 and wt51) and PBMCs from two patients with

ATL (**c**), HIV-1-infected T cell lines (J1.1 and ACH2) with or without TNF stimulation (**d**) and PBMCs from three HIV-1-infected individuals (people living with HIV-1, PLWH) (**e**). **f**,**g**, Effect of the OCR or three randomly selected proviral regions on the 5′-LTR or the 3′-LTR promoter activity. Jurkat T cells were used for luciferase reporter assay 48 h after transfection (**f**). The directionality of the OCR did not change the effect on the 5′-LTR or the 3′-LTR promoter activity (**g**). At least two independent experiments were performed. The bars and error bars represent the mean ± s.d. of results in triplicate experiments. *P* values were calculated using a two-sided, unpaired Student's *t*-test (NS, not significant).

uniquely undergoes a self-induced latency, leading to a spontaneous reduction in viraemia. HTLV-1 achieves persistent infection mainly by inducing clonal expansion of infected CD4[+] T cells[6], leading to excessive T cell activation[7], which contributes to the leukaemic transformation that results in adult T cell leukaemia and lymphoma (ATL)[8–10]. HTLV-1 also causes inflammatory diseases, such as tropical spastic paraparesis and HTLV-1-associated myelopathy (HAM). By contrast, HIV-1 causes vigorous viral production and maintains high viraemia in the blood, resulting in the apoptosis of infected CD4[+] T cells and thereby causing acquired immune deficiency syndrome[11]. The sense transcript of HIV-1 encodes a transactivator Tat, which forms a strong positive feedback loop with the proviral promoter, 5′-long terminal repeat (LTR), facilitating vigorous virus production[12]. HTLV-1 also encodes Tax in its sense transcript, a strong transactivator for the 5′-LTR and an analogue to the Tat of HIV-1 (ref. 13). However, the regulation of proviral transcription differs notably between HIV-1 and HTLV-1, which contributes to the different levels of viraemia observed in infected individuals. In the case

of HIV-1, proviral transcription in the sense direction is significantly higher than in the antisense. By contrast, sense transcription from the HTLV-1 5′-LTR is frequently silenced or expressed only intermittently[14,15], whereas antisense transcription from HTLV-1 3′-LTR is much more constant than sense transcription[16]. We reported previously that a viral enhancer region near the 3′-LTR plays a key role in the constant expression of antisense transcription[17]. In addition, we identified a CCCTC-binding factor (CTCF, a transcription factor known for its role in chromatin organization and insulator activity)-binding site, in the middle of the HTLV-1 provirus that is pivotal in maintaining distinct epigenetic features between 5′- and 3′-LTR[18] and forming chromatin looping between the provirus and the integrated host genome[19]. The silencing of sense transcription is key to achieving latent infection, which minimizes the expression of the viral antigen encoded in the sense transcript and allows escaping from immune surveillance. However, the mechanisms by which HTLV-1 establishes spontaneous latency are still unclear. Their elucidation should provide insights not only

into HTLV-1 infection but also into HIV-1 infection. Here we discover a key difference in the HIV-1 and HTLV-1 silencing mechanisms, driven by a short regulatory sequence recruiting transcription factors with specific roles in the host cells.

## Results

### OCR in the HTLV-1 provirus with a suppressive function

We compared plasma viral RNA levels in individuals infected with HIV-1 or HTLV-1. HIV-1-infected individuals showed significantly higher viral RNA levels than HTLV-1-infected individuals, whose viraemia was mostly undetectable (Fig. 1a and Supplementary Table 1). Only 2 of 19 HTLV-1 asymptomatic carriers had detectable viraemia, with no correlation to proviral load (PVL) (Extended Data Fig. 1a and Supplementary Table 2). Among various viral proteins, HIV-1 Tat and HTLV-1 Tax regulate proviral gene expression. Promoter assay using Jurkat T cells showed no significant difference in 5′-LTR transcriptional activity between HIV-1 and HTLV-1, regardless of Tat or Tax presence (Fig. 1b). This suggests that factors beyond these promoters, such as some other *cis*- or *trans*-elements, contribute to the stark contrast in virus productivity.

We hypothesized that an unidentified 5′ regulatory region in the HTLV-1 provirus suppresses the 5′-LTR promoter activity. To test this hypothesis, we performed an assay for transposase-accessible chromatin with sequencing (ATAC-seq) analysis to identify open chromatin regions (OCRs) in the provirus with HTLV-1-infected cells. We used JET cells, a Jurkat T cell line containing the tdTomato fluorescent reporter protein under the control of a Tax-responsive element[20]. These cells were infected with wild-type HTLV-1, and then cloning was performed by limiting dilution to obtain infected clones in a previous study[17]. These infected JET cell clones contained one copy of intact HTLV-1 provirus while sustaining the latent state (Extended Data Fig. 1b–e). We also analysed fresh peripheral blood mononuclear cells (PBMCs) from two patients with smouldering ATL, a clinical subtype of ATL with an indolent course. ATAC-seq identified two open chromatin peaks in the 3′ proviral region, corresponding to known insulator and enhancer elements[17,18] (Fig. 1c). In addition, we discovered a previously unreported OCR in the 5′ proviral side, overlapping the viral polymerase coding region. Unlike HTLV-1, HIV-1 showed no intragenic ATAC-seq peaks beyond the LTRs, including in fresh CD4+ T cells from HIV-1-infected individuals (Fig. 1d,e). No similar intragenic OCRs were detected in other delta retroviruses, such as HTLV-2, simian T-lymphotropic virus 1 (STLV-1) or bovine leukaemia virus (BLV) (Extended Data Fig. 1f), although we cannot exclude the possibility of underestimation by using infected cell lines, in which the provirus tends to be silenced by DNA hypermethylation in the 5′-LTR promoter[21]. To investigate the function of the OCR in the HTLV-1 provirus, we performed a reporter assay using the 5′-LTR and 3′-LTR as promoters. The OCR significantly suppressed 5′-LTR promoter activity whereas other intragenic proviral regions did not (Fig. 1f). By contrast, the OCR showed little suppressive effect on the 3′-LTR, the promoter of antisense transcription from the HTLV-1 provirus (Fig. 1g). As the 5′-LTR is a TATA-box-containing promoter, whereas the 3′-LTR lacks an antisense TATA box, they show distinct promoter characteristics[22,23]. Consequently, the effect of the OCR should differ between them.

### Molecular characterization of the silencer complex on the OCR

Transcription factor (TF) motif analysis identified several candidate sites within the OCR, showing high similarity to consensus sequences for TF binding (Supplementary Table 3). Integrating ATAC-seq data, we selected RUNX, ETS and GATA TF families as key candidates for silencer function. Among them, RUNX1, ETS1 and GATA3 were most highly expressed in CD4+ T cells (Extended Data Fig. 2a–c). RUNX1 showed the strongest suppressive function in reporter assays (Extended Data Fig. 2d). Although other family members may contribute, these data suggest that RUNX1, ETS1 and GATA3 play major roles, leading us to focus on them in this study.

To assess binding to the OCR, we performed chromatin immunoprecipitation with sequencing (ChIP-seq) using viral DNA-capture-seq[24]. RUNX1, ETS1, GATA3 and the RUNX co-factor CBFβ bound to the OCR in both HTLV-1-infected cell lines and fresh PBMCs (Fig. 2a,b). RUNX proteins act in promoters, enhancers, insulators and silencers, with tissue-specific functions dictated by expression patterns, post-transcriptional modifications and stoichiometry of RUNX-mediated transcriptional complex[25,26]. Further analysis regarding cofactors related to RUNX1 revealed that HDAC3 and Sin3A localized to the OCR, strongly in PBMCs and marginally in cell lines (Fig. 2a,b), suggesting that they mediate OCR silencing via the RUNX1–CBFβ complex in naturally infected cells. HTLV-1-infected cell lines tend to show hypermethylation in the 5′-LTR, making them less dependent on silencer function to maintain a latent state[21]. Additional studies are needed to fully elucidate the molecular mechanisms underlying the silencer complex.

The OCR contains three RUNX1, ETS1 and GATA3 binding sites (Extended Data Fig. 3a). A core silencer region encompassing these sites showed silencing activity comparable to the full OCR (Extended Data Fig. 3b). We analysed TF effects on OCR-mediated suppression using a 5′-LTR reporter assay in 293T cells, which have lower endogenous TF expression than Jurkat cells. RUNX1 suppressed transcription, ETS1 enhanced it and GATA3 had little effect (Fig. 2c). These TFs also localized to the LTRs, but ChIP-seq could not distinguish between 5′- and 3′-LTR binding owing to their identical sequences (Fig. 2a,b). RUNX1 and ETS1 affected 5′-LTR promoter activity (Extended Data Fig. 3c,d), but their effects were stronger when the OCR was present, suggesting that their interplay modulates transcription. ETS1 counteracted RUNX1-mediated suppression (Fig. 2d). Mutations in all three RUNX1 sites abolished silencing, while ETS1 and GATA3 site mutations had no effect (Fig. 2e), highlighting the importance of RUNX1 as a key DNA-binding molecule to the OCR. Partial silencing loss occurred when individual RUNX1 sites were mutated, indicating that all three contribute (Extended Data Fig. 3e). Impaired silencer function from RUNX1 mutations could lead to aberrant viral protein expression and influence clinical outcomes, such as early ATL or HAM and tropical spastic paraparesis[27]. However, these RUNX1 sites were highly conserved across geographic strains (Extended Data Fig. 3f,g). Some ATL cases had OCR deletions, but these also lacked the 5′-LTR, suggesting that proviral silencing is maintained regardless of the OCR (Extended Data Fig. 3h,i).

HTLV-1 infects various cell types[28] but primarily transforms CD4+ T cells. We hypothesized that the silencer function of the OCR is most

**Fig. 2 | Molecular characterization of the silencer complex on the OCR. a,b,** ChIP-seq signals with HTLV-1 DNA-capture analysis for RUNX1, CBFβ, GATA3, ETS1, HDAC3 and Sin3A in HTLV-1-infected Jurkat T cell clones (wt39 and wt51) (**a**) and PBMCs of patients with ATL (AI-5 and AI-9) (**b**). **c,** Effect of overexpression of RUNX1, GATA3 or ETS1 on the 5′-LTR promoter activity with the OCR in 293T cells. **d,** Effect of overexpression of RUNX1 and/or ETS1 on the 5′-LTR promoter activity with the OCR in 293T cells. **e,** Changes in OCR-mediated silencing by mutating RUNX, GATA3 and ETS1 binding sites within the OCR in Jurkat T cells. **f,** Transcriptional regulation of RUNX1 mutants (S67I, W79C and R174Q) in OCR-mediated silencing of HTLV-1 5′-LTR in 293T cells. Protein expression of RUNX1 mutants was confirmed twice by western blot. **g,** Representative RT-qPCR result of *tax* mRNA expression in MT1 and TBX-4B cells transduced with RUNX1. **h,** Measurement of Tax protein levels in RUNX1-overexpressed MT1 cells. RUNX1 was transduced using a retroviral vector system. One representative result from each flow cytometry assay (left) and the cumulative Tax positivity values from a triplicate assay (right) are shown. **i,** Effect of RUNX1 knockdown via shRNA on OCR-mediated silencing of the HTLV-1 5′-LTR. Molt4 cells carrying shRNA for RUNX1 were used for luciferase assay. Luciferase reporter assays were performed 48 h after transfection. The results are representative of at least two independent experiments. The bars and error bars represent the mean ± s.d. of the results of triplicate experiments. *P* values were calculated using a two-sided, unpaired Student's *t*-test.

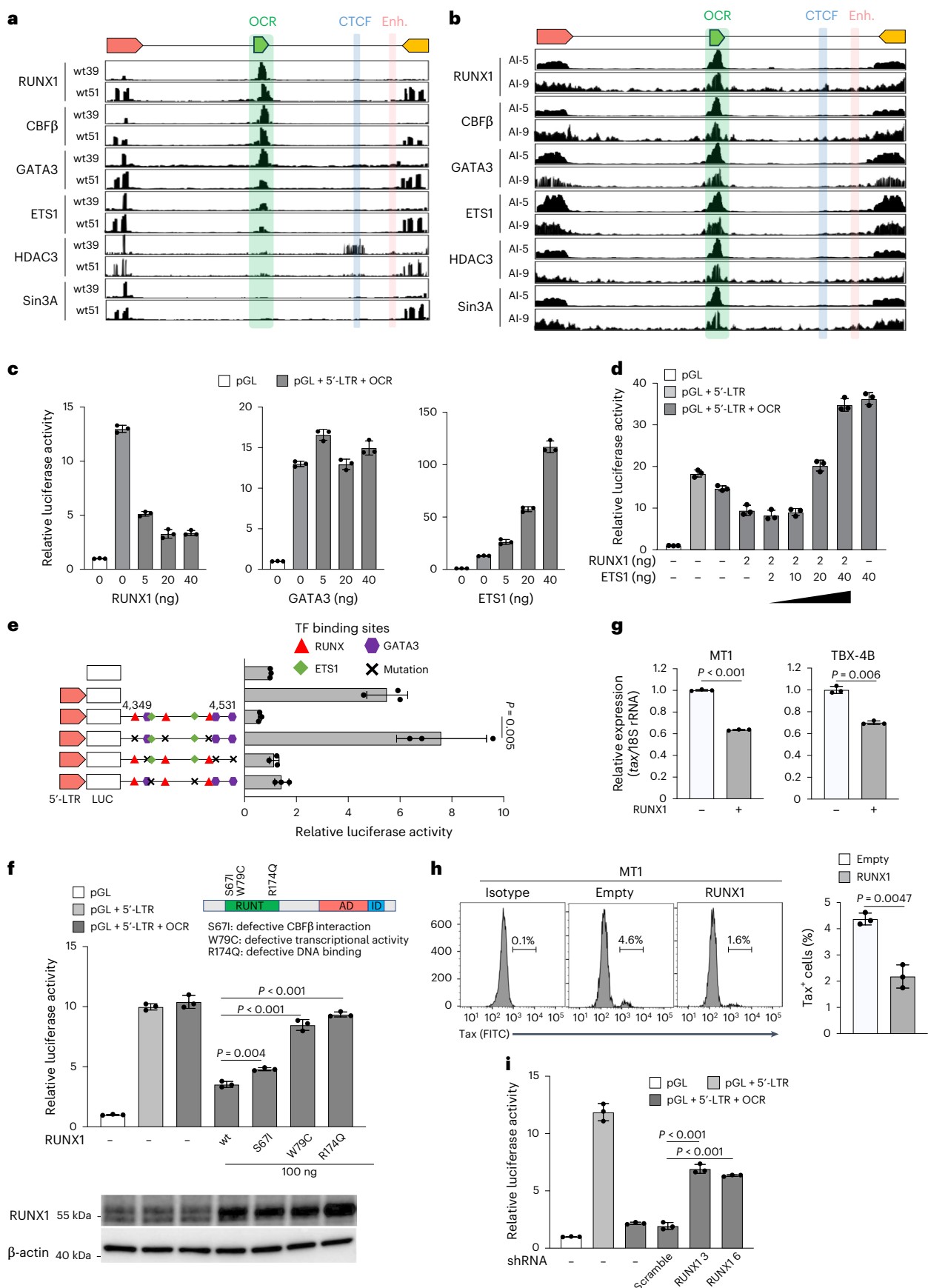

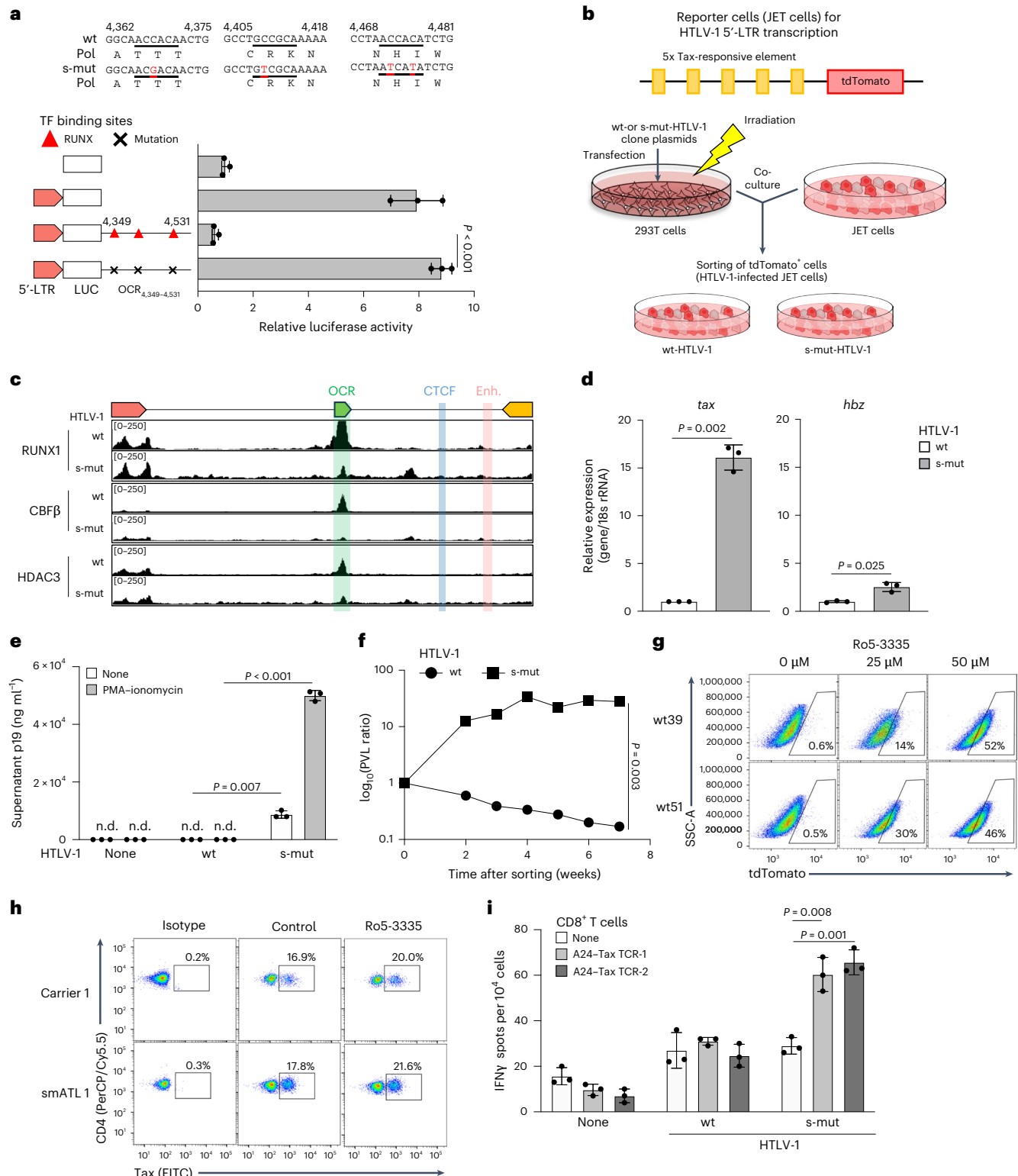

**Fig. 3 | Virological and immunological significance of the OCR function in HTLV-1 infection. a**, The nucleotide and protein sequences of silent mutations (s-mut) in the RUNX binding site (underlined) within the OCR. Mutated nucleotides are shown in red. Luciferase reporter assays were performed using Jurkat T cells 48 h after transfection. **b**, Experimental workflow illustrating the establishment of reporter cells (JET cells) infected with wt- or s-mut-HTLV-1. **c**, Signals from RUNX1, CBFβ or HDAC3 ChIP-seq in the HTLV-1 provirus for wt- and s-mut-HTLV-1-infected JET cells. **d**, Representative RT-qPCR results showing *tax* or *hbz* mRNA expression levels in JET cells infected with wt- or s-mut-HTLV-1. **e**, Supernatant p19 ELISA results from JET cells infected with wt- or s-mut-HTLV-1 with or without PMA–ionomycin stimulation. **f**, PVLs during long-term culture in

wt- or s-mut-HTLV-1-infected JET cells after tdTomato sorting. The data presented are representative of two independent experiments. **g**, Tax expression after treatment with RUNX1 inhibitor (Ro5-3335). Tax expression was analysed by expression of the reporter protein tdTomato. **h**, Effect of RUNX1 inhibitor on ex vivo Tax expression using PBMCs of ACs and patients with ATL. Tax-expressing CD4⁺ T cells were measured by flow cytometry. **i**, IFNγ ELISPOT assay results from Tax TCR-transduced CD8⁺ T cells cocultured with wt- or s-mut-HTLV-1-infected JET cells expressing HLA-A*24:02 (A24). At least two independent experiments were performed. The bars and error bars represent the mean ± s.d. of results in triplicate experiments. *P* values were calculated using a two-sided, unpaired Student's *t*-test (n.d., not detectable).

active in CD4+ T cells, enabling viral persistence by evading immune detection. A reporter assay in different cell lines confirmed the strongest OCR-mediated suppression in T cells, not B cells or other lineages (Extended Data Fig. 4a). We additionally analysed the silencer function of the OCR on the 5′-LTR using primary CD4+ T cells and immortalized macrophage-like cells derived from induced pluripotent stem cells (Extended Data Fig. 4b).

To identify the RUNX1 domain responsible for silencing, we performed a 5′-LTR reporter assay with wild-type and mutant RUNX1 proteins[29]. Mutations in the DNA-binding Runt domain, disrupting DNA binding, transcriptional activity or CBFβ interaction, impaired silencing (Fig. 2f). Overexpressing RUNX1 in ATL and HTLV-1-infected cell lines reduced endogenous *tax* expression at both RNA and protein levels (Fig. 2g,h). Conversely, RUNX1 knockdown diminished 5′-LTR suppression in Jurkat and Molt4 cells (Fig. 2i and Extended Data Fig. 4c–e). Previously reported ATL-associated RUNX1 mutants retained silencer function (Extended Data Fig. 4f).

### Virological and immunological significance of the OCR

We investigated the functional role of the OCR from a virological perspective. First, we identified mutations in the OCR that retained the *pol* gene coding sequence but lost transcriptional repression. Four nucleotide substitutions in RUNX1 binding sites abrogated silencing in the 5′-LTR promoter assay (Fig. 3a). We then generated a recombinant HTLV-1 with these mutations and established JET cells infected with either wild-type (wt) or silencer mutant (s-mut) HTLV-1 (Fig. 3b). Binding of RUNX1, CBFβ and HDAC3 to the silencer region significantly decreased in s-mut-infected cells (Fig. 3c). Consistently, *tax* expression was markedly upregulated in s-mut-HTLV-1-infected cells, whereas HBZ levels increased but less prominently (Fig. 3d).

Next, we analysed viral productivity. The s-mut virus produced significantly more virus in culture supernatants, while wt-HTLV-1 did not, even with phorbol 12-myristate 13-acetate (PMA)–ionomycin stimulation (Fig. 3e). To assess viral persistence, we cultured infected Jurkat T cells (Fig. 3b) for an extended period. The PVL decreased in wt-HTLV-1-infected cells but increased in s-mut-infected cells, consistent with higher viral gene expression and productivity while maintaining provirus as the dominant form of viral DNA (Fig. 3f and Extended Data Fig. 5a). In line with this, active histone modifications (H3K4me3, H3K9ac) at the 5′ side of provirus were reduced in wt- but not s-mut-HTLV-1-infected cells (Extended Data Fig. 5b). Chromatin conformation capture showed closer 5′-LTR-OCR proximity in wt- compared with s-mut-infected cells (Extended Data Fig. 5c,d), suggesting that the silencer complex influences epigenetic regulation of HTLV-1 in addition to the proviral insulator region.

Given the role of OCR in silencing, we examined whether it facilitates immune evasion by minimizing viral antigen expression. Using Ro5-3335, a RUNX1 inhibitor[30], we assessed its impact on viral antigen expression. The drug showed no cytotoxicity at the tested concentration in HTLV-1- or HIV-1-infected cell lines (Extended Data Fig. 6a). In JET cell clones, Ro5-3335 significantly increased tdTomato expression in a dose-dependent manner (Fig. 3g and Extended Data Fig. 6b). As reported previously[31], high-dose Ro5-3335 modestly increased HIV-1 proviral expression (Extended Data Fig. 6b,c).

Tax expression, typically undetectable in freshly isolated PBMCs from HTLV-1-infected individuals but inducible by ex vivo culture[32], was enhanced by Ro5-3335 (Fig. 3h and Extended Data Fig. 6d).

To determine whether OCR impairment increases susceptibility to cytotoxic T lymphocyte (CTL) responses, we compared Jurkat T cells infected with wt- or s-mut-HTLV-1. We used HTLV-1 Tax as the viral antigen for presentation by HLA-class I (HLA-A*24:02), a well-characterized and dominant HTLV-1 epitope[33]. We transduced the Tax-specific TCRs identified in the previous study[34] into primary T cells and co-cultured with JET cells infected with the wt or the s-mut virus. We performed an enzyme-linked immunospot (ELISPOT) assay and found that infected cells with the s-mut virus showed higher immunogenicity than those with the wt virus (Fig. 3i and Extended Data Fig. 6e). Notably, the RUNX1 inhibitor treatment increased the susceptibility of infected cells against anti-viral CTL in both cytokine production and cytotoxicity assay by using anti-Tax-specific CTLs (Extended Data Fig. 6f,g). Although we cannot exclude the possibility that off-target effects of the RUNX1 inhibitor may contribute to the experimental outcome, these results establish the immunological significance of the OCR in controlling the viral antigen Tax.

### Single-cell analysis of CD4+ T cells from people with HTLV-1

Although HTLV-1-infected cells typically remain latent, the silenced provirus can be reactivated, triggering a transcriptional burst[14,32] (Fig. 3h). This reactivation facilitates viral spread and de novo infection, although in vivo evidence is lacking. We cultivated primary CD4+ T cells from three indolent types of ATL case ex vivo overnight to induce transcriptional burst and performed multiome, simultaneous single-cell RNA sequencing (sc-RNA-seq) and sc-ATAC-seq analysis using cells both before and after cultivation. These patients with ATL contained the oligoclonal expansion of infected cells and retained the reactivation ability of *tax* expression after ex vivo culture (Extended Data Fig. 7a–d). Subsequently, we conducted cell clustering analysis by integrating sc-RNA-seq and sc-ATAC-seq data (Fig. 4a,b and Extended Data Figs. 8a,b and 9a,b). Infected cells were identified by their transcriptional characteristics (Fig. 4c and Extended Data Figs. 8c and 9c; see Methods for more details), and we further refined cell clustering using only the infected cells. Two distinct clusters emerged: one exclusively post-cultivation and another present both pre- and post-cultivation. The former corresponded to a transcriptional burst cluster, while the latter remained latent. Both clusters highly expressed CADM1, a marker of HTLV-1-infected cells[35,36] (Fig. 4f and Extended Data Figs. 8f and 9f).

ATAC-seq analysis revealed high chromatin openness across the HTLV-1 proviral region in the burst cluster, consistent with active transcription (Fig. 4g–h, Extended Data Figs. 8g–h and 9g–h). By contrast, the latent cluster showed three ATAC-seq peaks between the 5′- and 3′-LTRs, aligning with known regulatory regions: the OCR, insulator[18] and enhancer[17].

We further examined gene expression dynamics of transcription factors during transcriptional bursts using sc-multiome data. Expression of RUNX1 and ETS1 decreased, whereas that of GATA3 and Sin3A increased (Fig. 4i and Extended Data Figs. 9i and 10i). Even though the expression level of RUNX1 was decreased, there was a moderate decrease in RUNX1 TF motif activity in the burst cell population

---

**Fig. 4 | Single-cell multiome analysis of CD4+ T cells from a smouldering ATL case (a5). a,b**, WNN UMAP projection of all cells, both before and after cultivation identified using multimodal neighbours by weighting a combination of ATAC and RNA data. Cells are labelled by cell type (**a**) and before or after cultivation (**b**). **c**, Heatmaps of expression levels of viral RNA and CADM1, a marker of HTLV-1-infected cells. **d,e**, WNN UMAP projection of only HTLV-1-infected cells, both before and after cultivation identified using multimodal neighbours by weighting a combination of ATAC and RNA data. Cells are labelled by cell type (**d**) and before or after cultivation (**e**). **f**, Heatmaps of expression levels of viral RNA and CADM1, a marker of HTLV-1 infected cells. **g**, Aggregated and normalized ATAC signals of burst and latent cell clusters in the proviral region. **h**, Violin plots of viral RNA expression in burst and latent cell clusters. **i**, Violin plots of infected cells and UMAP projections of all cells with a heatmap showing the RNA expression level of transcription factors related to the OCR in burst and latent cell clusters. **j**, Heatmaps of RUNX1 motif activity and a correlation plot showing the relationship between the sense transcription of HTLV-1 (horizontal axis) and the motif activity of RUNX1 (vertical axis). The blue line represents the regression line. The plot is labelled for burst and latent clusters. **k**, Schematic figure of the OCR-mediated silencing in HTLV-1 provirus.

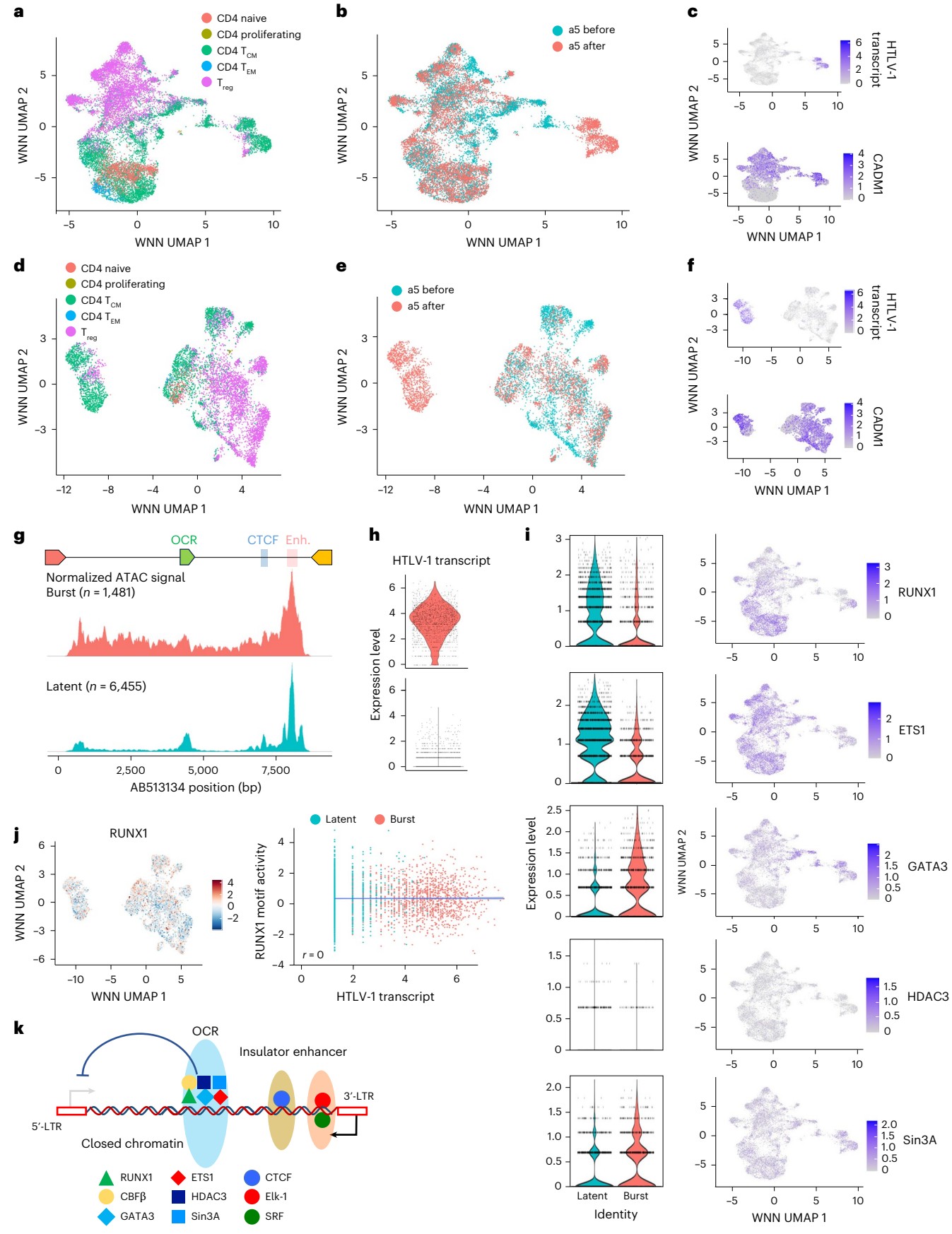

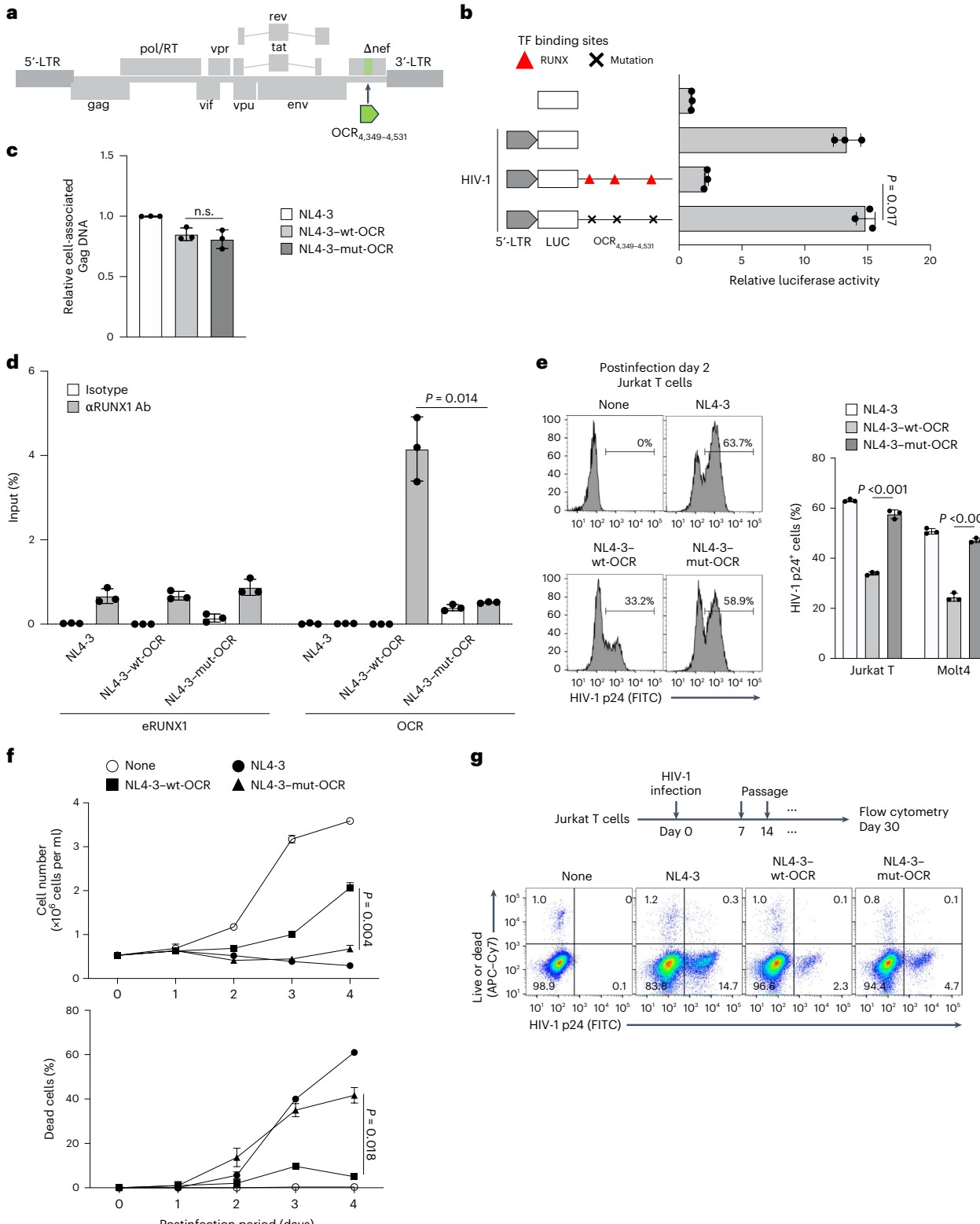

**Fig. 5 | Introduction of HTLV-1 OCR into rHIV-1 decreases proviral expression, virus production and cytopathic effect. a**, Schematic representation of the rHIV-1 construction with wt- or mt-OCR derived from HTLV-1. The NL4-3 plasmid was used to generate the rHIV-1 constructs. **b**, Effect of HTLV-1 wt- or mt-OCR on the HIV-1 5′-LTR promoter activity. Luciferase reporter assays were performed using Jurkat T cells 48 h after transfection. **c**, Quantification of cell-associated viral DNAs after infection with each rHIV-1. gDNAs were extracted from Jurkat T cells infected with each rHIV-1. **d**, RUNX1 ChIP–qPCR analysis of Jurkat T cells infected with rHIV-1 with wt- or mt-OCR. Enhancer RUNX1 region (eRUNX1) was used as a positive control for ChIP. **e**, HIV-1 production in Jurkat T cells infected with rHIV-1. Intracellular HIV-1 p24⁺ cells were detected using flow cytometry. **f**, Assessment of the cytopathic effect induced by rHIV-1. Cell viability and dead cell counts were determined using the trypan blue exclusion method. **g**, A representative workflow and flow cytometry results of the WIPE assay using each rHIV-1 are presented. At least two independent experiments were performed. The bars and error bars represent the mean ± s.d. of results in triplicate experiments. P values were calculated using a two-sided unpaired Student's t-test.

(Fig. 4j and Extended Data Figs. 9j and 10j), indicating that changes in expression of cofactors, such as ETS1 and GATA3, can be key determinants for the silencer function. In particular, ETS1 strongly upregulates transcriptional activity mediated by the 5′-LTR and OCR, making it a crucial molecular switch for RUNX complex function (Fig. 2c)[15]. Further studies are needed to elucidate how intragenic regulatory elements—silencer, insulator and enhancer—coordinate proviral expression and retroviral latency in the host.

### Effect of the HTLV-1 OCR on the virological aspect of HIV-1

We tested the hypothesis that the RUNX1-mediated HTLV-1 silencer functions as a dominant silencer in both HIV-1 and HTLV-1 (Fig. 1a). Here we aimed to analyse the effect of the OCR on the HIV-1 5′-LTR (Fig. 5a) by constructing a novel recombinant HIV-1 (rHIV-1) molecular plasmid clone. This involved inserting either the wt or the mutated OCR into the HIV-1 *nef* region (Fig. 5a,b), which is a well-established region for the insertion of external genes of interest, such as reporter fluorescent protein[37]. Although this region is distant from the 5′-LTR in the HIV provirus, chromatin looping has been reported between the 5′- and 3′-LTRs during active transcription of HIV-1 (ref. 38), suggesting that the OCR in the *nef* region may suppress HIV-1 5′-LTR activity.

Using the NL4-3 plasmid[39], we generated rHIV-1 containing either wt- or HTLV-1 silencer mutant (mt)-OCR and conducted single-round infections in Jurkat T cells. One day postinfection, intracellular HIV-DNA levels showed no significant differences among the viruses (Fig. 5c). RUNX1 ChIP-qPCR confirmed RUNX1 binding to the inserted silencer region in the integrated provirus (Fig. 5d). The level of intracellular p24 gag expression at 2 days postinfection was dramatically decreased in the rHIV-1 with the wt-OCR but not that with the mt-OCR lacking RUNX1 binding capacity (Fig. 5e). Notably, the cytopathic effect induced by viral infection was significantly decreased by insertion of the wt-OCR but not by the mutated OCR (Fig. 5f).

To assess the role of OCR in viral persistence, we used the widely distributed intact provirus elimination (WIPE) assay[40]. Introduction of the OCR remarkably decreased viral persistence in the WIPE assay, but the effect was partially cancelled by mutation in the RUNX-binding sites (Fig. 5g). These findings show that OCR exerts a dominant silencing effect on HIV-1, highlighting its role in shaping distinct infection outcomes for HIV-1 and HTLV-1.

## Discussion

HTLV-1 spontaneously induces latent infection even without antiretroviral drugs, in sharp contrast to HIV-1 (Fig. 1a). HTLV-1 transcription from the 5′-LTR is suppressed by DNA methylation and/or deletion of the 5′-LTR in ATL cells and transformed infected cells[21,41]. Apart from such cases, infected cells generally retain a complete virus sequence, in which proviral expression is reversibly silenced and can be reactivated[32,42]. From a viral perspective, this property is crucial for de novo infection and persistent human infection. However, before this study, the mechanism underlying selective and reversible silencing of HTLV-1 5′-LTR transcription was unclear.

We identified a previously unrecognized transcriptionally suppressive region between the 5′-LTR and the insulator region, serving as a binding site for RUNX1 and its cofactors to mediate viral silencing. This molecular mechanism may maintain HTLV-1 latency most of the time. In addition, the HTLV-1 silencer element has little effect on antisense transcription from the 3′-LTR (Fig. 1f,g). The CTCF-binding site in the pX region probably functions as an insulator between the 5′- and 3′-LTRs. These mechanisms sustain continuous HBZ expression, which is known to contribute to the proliferation of infected cells[16], while maintaining suppression of viral antigen transcription from the HTLV-1 5′-LTR. However, this latency can be cancelled by extracellular stimulation or stress, which induces changes in glucose metabolism and oxygen tension, resulting in transcriptional burst from the HTLV-1 provirus[43].

We also provide evidence that the HTLV-1 silencer element helps re-establish latency post-transcriptional burst, as shown by sc-multiome analysis of fresh PBMCs from infected individuals. Several negative regulatory mechanisms for HTLV-1 sense transcription induced by regulatory and accessory viral proteins were reported previously, including HBZ, Rex, p30 and others[44–46]. Beyond these, HTLV-1 exploits the host factor RUNX1 to establish latency, evade immunity and maintain persistence. We propose a hypothetical schematic model illustrating the OCR-mediated HTLV-1 latency mechanism (Fig. 4k and Extended Data Fig. 10), but they do not represent the data. Considering the heterogeneity of the host cell status and integration site in each infected clone, further experiments will clarify how these regulatory factors independently or cooperatively determine the fate of the individual infected cell, leading to either a transcriptional burst or latency.

Notably, rHIV-1 containing the HTLV-1 silencer element showed a remarkable shift from vigorous virus production to the latent-like phenotype as HTLV-1. Despite recent advances in antiretroviral therapy, eradicating HIV-1 in people living with the virus remains highly challenging. Latently infected cells, though rare, persist and form viral reservoirs despite treatment. Our study suggests that HTLV-1 latency is mainly driven by proviral mechanisms, although integration site context also matters[47]. By contrast, HIV-1 lacks such regulatory elements, implying two key differences: (1) HIV-1 is less likely to establish latency than HTLV-1, and (2) HIV-1 latency relies more on the host's genetic and epigenetic landscape, consistent with previous reports[48].

In summary, we identified mechanisms governing reversible HTLV-1 latency, distinct from those in human endogenous retroviruses and the murine leukaemia virus[1–3], advancing our understanding of retroviral biology. Furthermore, using a reverse-genetics approach with HTLV-1 and HIV-1 recombinant viruses has enabled us to elucidate unique mechanisms of silencing and reactivation of these retroviruses. This comparative reverse-genetics analysis fills a significant gap in our current understanding, aligning with recent genetic and epigenetic studies that investigated silencing mechanisms in HIV-1 (refs. 48,49). Our study also reveals how HTLV-1 leverages the proliferation of infected T cells over viral replication and spreading, the underlying mechanism of which is not fully understood until now[9,50]. By elucidating the roles of the transcription factors (RUNX1, GATA3 and ETS1) in the control of the reversible viral latency, we have identified potential targets for therapeutic intervention, opening new avenues for research and treatment strategies in viral latency and oncogenesis.

Our discovery highlights a key difference in HIV-1 and HTLV-1 silencing mechanisms, driven by a short regulatory sequence recruiting three transcription factors with specific roles in T cells[51]. These insights deepen our understanding of viral evolution and set the stage for future studies on virus–host co-evolution.

## Methods

### Ethics statement

All protocols involving human participants were reviewed and approved by the Kumamoto University Institutional Review Board (approval numbers 248 and 263). The study was carried out in accordance with the guidelines proposed in the Declaration of Helsinki. Informed written consent was obtained from all participants in this study. All participants were recruited as volunteers.

### Cell culture

The T cell lines (Jurkat, Molt4, Kit225(+), MoT and Si-2) were obtained from the American Type Culture Collection (TIB-152, CRL-1582, CRL-1990, CRL-8066) or the Japanese Collection of Research Bioresources Cell Bank (JCRB1321). Foetal lamb kidney (FLK)-BLV was provided by Y. Aida (The University of Tokyo)[52]. CEM and THP1 cell lines were obtained from H. Takeuchi (Tokyo Science University). The 293T, PG13, HeLa and K562 cell lines were obtained from the American Type Culture Collection (CRL-3216, CRL-10686, CCL-2 and CCL-243,

respectively). The BCBL1, BC-2, KMS-12-PE and YT-1 cell lines were gifted by S. Okada (Kumamoto University). The iPS cell-derived immortalized macrophage-like (iPS-ML) cells were gifted by S. Suzu (Kumamoto University). The ATL cell line MT1 was obtained from M. Maeda (Kyoto University)[53]. The HTLV-1-infected cell line (TBX-4B) was obtained from C.R.M. Bangham (Imperial College London)[54]. JET cells are Jurkat T cells expressing tdTomato under the control of five tandemly repeated Tax-responsive element[55]. The wt39 and wt51 cells were established from JET cells infected with the HTLV-1 molecular clone by limiting dilution in a previous study[17]. Suspension cell lines such as T cell lines except MoT were cultured in complete RPMI-1640 medium (developed by Roswell Park Memorial Institute; Nacalai Tesque) supplemented with 10% foetal bovine serum (FBS), 100 U ml$^{-1}$ penicillin and 100 µg ml$^{-1}$ streptomycin. MoT cells were cultured in Iscove's modified Dulbecco's medium supplemented with 20% FBS. The iPS-ML cells were cultured in complete Alpha MEM-20% foetal calf serum (FCS) containing 100 ng ml$^{-1}$ recombinant human macrophage colony-stimulating factor (PeproTech) and 10 ng ml$^{-1}$ recombinant human granulocyte–macrophage colony-stimulating factor (PeproTech) as described previously[56]. Adherent cell lines such as 293T cells and retroviral packaging cell lines (Plat-GP and PG13 cells) were cultured in DMEM medium (Nacalai Tesque) supplemented with 10% FBS, 100 U ml$^{-1}$ penicillin and 100 µg ml$^{-1}$ streptomycin. Blasticidin S (Nacalai Tesque) was added to DMEM medium for Plat-GP cells. TBX-4B cells were cultured in complete RPMI-1640 medium supplemented with IL-2 (200 U ml$^{-1}$; PeproTech).

## Viral RNA measurement in plasma

Plasma samples were collected from HIV-1-infected individuals before antiretroviral therapy. Plasma viral RNA levels were measured using the COBAS AmpliPrep/COBAS TaqMan HIV-1 Test version 2.0 (Roche Diagnostics). The National Center for Global Health and Medicine Ethics Committee approved this study (NCGM-A-003161-01), and each donor living with HIV-1 provided written informed consent. The age, number and clinical information of the participants are provided in Supplementary Table 1.

We extracted RNA from 50–140 µl of plasma separated from heparinized blood of asymptomatic HTLV-1 carriers. RNA was extracted using the QIAamp Viral RNA Mini Kit (Qiagen) according to the manufacturer's instructions and treated with DNase I. cDNA was synthesized using a ReverTra Ace qPCR RT Master Mix (Toyobo) according to the manufacturer's instructions. Droplet digital PCR (ddPCR) was performed using primers and a probe targeting the HTLV-1 *tax* region. PCR was performed in a QX200 Droplet Digital PCR system (Bio-Rad) with the following settings: 95 °C for 10 min, followed by 39 cycles of 94 °C for 30 s, 58 °C for 60 s and final 98 °C for 10 min and 4 °C for hold. Data were analysed using QuantaSoft software (Bio-Rad). Primer sequences are listed in Supplementary Table 5. The age, number and clinical information of the participants are provided in Supplementary Table 2.

## ATAC-seq

ATAC-seq was performed as described previously[57]. Briefly, cells (1–5 × 10$^6$) were treated with cold lysis buffer (10 mM Tris−HCl (pH 7.5), 10 mM NaCl, 3 mM MgCl$_2$, 0.1% NP-40, 0.1% Tween-20 and 0.01% digitonin). After lysis, the cells were treated with Tn5 Transposase (Nextera) at 37 °C for 30 min and the treated DNAs were purified using a Qiagen MinElute kit. Sample libraries were prepared using the NEBNext Ultra II DNA Library Prep Kit for Illumina and Multiplex Oligos for Illumina (New England Biolabs). Library efficiency was quantified using P5 and P7 primers and sequenced using Illumina NextSeq[42].

## ChIP assay

ChIP assays were performed using the SimpleChIP Enzymatic Chromatin IP Kit (Cell Signaling Technology) according to the manufacturer's instructions. Briefly, cells (4 × 10$^6$) were fixed in 1% formaldehyde for 10 min at room temperature, quenched with glycine solution and washed in ice-cold PBS. Nuclei were extracted by lysing cells with lysis buffer (buffer A). Samples were then digested with MNase for 20 min at 37 °C and sonicated using Bioruptor UCD-300 (Tosyodenki) with a cycle of 30 s on and 30 s off for 5–8 min to disrupt the nuclear membrane. Extracted chromatin was immunoprecipitated using anti-RUNX1 (Abcam ab23980), anti-CBFβ (Abcam ab195411), anti-GATA3 (CST 5852P), anti-HDAC3 (Invitrogen PA5-85378), anti-ETS1 (CST 14069S), anti-Sin3A (CST 7691), anti-H3K4me3 (CST 9751) and anti-H3K9Ac (Millipore 06-942). All antibodies were used at a 1:250 dilution.

ChIP-qPCR was performed to validate each ChIP assay. For observing the epigenetic changes in JET cells infected with wt-HTLV-1 or s-mut-HTLV-1, H3K4me3 and H3K9Ac ChIP DNA were quantified (relative to input DNA). Eight primer pairs spanning the HTLV-1 genome were used for relative quantification of ChIP DNA[18].

For ChIP-seq, library synthesis from ChIP DNA was performed as described in the ATAC-seq section. To increase the detection sensitivity, we enriched the ChIP DNA libraries with the HTLV-1 DNA capture-seq method[42]. Briefly, ChIP DNA library was hybridized with biotinylated DNA probes spanning the whole HTLV-1 provirus. Then, HTLV-1-enriched sequences were captured using streptavidin beads. The enriched libraries were sequenced by Illumina Miseq.

## ATAC-seq and ChIP-seq data analysis

Paired or single-end read adaptor sequences were trimmed using Cutadapt (version 1.18) from the FASTA files. The reads were then cleaned with PRINSEQ (version prinseq-lite-0.20.4). The resulting reads were mapped to the custom genome hg19 and HTLV-1 AB513134 using BWA-MEM. BAM files were filtered using SAMtools (version 1.18) and Picard command line (http://broadinstitute.github.io/picard/) to obtain the BAM file containing reads aligned to HTLV-1. For mapping ATAC-seq data from cells other than HTLV-1-infected cells, we used the appropriate host genome and provirus genome separately to prepare a custom reference genome. The host genome for J1.1, ACH2 and MoT is hg19 while the host genomes for Si-2 and FLK-BLV are macFas5 and bosTau9, respectively. The provirus genomes for these cell lines are HXB2 (K03455.1), HXB2 (K03455.1), HTLV-2 (LC534557.1), STLV-1 (JX987040.1) and BLV (EF600696), respectively. BAM files were indexed using SAMtools and visualized using Integrative Genomic Viewer.

## Luciferase reporter assay

Inserted sequences for luciferase reporter assay are listed in Supplementary Table 5. Promoter constructs from HTLV-1 and HIV-1 were inserted into multiple cloning sites of pGL4.10 (Promega), which contains the luciferase reporter gene. Three randomly selected regions in the HTLV-1 genome (1,500, 3,000 and 6,000), OCR and its mutants were inserted into the BamHI restriction site (downstream of the luciferase reporter gene and SV40 poly-A signal) in pGL4.10. Trans-activator Tax (gifted by J. Fujisawa) and Tat (gifted by Y. Ariumi) were inserted into pCG and pcDNA3.1, respectively, with expression driven by the CMV promoter. Human RUNX genes and its mutants were gifted by M. Osato (National University of Singapore). Mutated RUNX1 sequences (P187R and F262V) in patients with ATL were obtained from a whole genome sequencing dataset[58] and inserted into the pEF-BOS plasmid. Jurkat T cells (2 × 10$^5$) or 293T cells (1 × 10$^5$) were transfected with a total of 1 µg of plasmids using Lipofectamine LTX with Plus Reagent (Thermo Fisher Scientific). For primary cells, CD4$^+$ T cells were purified from PBMCs of healthy donors using a Pan T Cell Isolation kit (Miltenyi Biotec) and biotin-labelled anti-CD8 mAb (RPA-T8; Biolegend) according to the manufacturer's instructions. The purified CD4$^+$ T cells were stimulated with Dynabeads Human T-Activator CD3/CD28 (Thermo Fisher Scientific) and cultured in complete RPMI-1640 medium supplemented with IL-2 (100 U ml$^{-1}$) and IL-15 (10 ng ml$^{-1}$). After 48 h, stimulated T cells were electroporated using NEPA21 (NEPAGENE) and a 2-mm-gap cuvette (EC-002S, NEPAGENE). The electroporation programme was as follows: 275 V, 1 ms and 6 times with a 50-ms interval

for the poring pulse, and 20 V, 50 ms and 3 times with a 50-ms interval for the transfer pulse. iPS-ML cells were also electroporated in a similar way to primary CD4[+] T cells. After 48 h of the transfection or electroporation, luciferase assays were performed using the Dual-Glo Luciferase Assay System (Promega) according to the manufacturer's instructions, and luminescence was measured using a GloMax 20/20 Luminometer (Promega).

## Reverse-transcription quantitative PCR

RNA was extracted using the RNeasy Mini Kit (Qiagen) according to the manufacturer's instructions and treated with DNase I. cDNA was synthesized using ReverTra Ace qPCR RT Master Mix (Toyobo) according to the manufacturer's instructions. qPCR was performed using KOD SYBR qPCR mix (Toyobo) and run on an Applied Biosystems StepOnePlus Real-Time PCR System (Thermo Fisher Scientific). Primers used are listed in Supplementary Table 5.

## Generation of retroviral vectors for RUNX1 expression

The pMXs plasmid was used as transfer plasmid for the generation of retroviral vectors. To detect and sort the transduced cells, the sequences encoding human RUNX1 and truncated nerve growth factor receptor (ΔNGFR) were fused with P2A (self-cleaving 2A peptides derived from porcine teschovirus-1). Gene transduction by retroviral vectors was performed using the Plat-GP and PG13 cell-based retrovirus system and Retronectin (Takara Bio) according to the manufacturer's instructions. Transduced MT-1 and TBX-4B cells were sorted using PE-labelled anti-NGFR mAb (ME20.4, BioLegend) and anti-PE Micro-Beads (Miltenyi Biotec) according to the manufacturer's instructions. To analyse Tax protein levels, RUNX1-transduced cells were treated with a Foxp3 transcription factor staining buffer set (Thermo Fisher Scientific) and then stained with fluorescein isothiocyanate (FITC)-labelled anti-Tax monoclonal antibody (Lt4) for 30 min at room temperature. Dead cells were distinguished using the LIVE/DEAD Fixable Near-IR Cell Stain Kit (Thermo Fisher Scientific). Stained cells were analysed with FACSVerse (BD Biosciences), and data analysis was performed using FlowJo software (version 10.7.1, Tree Star).

## Western blot

Western blotting was performed as described previously[59]. Cells were collected, washed with PBS twice and lysed in lysis buffer (25 mM HEPES (pH 7.2), 20% glycerol, 125 mM NaCl, 1% Nonidet P40 (NP40) substitute (Nacalai Tesque)). After the quantification of total protein using a protein assay dye (Bio-Rad), lysates were diluted with 2× SDS sample buffer (100 mM Tris–HCl (pH 6.8), 4% SDS, 12% β-mercaptoethanol, 20% glycerol, 0.05% bromophenol blue) and boiled for 10 min. Proteins in the cell lysates (5 μg of total protein) were separated using sodium dodecyl sulfate-polyacrylamide gel electrophoresis (SDS-PAGE) and transferred to polyvinylidene difluoride (PVDF) membranes (Millipore). Membranes were blocked with 5% milk in PBS containing 0.1% Tween 20 (0.1% PBST) and incubated in 4% milk and 0.1% PBST containing primary antibodies: mouse anti-RUNX1 (A-2; Santa Cruz Biotechnology) in 1:2,000 dilution and rabbit anti-beta actin (13E5; Cell Signaling Technology) in 1:5,000 dilution. Subsequently, the membranes were incubated with horseradish peroxidase (HRP)-conjugated secondary antibodies: donkey anti-rabbit IgG-HRP (Jackson ImmunoResearch) and donkey anti-mouse IgG-HRP (Jackson ImmunoResearch). Each secondary antibody was used in 1:50,000 dilution. SuperSignal West Femto Maximum Sensitivity Substrate (Thermo Fisher Scientific) or SuperSignal Atto (Thermo Fisher Scientific) was used for HRP detection. Bands were visualized using the Amersham Imager 600 (Amersham).

## RUNX1 knockdown by small hairpin RNA

To suppress RUNX1 expression in Jurkat T cells, we constructed small hairpin RNA (shRNA)-expressing retroviral vectors using pSINsi-hU6 (Takara Bio). The generation and transduction of retroviral vectors for shRNA-RUNX1 were carried out using the Plat-GP cell-based retrovirus system and Retronectin (Takara Bio) according to the manufacturer's instructions. Sequences of shRNA used are listed in Supplementary Table 5. At 48 h post-transfection in Jurkat T cells, RUNX1 protein levels were analysed using PE-labelled anti-RUNX1 antibodies (RXDMC, Invitrogen) following pretreatment with a Foxp3 transcription factor staining buffer set (Thermo Fisher Scientific). Stained cells were analysed using FACSVerse (BD Biosciences).

## Generation of recombinant HTLV-1

Generation of recombinant HTLV-1 was performed as reported previously[17]. Briefly, a 1.2-kbp region in the HTLV-1 *pol* gene containing RUNX binding site mutations, introduced by site-directed mutagenesis, was amplified using PCR. The amplicon was inserted in pX1 MT-M using XbaI and SphI restriction sites. The 293T cells were transfected with wt or s-mut (RUNX binding site mutant) HTLV-1 molecular clone pX1 MT-M by polyethylenimine, and then irradiated with 30 Gy. The irradiated 293T cells were co-cultured with JET cells for 3 days. tdTomato-positive cells were then sorted using FACSAria (Becton, Dickinson and Company) and cultured in complete RPMI medium.

## p19 ELISA

JET cells ($1 \times 10^5$) infected with either the wt- or s-mut-HTLV-1 were suspended in complete RPMI medium and seeded into a 48-well plate. To enhance p19 production, infected JET cells were stimulated with 100 ng ml$^{-1}$ PMA, 2 μM ionomycin and 10 ng ml$^{-1}$ TNF. After 48 h, the supernatant was collected and p19 was measured using RETROtek HTLV p19 Antigen ELISA (ZeptoMetrix) following the manufacturer's instructions.

## PVL measurement

For JET cells, the PVL or total cell-associated viral DNA was calculated by quantifying the copy number of the *tax* gene, normalized to the copy number of the albumin (*ALB*) gene, using ddPCR as previously described with minor modifications[42]. PVL or total cell-associated viral DNA was calculated as follows: PVL (%) = [(copy number of *tax*)/(copy number of *ALB*)/2] × 100. The copy number of 2 LTR circles was also quantified using ddPCR. For Jurkat T cells infected with rHIV-1, *gag* and 2 LTR circles of HIV-1 were quantified to calculate the total or an episomal form of cell-associated viral DNA. Primer sequences are listed in Supplementary Table 5.

## Generation of retroviral vectors for TCR transduction

Transfer genes were inserted into the pMXs plasmid. To detect and sort the transduced cells, the sequences encoded by *TCRαβ* and truncated epidermal growth factor receptor (ΔEGFR) or human RUNX1 and ΔNGFR fused with P2A. Gene transduction by retroviral vector was performed using the Plat-GP and PG13 cell-based retrovirus system and Retronectin (Takara Bio) according to the manufacturer's instructions.

## Tax-specific TCR cloning and its reconstitution in primary T cells

HLA-A*24:02 (A24)-restricted Tax$_{301-309}$-specific TCR-1 and TCR-2 were cloned from CD8[+] T cells of one patient with HAM[34]. To detect Tax-specific CD8[+] T cells, we stained whole CD8[+] T cells from the patient with HLA-A24−Tax$_{301-309}$ dextramer (Immudex). Sequences of A24−Tax TCR-1 (α-chain CDR3: CAVIGGGFKTIF; β-chain CDR3: CASKPN-RADTQYF) and TCR-2 (α-chain CDR3: CATDAAGNKLTF; β-chain CDR3: CASIRDREETQYF) were obtained by RNA-seq (Azenta Life Sciences). To assess specificity against the Tax antigen, TCR-deficient Jurkat T cells were retrovirally transduced with the *TCRαβ* gene. K562 cells were also retrovirally transduced with the HLA-A*24:02 gene and pulsed with Tax$_{301-309}$ (Genscript) at 10 μg ml$^{-1}$ as antigen-presenting cells (APCs). The TCR-transduced Jurkat T cells and APCs were cocultured

for 24 h, and then, IL-2 production was analysed using the human IL-2 ELISPOT Kit (Mabtech). Human primary T cells were purified using the Pan T Cell Isolation kit (Miltenyi Biotec) according to the manufacturer's instructions. Purified T cells were stimulated with Dynabeads Human T-Activator CD3/CD28 (Thermo Fisher Scientific) and cultured in complete RPMI-1640 medium supplemented with IL-2 (100 U ml$^{-1}$) and IL-15 (10 ng ml$^{-1}$). After 48 h, stimulated T cells were transduced with retroviral vector expressing A24–Tax-specific TCRs by using a Retronectin-coated plate (Takara Bio). In 2–3 weeks post-stimulation, the CD8$^+$ T cells (10$^4$ cells) were co-cultured with Jurkat T cells as well as JET-wt and JET-s-mut cells (2 × 10$^5$ cells) transduced with HLA-A*24:02, and then cytokine production was detected using the human IFNγ ELISPOT Kit (Mabtech) and the ELIPHOTO Counter (Minerva Tech).

## RUNX1 inhibitor treatment

Ro5-3335 (RUNX1 inhibitor) was purchased from Tocris Bioscience and dissolved in dimethyl sulfoxide (Sigma). HTLV-1-infected JET cell clones (wt39 and wt51), J-Lat cells (9.2 and 10.6) and PBMCs from asymptomatic carriers (ACs) and patients with ATL were treated with Ro5-3335 for 24 h. Tax expression in JET cell clones and HIV-1 5'-LTR reactivation in J-Lat cells were measured by tdTomato and GFP expression, respectively. Ro5-3335-treated PBMCs were stained with PerCP–Cy5.5-labelled anti-CD4 mAb (OKT4, BioLegend) and FITC-labelled anti-Tax mAb. Dead cells were distinguished using the LIVE/DEAD Fixable Near-IR Cell Stain Kit. Cell viability in the treated cells was counted using the trypan blue dye exclusion method. Stained cells were analysed using FACSVerse (BD Biosciences) and SH800S Cell Sorter (Sony Biotechnology).

## CTL assay

TCR-transduced CD8$^+$ T cells were prepared by retroviral transduction of primary CD8$^+$ T cells with mock, A24–Tax TCR-1 and TCR-2. After 3 days of retroviral transduction, the CD8$^+$ T cells were sorted using PE-labelled anti-EGFR mAb (AY13, BioLegend) and anti-PE MicroBeads (Miltenyi Biotec) according to the manufacturer's instructions and expanded in complete RPMI-1640 medium supplemented with IL-2 and IL-15 for 2 weeks. Target cells (5 × 10$^4$ cells) were treated with 50 µM Ro5-3335 (RUNX1 inhibitor) for 24 h and co-cultured with TCR-transduced CD8$^+$ T cells for 5 h. The cells were stained with propidium iodide (PI) to detect dead cells, BV510-labelled anti-CD8 mAb (RPA-T8; Biolegend) and APC-labelled anti-EGFR mAb (AY13; Biolegend) to exclude effector cells. Dead cells were detected using FACSVerse (BD Biosciences) as PI(+), CD8a(−), EGFR(−) cells.

## Chromatin conformation capture

Two million cells were cross-linked with 1% formaldehyde at room temperature for 10 min, quenched with glycine solution and washed with ice-cold PBS. Cells were lysed for 1 h at 4 °C with lysis buffer (10 mM Tris–HCl (pH 8.0), 10 mM NaCl, 0.2% Nonidet P-40 and Protease Inhibitor Mixture (Thermo Fisher Scientific)). For genomic DNA (gDNA) extraction, the manufacturer's instructions for the Qiagen Blood and Tissue Kit were followed. gDNA and lysed nuclei were digested with 600 units of DpnII (NEB) overnight at 37 °C. The digested DNA or chromatin was then diluted in 5 ml of nuclease-free water and ligated with 400 units of T4 DNA ligase (NEB) for 2 h at 4 °C. The cross-links in the chromatin were reversed with proteinase K at 65 °C for 5 h. The DNA was purified using phenol–chloroform extraction and ethanol precipitation following the manufacturer's instructions (Phenol:Chloroform:Isoamyl Alcohol, Nacalai Tesque). DNA precipitate was resuspended in 300 µl of nuclease-free water. Then, 2 µl of the purified DNA was used for ddPCR to quantify the LTR-OCR fragment, albumin and HTLV-1 *tax*. Finally, the LTR-OCR fragment frequency was calculated per HTLV-1 *tax*.

## Generation and infection of rHIV-1

pNL43-HIV-env (−) OCR and pNL43-HIV-env (−) mut-OCR were derived from pNL4-3/ΔBglII (ref. 60). Briefly, the *env* gene was deleted between

the BglII restriction sites. The NotI restriction site was inserted at the 5'-end of *nef* by site-directed mutagenesis. The *OCR* gene (183 bp) was amplified by PCR using primers with the NotI site (forward) and XhoI (reverse). The amplicon was inserted between the NotI and XhoI restriction sites in the *nef* gene. The amount of p24 Gag in the concentrated virus was quantified using an HIV-1 p24 antigen ELISA kit (ZeptoMetrix) according to the manufacturer's instructions.

Jurkat T and Molt4 cells were infected with rHIV-1 using a retronectin-coated plate (treated with rHIV-1 at 0.1 mg ml$^{-1}$). Live and dead cells in the infected cells were counted by the trypan blue dye exclusion method. To measure cell-associated HIV-1 DNA after 24 h, genomic DNA from the Jurkat T cells was extracted using the DNeasy Mini Kit (Qiagen) according to the manufacturer's instructions and HIV-1 *gag* DNA was measured using KOD SYBR qPCR mix and Applied Biosystems StepOnePlus Real-Time PCR System. *gag* and *ALB* primers used are listed in Supplementary Table 5. At 48 h postinfection, the infected Jurkat and Molt4 cells were treated with Foxp3 transcription factor staining buffer set (Thermo Fisher Scientific) and stained with FITC-labelled anti-HIV p24 mAb (Beckman) for 30 min at room temperature. Dead cells were distinguished using the LIVE/DEAD Fixable Near-IR Cell Stain Kit. Stained cells were analysed with FACSVerse (BD Biosciences).

## Prediction of transcription factor binding to the OCR

We performed a prediction analysis for transcription factors binding to the OCR using TFBIND (https://tfbind.hgc.jp). Single nucleotide polymorphism analysis for three RUNX binding sites in OCR was performed using reference HTLV-1 genome sequences from the infected individuals (AC, HAM and ATL)[42].

For ATAC-seq motif enrichment analysis, paired-end reads from wt39 and wt51 were used. Adaptor sequences were trimmed using Cutadapt (version 1.18) from Read 1 and Read 2 files. The reads were then cleaned in paired-end mode with PRINSEQ (version prinseq-lite-0.20.4). The resulting reads were mapped to custom genome hg19 and HTLV-1 AB513134 using BWA-MEM. BAM files were filtered using SAMtools (version 1.18) and Picard command line (http://broadinstitute.github.io/picard/) to remove reads aligned with mitochondrial chromosomes, duplicate reads, low-quality alignments and multi-mapped reads. Peak calling was performed using MACS2 following the ENCODE ATAC-seq pipeline. Paired-end BAM files were first converted to BEDPE format. Peaks were identified using the options -f BEDPE −nomodel −shift -37 −extsize 73 -g hs -B −keep-dup all, which adjust read coverage to align with nucleosome-centred regions (-145 bp). Motif enrichment analysis was performed by inputting the peak summit bed file from MACS2 and using the script findMotifsGenome.pI in HOMER motif analysis (http://homer.ucsd.edu/homer/motif/)[61]. Common transcription factors with high probability from TFbind and HOMER were used to generate the table (Supplementary Table 1).

## scATAC-seq data processing for HIV-1-infected individuals

To ascertain whether the HIV-1 provirus encompasses any OCRs in primary cells, SRA files containing the scATAC-seq data from three viraemic individuals were retrieved from the SRA database (accession numbers: SRX21228864, SRX21228865, SRX21228866). These patients were analysed in a previous study[62]. The SRA files were converted to FASTQ files using the 'fastq-dump' tool from SRAtoolkit (https://github.com/ncbi/sra-tools/wiki). A custom genome was generated by combining hg38 (GRCh38.p14) and HXB2 (accession number: K03455.1), using cellranger-arc mkref (10x Genomics). The raw FASTQ files containing the scATAC sequences were then aligned to the hg38–HXB2 custom genome using 'cellranger-atac count' (v2.1.0). Cell barcodes passing the knee call by Cell Ranger were used for downstream analysis. Pseudo-bulk ATAC-sequence analysis was performed using the

position-sorted BAM files from the 'cellranger-atac count' output, and Integrative Genomic Viewer[63] was used to visualize the OCRs within the HIV-1 provirus.

### Single-cell multiome (ATAC and GEX) for PBMCs of three patients with ATL

CD4[+] T cells were negatively isolated from PBMCs of three smouldering ATL cases, a clinical subtype of ATL with an indolent course, using the Pan T Cell Isolation kit (Miltenyi Biotec) and biotin-labelled anti-CD8 mAb (RPA-T8; Biolegend) and CD45RA mAb (HI100; Biolegend) according to the manufacturer's instructions, with or without ex vivo cultivation for 24 h. The CD4[+] T cells were processed for nuclei isolation according to Nuclei Isolation for Single Cell Multiome ATAC + Gene Expression protocol (10x Genomics; CG000365 Rev B) followed by Chromium Next GEM Single Cell Multiome ATAC + Gene Expression protocol (10x Genomics; CG000338 Rev E). Libraries were sequenced on DNBSeq following sequencing recommendations by 10x for single-cell multiome and demultiplexed before downstream analysis. Barcode processing, read trimming, read alignment, peak calling and read count of ATAC and gene expression (GEX) reads were performed from sequence output FASTQ files using Cell Ranger ARC (version 2.0.2). Custom genome GRCh38 (hg38) and AB513134 as separate chromosome were used as the human reference and HTLV-1 reference, respectively.

R (version 4.1.3) was used for secondary analysis of the data. Filtered feature barcode matrices from Cell Ranger were loaded into R using functions from Seurat (version 4.3.0.1) and Signac (version 1.9.0) to create Seurat Object. Cells underwent quality control by filtering based on RNA count, mitochondrial gene percentage and ATAC count. Cell-free mRNA contamination was removed using SoupX[64]; doublets were predicted and removed with DoubletFinder[65]. Finally, a total of 15,567 cells before ex vivo cultivation (8,486 a5, 3,682 a6, 3,399 a9) and 22,577 cells after ex vivo cultivation (8,375 a5, 10,160 a6, 4,042 a9) were retained for downstream analysis.

GEX data were normalized using the SCTransform function in Seurat[66], followed by principal component analysis and uniform manifold approximation and projection (UMAP) for dimensionality reduction. ATAC data were normalized using term frequency-inverse document frequency normalization, followed by singular value decomposition and UMAP function in Signac[67]. Both GEX and ATAC data were subjected to shared nearest neighbour clustering. Harmony[68] was applied to each modality to integrate data from cells both before and after ex vivo cultivation and correct for batch effects. Integration of both modalities, GEX and ATAC data were achieved through weighted-nearest neighbour (WNN) analysis.

Cell types were annotated by performing multimodal reference mapping[69] to a previously annotated reference dataset. In this study, we focused on CD4[+] T cells; therefore, cells annotated as CD4 CTL, CD4[+] naive T cells, CD4[+] proliferating T cells, CD4[+] central memory T cells ($T_{CM}$), CD4[+] effector memory T cells (CD4 TEM) and CD4[+] regulatory T cells ($T_{reg}$) were retained for further analysis. HTLV-1-infected cells were defined based on the expression of marker genes (HTLV-1 provirus and CADM1) or the presence of ATAC-seq reads (more than 1) that mapped to viral sequences. It is important to note that, by default, Cell Ranger ARC counts read from the sense orientation only, including both exonic and intronic reads, while ignoring antisense reads for UMI quantification. Theoretically, sc-multiome can distinguish between sense and antisense, but it makes mistakes. Therefore, a large number of anti-sense reads derived from sense are obtained. When either the definition by marker gene expression or the definition by ATAC-seq reads was met, we defined them as infected cells. A total of 6,344 cells before ex vivo cultivation (3,764 a5, 620 a6, 1,960 a9) and 9,275 cells after ex vivo cultivation (4,172 a5, 2,464 a6, 2,639 a9) were defined as infected cells. Motif activity analysis per cell was computed using chromVAR[70] by importing the motif reference database from JASPAR2020 (ref. 71).

### HAS-Flow and Tax staining for three patients with smouldering ATL

For HTLV-1-infected cell analysis using flow cytometry (HAS-Flow), CD4[+] T cells were stained with anti-CADM1 mAb (3E1; MBL), secondary antibody (Alexa Fluor647-labelled goat anti-chicken IgY H&L ab; Abcam, ab150171) and BV421-labelled anti-CD7 mAb (M-T701; BD Biosciences) according to the manufacturer's instructions. For Tax staining, isolated CD4[+] T cells were cultured in complete RPMI-1640 medium for at least 12 h. After cultivation, the cells were stained with APC-labelled anti-CD45RO mAb (UCHL1; Biolegend) and then treated using the Foxp3 transcription factor staining buffer set (Thermo Fisher Scientific). Tax expression was detected using FITC-labelled anti-Tax mAb (Lt4). Stained cells were analysed with FACSVerse (BD Biosciences). The sex, number and age of HTLV-1-infected patients are provided in Supplemental Table 6.

### Single-cell multiome (ATAC and GEX) for PBMCs of healthy donor

The single-cell multiome dataset of healthy male donor PBMCs was retrieved from 10x Genomics at https://www.10xgenomics.com/resources/datasets/10-k-human-pbm-cs-multiome-v-1-0-chromium-x-1-standard-2-0-0. The dataset was processed using Cell Ranger ARC 2.0.0 with genome GRCh38 (hg38) as the reference. R (version 4.3.1) was used to perform secondary analysis of the data. Secondary data processing and quality control followed a similar protocol to that used for the single-cell multiome analysis of PBMCs from three patients with ATL, with the exception of integration by Harmony. Finally, a total of 7,431 cells composed of different PBMC cell types (B, DC, NK, monocytes, CD4 T, CD8 T, other T and others) were retained for downstream analysis. Gene expression levels of RUNX, GATA and ETS family in each PBMC cell type were visualized as violin plots generated from the VlnPlot function in Seurat.

### WIPE assay

WIPE assay was performed as previously described[40,72]. Briefly, 1 million Jurkat T cells were infected with 500 ng of rHIV with either wt- or mut-OCR and cultured in complete RPMI-1640 medium for 30 days. rHIV-infected cells were detected by staining with the LIVE/DEAD Fixable Near-IR Cell Stain Kit (Thermo Fisher Scientific) and FITC-labelled anti-HIV p24 mAb (Beckman). The stained cells were then analysed by flow cytometry using a FACSVerse (BD Biosciences).

### Statistical analysis

Data were analysed using a chi-squared test or unpaired two-tailed Student's $t$-test, conducted with GraphPad Prism 7 software (GraphPad Software) unless otherwise specified. Statistical significance was defined as $P < 0.001$ for too-low $P$ values.

No statistical methods were used to predetermine sample sizes, but our sample sizes are similar to those reported before[7]. Data distribution was assumed to be normal, but this was not formally tested. Data collection and analysis were not performed blind to the conditions of the experiments.

### Reporting summary

Further information on research design is available in the Nature Portfolio Reporting Summary linked to this article.

## Data availability

Data supporting the findings reported in this study are provided in the Article. Raw sequence files (fastq) for ATAC-seq, ChIP-seq and genomic DNA have been deposited in the NCBI SRA under the Bioproject accession PRJNA1236037. The accession numbers for each ChIP-seq fastq file and processed bam files are provided in the Reporting Summary. Source data are provided with this paper.

## Code availability

The code used for ChIP-seq and ATAC-seq data analysis is available via GitHub at https://github.com/satoulab/Enhancer_HTLV_NatComm/tree/main/SCRIPTS/ChIP-seq. The code for analysing the integration site for wt39 and wt51 is also available via GitHub at https://github.com/satoulab/Enhancer_HTLV_NatComm/tree/main/SCRIPTS/DNA-seq_including_IS_analysis.

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

## Acknowledgements

We thank M. Matsuo for dedicated technical support and valuable discussions; N. Ninomiya, Y. Matsuoka, E. Harada and K. Ichiraha for dedicated technical support in Kumamoto University; M. Osato (National University of Singapore) for providing plasmids containing RUNX mutants; S. Okada (Kumamoto University) for providing B cell and myeloid cell lines; M. Maeda (Kyoto University) for providing the ATL cell line (MT1); H. Takeuchi (Tokyo Science University) for providing CEM and THP1 cell lines; Y. Aida (The University of Tokyo) for providing FLK-BLV; S. Suzu (Kumamoto University) for providing iPS-ML cells; C.R.M. Banghum (Imperial College London) for providing TBX-4B; J. Fujisawa (Kansai Medical University) for providing plasmids containing Tax; Y. Ariumi (Nagasaki University) for providing plasmids containing Tat; and S. Hino (Kumamoto University) for critical reading of the paper. This study was supported by grants from the Japan Society for the Promotion of Science (JSPS) KAKENHI (JP20H03724 and 25K02693 to Y.S., JP21K08494 and 25K10381 to K.S.) and Japan Agency for Medical Research and Development (AMED) (JP23fk0410052, JP23wm0325068 and JP23jm0210074 to Y.S.); the Grant for Joint Research Project of the Institute of Medical Science, the University of Tokyo, to Y.S.; JST MIRAI to Y.S.; JSPS Core-to-Core Program (JPJSCCA2020008) to Y.S.; and a grant from the Shinnihon Foundation of Advanced Medical Treatment Research to K.S. and Y.S. The funders had no role in the study design, data collection, data interpretation or the discussion regarding submission for publication.

## Author contributions

Y.S. contributed to the study conceptualization. K.S. and Y.S. designed the research. K.S., A.R., K. Monde, Takaharu Ueno, M.B.H., S.N.S.,

M.I.J. and T.I. performed the research. A.R., K.N., S.A.R., M. Takatori and W.S. performed bioinformatics analysis. K.S., A.R., K.O., M.O. and Y.S. analysed the data. K. Matsuda, M.U., Y.Y., Takamasa Ueno, K.T., Y.T., M. Tokunaga, K. Maeda and A.U. contributed materials, reagents and analytical tools. K.S., A.R. and Y.S. wrote the original draft. All authors reviewed and revised the draft and approved the final version of the paper for submission.

## Competing interests

The authors declare no competing interests.

## Additional information

**Extended data** is available for this paper at https://doi.org/10.1038/s41564-025-02006-7.

**Correspondence and requests for materials** should be addressed to Yorifumi Satou.

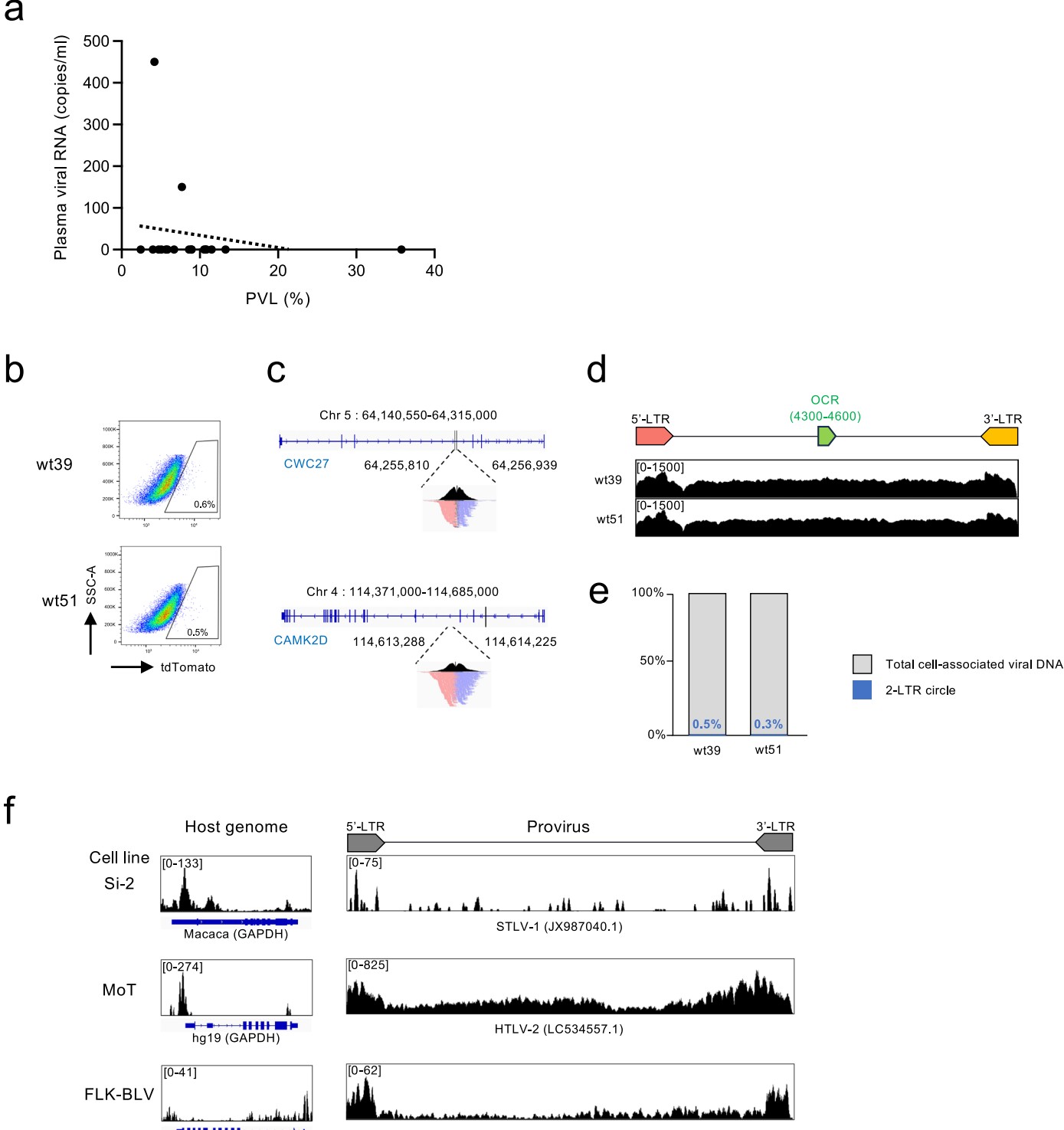

**Extended Data Fig. 1 | Correlation between plasma viral RNA and proviral DNA load (PVL) in HTLV-1 infected asymptomatic carriers, characteristics of HTLV-1-infected Jurkat T cell clones (wt39 and wt51), and ATAC-seq result of cell-lines infected with STLV-1, HTLV-2, or BLV. a**, Correlation between plasma viral RNA level and PVL in 19 HTLV-1 asymptomatic carriers. **b**, tdTomato expression driven by Tax-responsive element from wt39 and wt51 was analyzed by flowcytometry. Uninfected JET cells were used as negative control for gating. **c-d**, wt39 and wt51 cells contain single integration of intact HTLV-1 provirus, which was determined by HTLV-1 DNA capture seq in previous study[17]. **e**, Total cell-associated HTLV-1 DNA was measured by primers for tax region. 2-LTR circle was quantified as an episomal form of HTLV-1 DNA. **f**, ATAC-seq was performed for STLV-1, HTLV-2 or BLV infected cell-lines (Si-2, MoT or FLK-BLV).

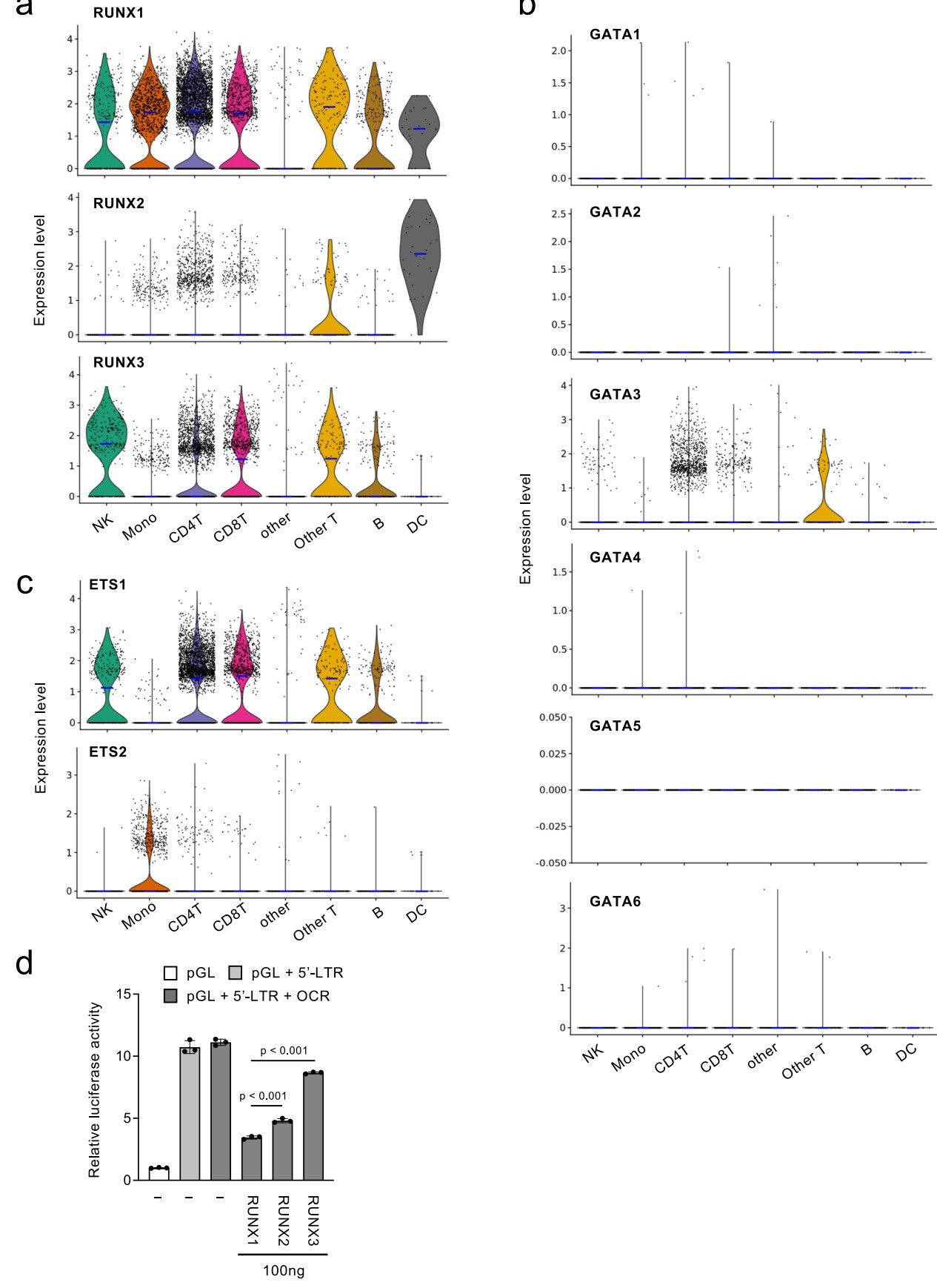

**Extended Data Fig. 2 | See next page for caption.**

**Extended Data Fig. 2 | Expression profile of RUNX, ETS, and GATA family genes in PBMC subset. a-c**, Single cell multiome (ATAC and GEX) analysis using data from healthy donor. Violin plots of RNA expression levels of RUNX family (**a**), GATA family (**b**) and ETS family (**c**) across different PBMC cell types. **d**, 5′-LTR transcriptional regulation of HTLV-1 OCR by RUNX family. 293 T cells were used for luciferase reporter assay at 48 hours after transfection. At least 2 independent experiments were performed. Bars and error bars represent the mean ± SD of results in triplicate experiments. *p* values were calculated using a two-sided, unpaired Student's *t*-test.

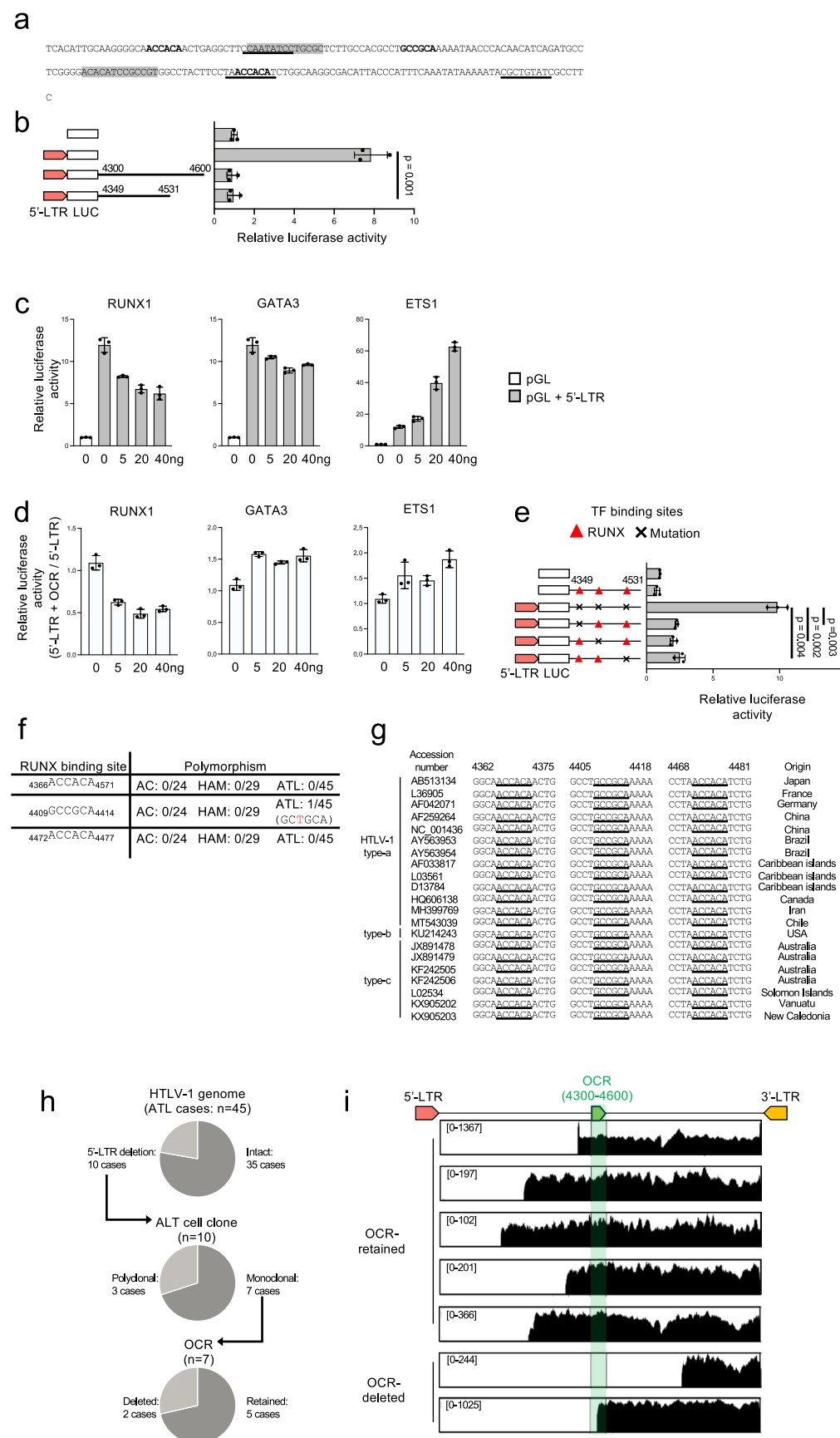

**Extended Data Fig. 3 | See next page for caption.**

**Extended Data Fig. 3 | Molecular characterization of the OCR silencer and conservation of the RUNX-binding site sequence in the OCR among various HTLV-1 strains. a**, Sequence of HTLV-1 OCR$_{4349\text{-}4531}$. RUNX binding sites were shown in bold, GATA3 binding sites are underlined and ETS1 binding sites are highlighted in gray. **b**, Luciferase assay of reporter plasmids containing essential region for the suppressive effect of 5′-LTR mediated by OCR. Jurkat T cells were used for luciferase assay. **c**, Effect of overexpression of RUNX1, GATA3 or ETS1 on promoter activity of the HTLV-1 5′-LTR alone in 293 T cells. **d**, The ratio of relative luciferase values between 5′-LTR + OCR and 5′-LTR is shown using data from Fig. 2c and Extended Data Fig. 3c. **e**, Effect of mutations in RUNX-binding sites in the OCR on the 5′-LTR promoter activity. Jurkat T cells were used for luciferase reporter assay at 48 hours after transfection. At least 2 independent experiments were performed. Bars and error bars represent the mean ± SD of results in triplicate experiments. **f**, Sequence analysis of three RUNX-binding sites in the OCR. Reference HTLV-1 sequence was compared with the sequences from AC (n = 24), HAM (n = 29) and ATL (n = 45). Sequence variant in the RUNX binding site was shown in red. **g**, Conservation of three RUNX-binding motifs within the OCR region, shown in underline, in HTLV-1 strains from different geographic locations. **h-i**, Characteristics of HTLV-1 provirus obtained in a previous study and additional analysis regarding the OCR. Seven ATL cases showed monoclonal expansion of single infected clone with defective provirus among 45 ATL cases (**h**). Presence or absence of OCR, highlighted in green, in the selected 7 ATL cases (**i**). Bars and error bars represent the mean ± SD of results in triplicate experiments. *p* values were calculated using a two-sided, unpaired Student's *t*-test.

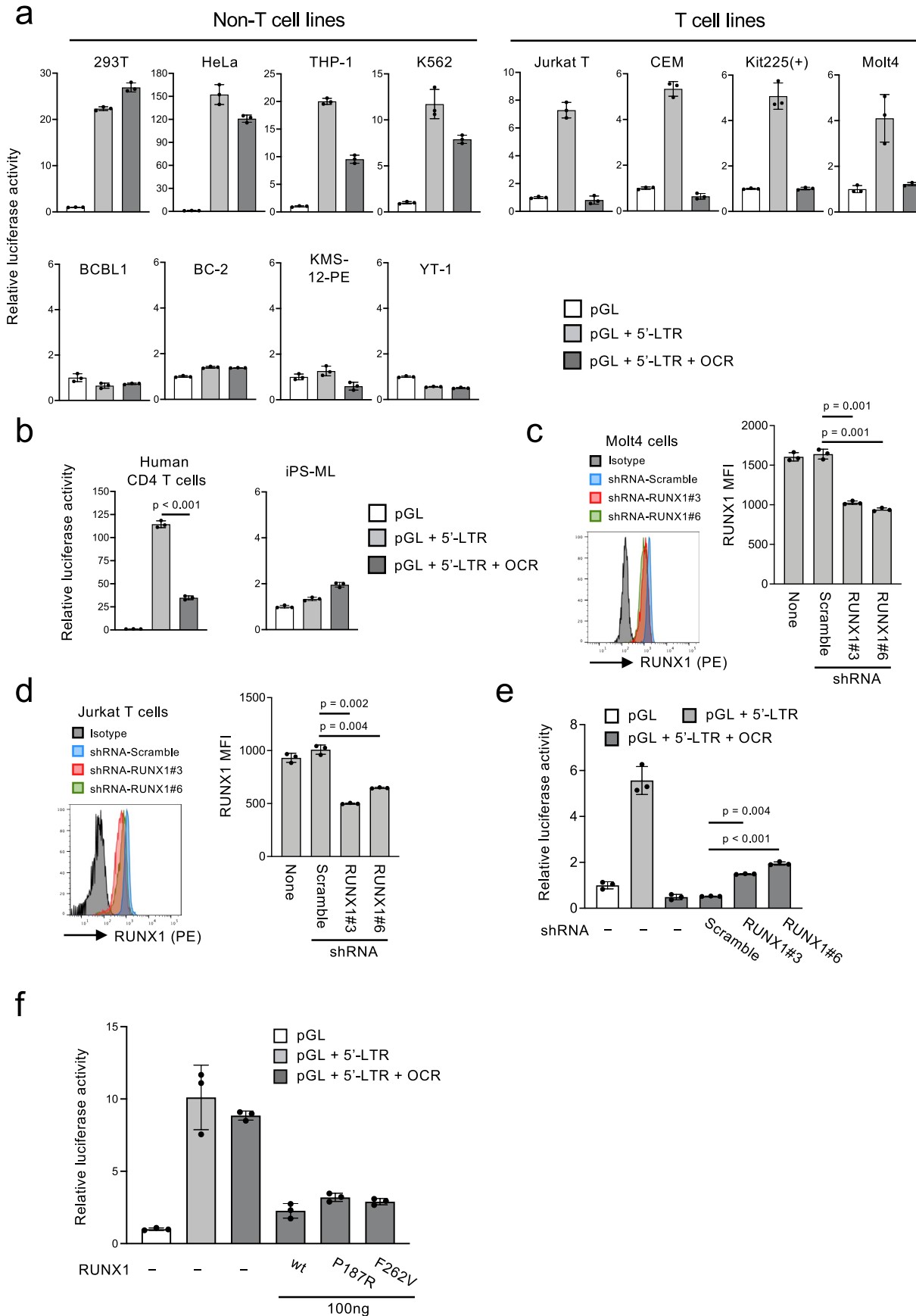

**Extended Data Fig. 4 | See next page for caption.**

**Extended Data Fig. 4 | Cell type-specificity and molecular mechanisms underlying the OCR function. a-b**, The OCR showed suppressive effect on the 5'-LTR in T cells but not in non-T cell lines. Non-T and T cell lines (**a**) human primary CD4 + T cells and iPS-ML cells (**b**) were used for the luciferase assays at 48 hours after transfection. **c-d**, Effect of the shRNA targeting RUNX1 on RUNX1 expression in Molt4 cells (**c**) and Jurkat T cells (**d**). RUNX1 protein levels were analyzed by flow cytometry analysis with anti-RUNX1 mAb. **e**, Effect of RUNX1 knock down on the OCR function. **f**, OCR mediated silencing of 5'-LTR by RUNX1 mutants (P187R and F262V) reported in ATL patients. 293 T cells were used for luciferase reporter assay. At least 2 independent experiments were performed. Bars and error bars represent the mean ± SD of results in triplicate experiments. *p* values were calculated using a two-sided, unpaired Student's *t*-test.

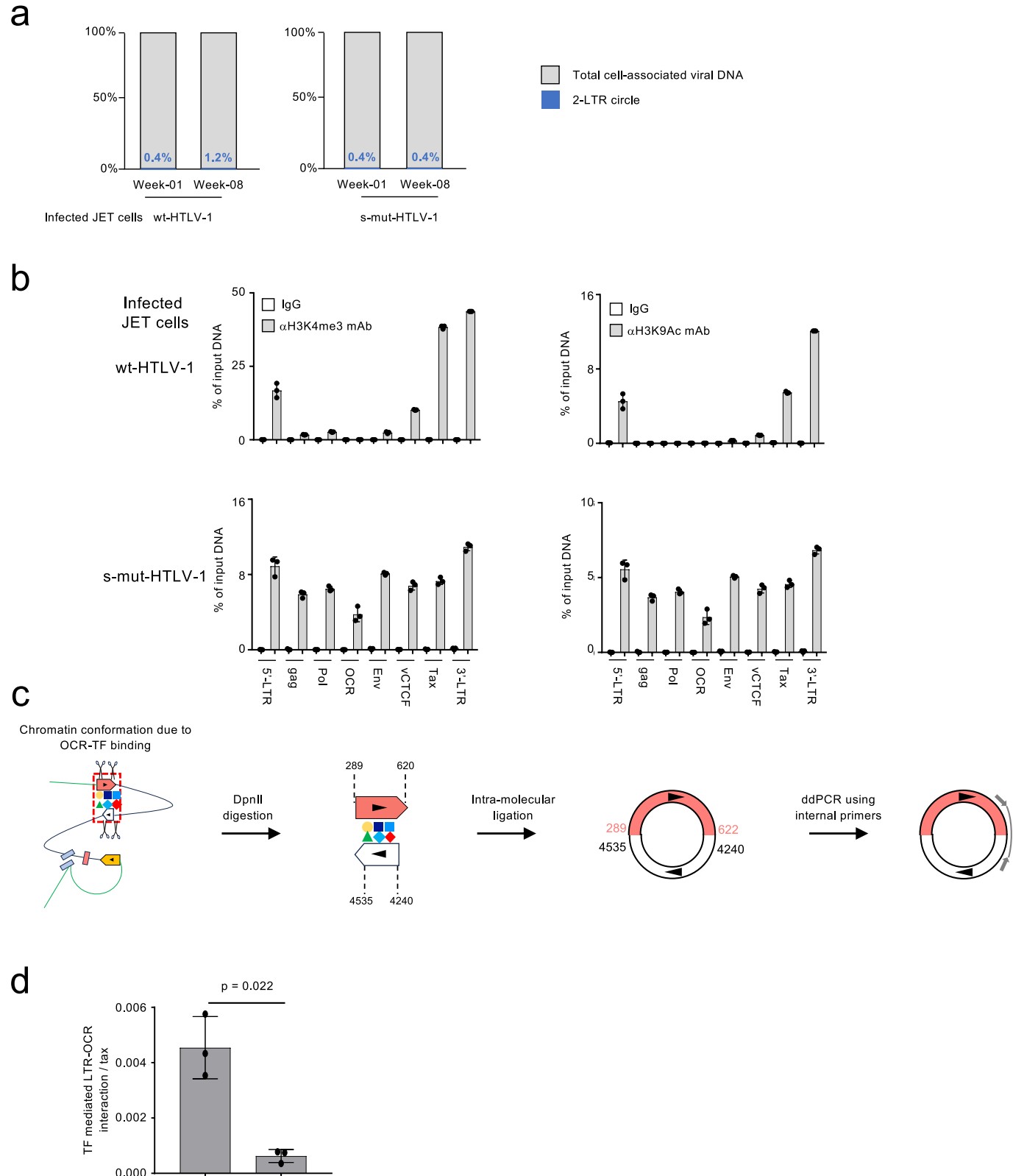

**Extended Data Fig. 5 | Epigenetic characteristics observed in proviral regions infected with wild-type (wt) or silencer-mutated (s-mut) HTLV-1.** **a**, Quantification of cell-associated HTLV-1, measured by tax as total HTLV-1 and 2-LTR circle as episomal HTLV-1 DNA in JET cells infected with wt- or s-mut-HTLV-1. **b**, ChIP-quantitative PCR (qPCR) results for H3K4me3 and H3K9Ac of HTLV-1 infected JET cells in the HTLV-1 provirus. **c**, Schematic representation of Chromatin Conformation Capture (3 C) to evaluate HTLV-1 LTR-OCR proximity. **d**, Quantification of LTR-OCR proximity in JET cells infected with wt- or s-mut-HTLV-1. A representative data from 2 independent experiments is shown. Bars and error bars represent the mean ± SD of results in triplicate experiments. *p* value was calculated using a two-sided, unpaired Student's *t*-test.

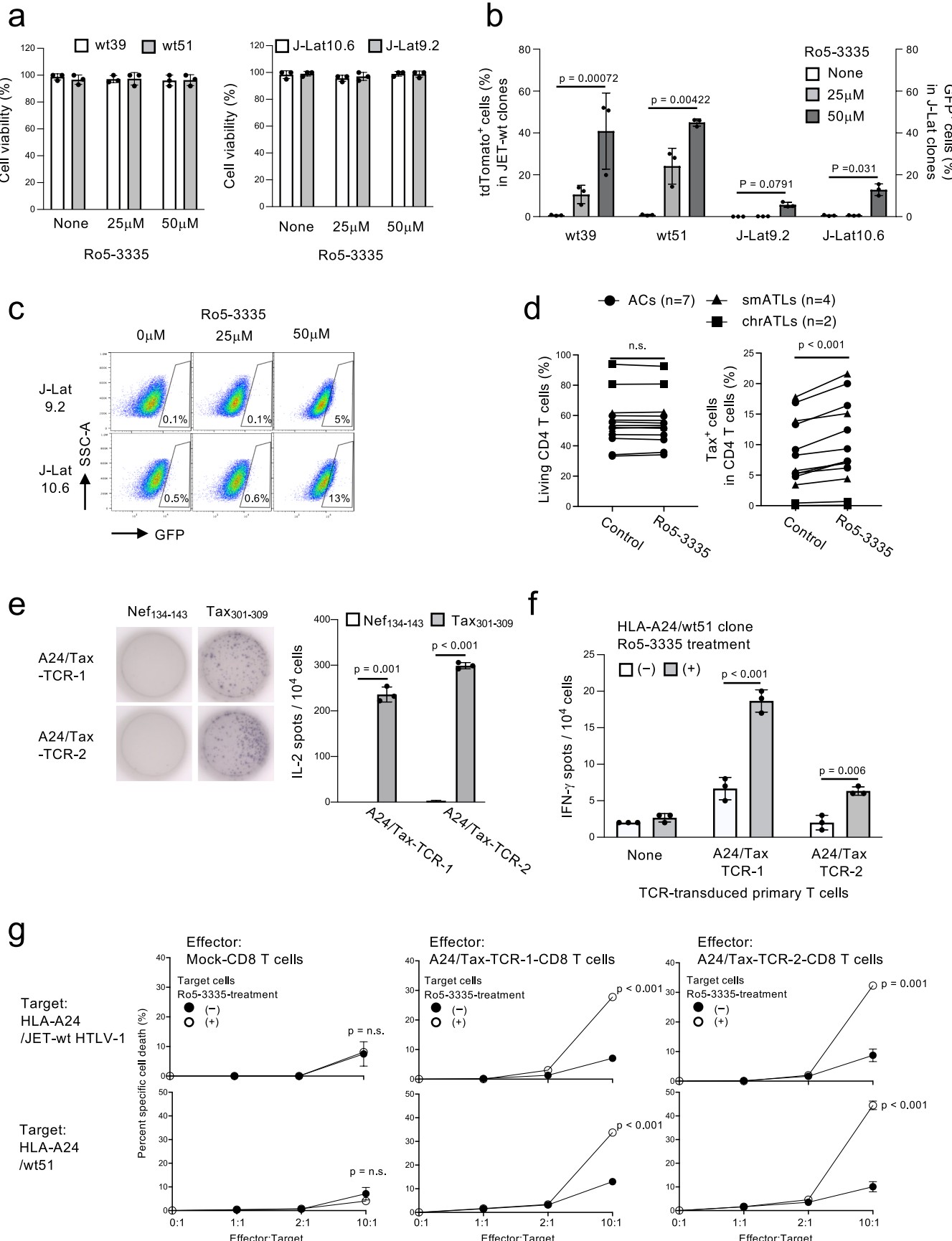

**Extended Data Fig. 6 | See next page for caption.**

**Extended Data Fig. 6 | Effect of RUNX1 inhibitor (Ro 5-3335) on proviral expression and immunogenicity against anti-Tax CTL. a**, Cell viabilities of HTLV-1-infected clones (wt39 and wt51) and HIV-1-infected clones (J-Lat10.6 and J-Lat9.2) were analyzed by the trypan blue exclusion method after treatment with RUNX1 inhibitor for 24 hours. **b**, Effect of the RUNX1 inhibitor on the HIV-1 or the HTLV-1 LTR activity. Both infected clones were treated with RUNX1 inhibitor for 24 hours and HTLV-1 Tax expression and HIV-1 5′-LTR transcription activity were analyzed using tdTomato and GFP, respectively. **c**, GFP expression levels in HIV-1 infected clones after treated with RUNX1 inhibitor for 24 hours was analyzed by flow cytometry. **d**, Effect of the RUNX1 inhibitor treatment on PBMC from HTLV-1 carriers and ATL patients. Expression of CD4 and Tax was measured by flow cytometry after 24 hours of RUNX1 inhibitor treatment. Characteristics of infected individuals analyzed were shown in Supplementary Table 4. **e-f**, Effect of

the RUNX inhibitor on the susceptibility against anti-Tax CTL. Two HLA-A24/Tax-specific TCR-transduced Jurkat T cells were cocultured with HLA-A24-transduced K562 cells pulsed with $Tax_{301\text{-}309}$ for 24 hours. IL-2 production was analyzed by ELISPOT assay (**e**). HTLV-1-infected clone (wt51) expressing HLA-A24 were treated with the RUNX inhibitor for 24 hours and then cocultured with TCR-transduced primary CD8 + T cells. IFN-g production was analyzed by ELISPOT assay after 24 hours coculture **(f). g**, HTLV-1-infected Jurkat T cells treated with Ro 5-3335 showed increased susceptibility to killing by Tax-specific CTLs. HTLV-1-infected Jurkat T cells expressing HLA-A24 were treated with the RUNX inhibitor for 24 hours and then cocultured with TCR-transduced primary CD8 + T cells. Two independent experiments were performed. Bars and error bars represent the mean ± SD of results in triplicate experiments. $P$ values were calculated using a two-sided, paired Student's $t$-test (n.s., not significant).

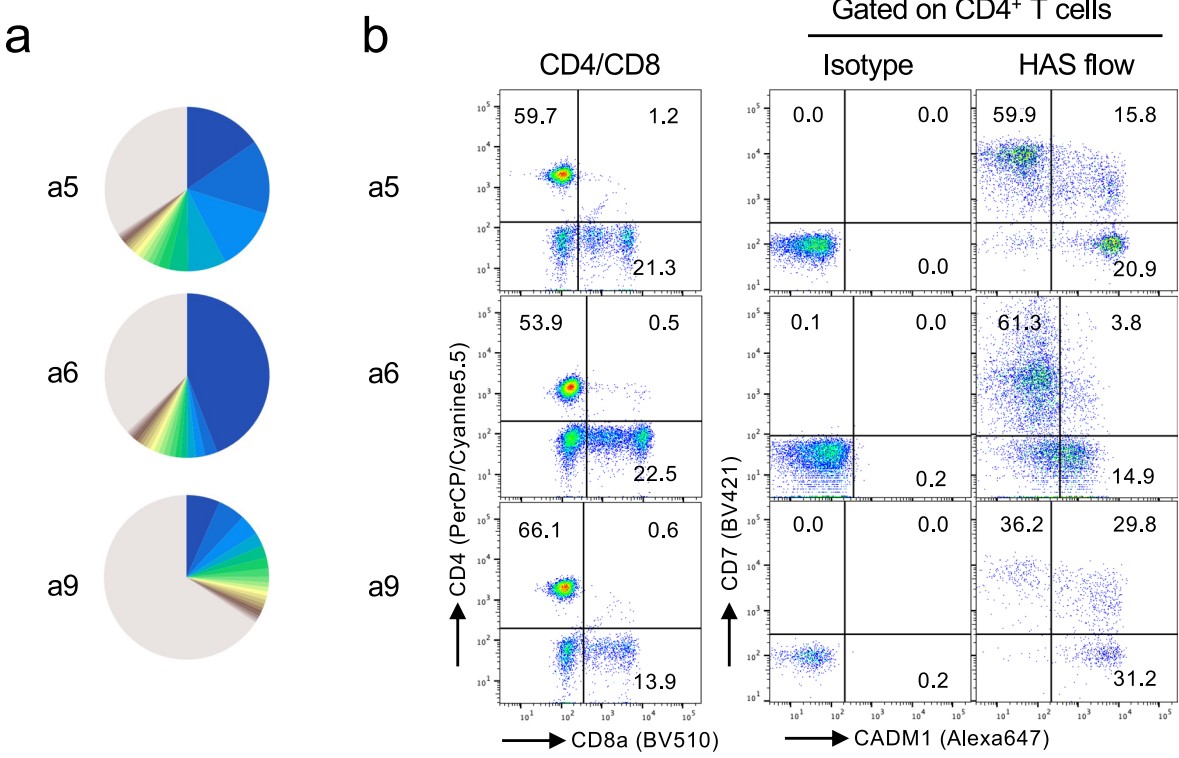

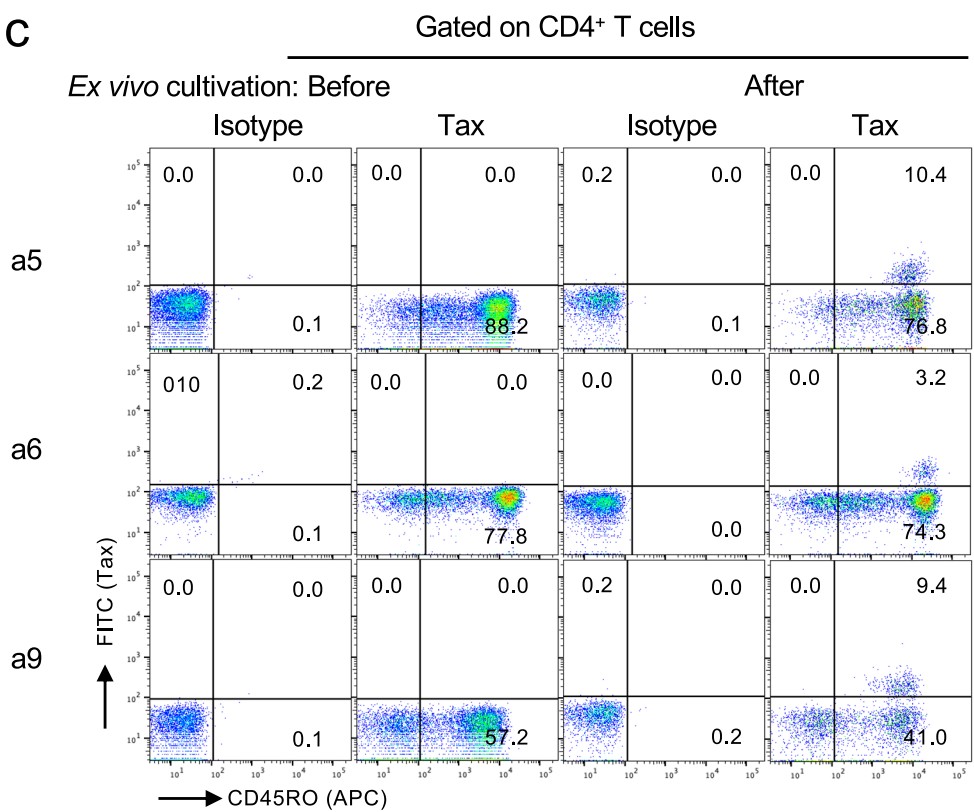

**Extended Data Fig. 7 | Characteristics of three smoldering ATL (smATL) cases. a**, Clonality of HTLV-1-infected cells obtained by Ligation-Mediated PCR (LM-PCR) sequencing. The pie chart illustrates the relative abundance of each individual clone with a distinct integration site. Each slice of the pie represents one clone with the same integration site, and the size of each slice in the pie chart is proportional to the relative frequency of each clone. **b-c**, HTLV-1-infected CD4 + T cells in PBMCs from smATL patients were analyzed by flow cytometry (HAS flow) (**b**) and Tax staining (**c**).

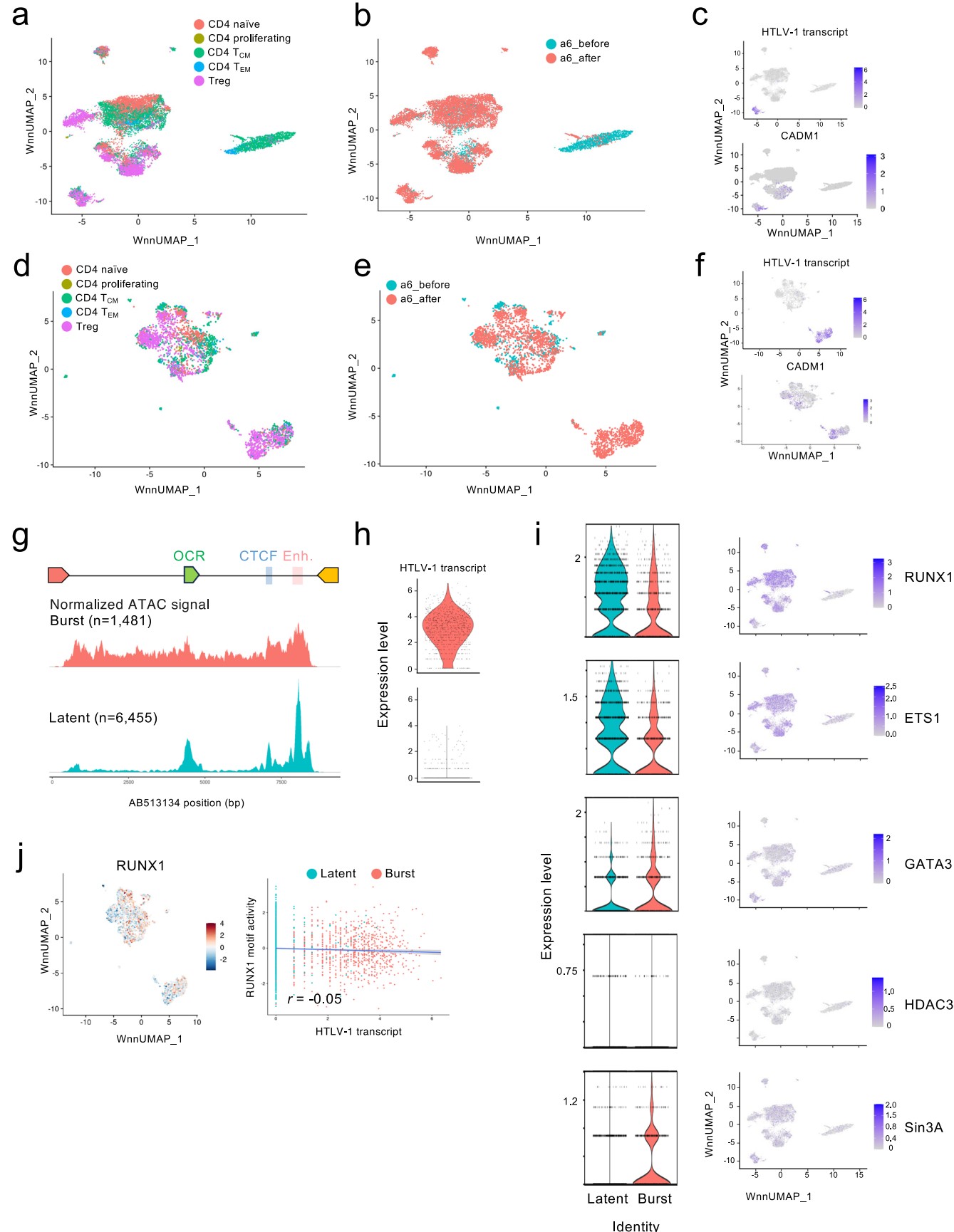

**Extended Data Fig. 8 | See next page for caption.**

**Extended Data Fig. 8 | Single Cell Multiome analysis of CD4 + T cells from a smoldering ATL case (a6). a-b**, WNN UMAP projection of all cells, both before and after cultivation identified using multimodal neighbors by weighting a combination of ATAC and RNA data. Cells are labeled for cell types (**a**) and before or after cultivation (**b**). **c**, Heatmaps of expression levels of viral RNA and CADM1, a marker of HTLV-1 infected cells. **d-e**, WNN UMAP projection of only HTLV-1 infected cells, both before and after cultivation identified using multimodal neighbors by weighting a combination of ATAC and RNA data. Cells are labeled for cell types (**d**) and before or after cultivation (**e**). **f**, Heatmaps of

expression levels of viral RNA and CADM1, a marker of HTLV-1 infected cells. **g**, Aggregated and normalized ATAC signals of burst and latent cell clusters in the proviral region. **h**, Violin plots of viral RNA expression in burst and latent cell clusters. **i**, Violin plots of infected cells and UMAP projections of all cells with a heatmap showing RNA expression level of transcription factors related with the OCR in burst and latent cell clusters. **j**, Heatmaps of RUNX1 motif activity and a correlation plot showing the relationship between the transcription level of HTLV-1 (horizontal axis) and the motif activity of RUNX1 (vertical axis). The blue line represents the regression line. Plot is labeled for burst and latent clusters.

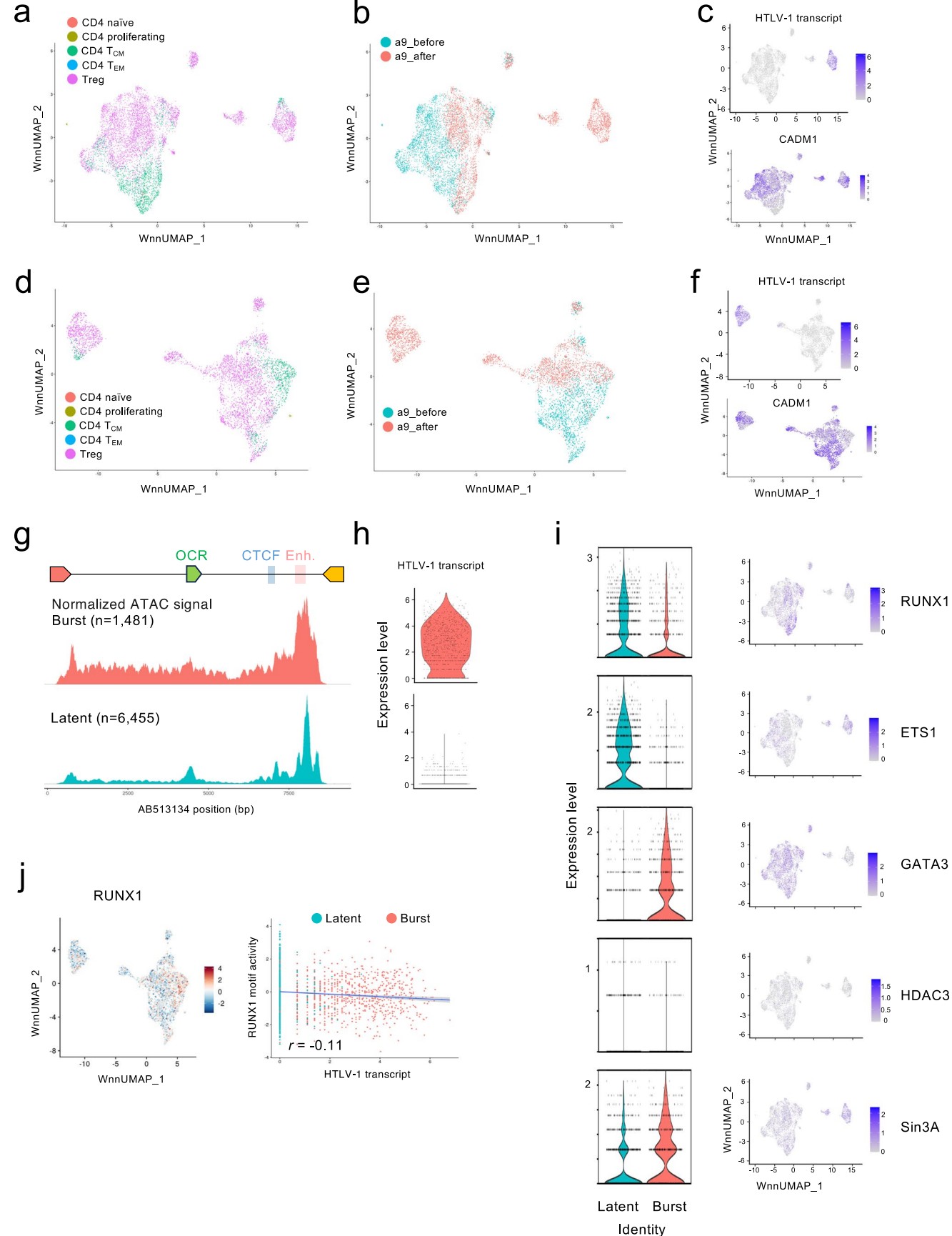

**Extended Data Fig. 9 | See next page for caption.**

**Extended Data Fig. 9 | Single Cell Multiome analysis of CD4 + T cells from a smoldering ATL case (a9). a-b**, WNN UMAP projection of all cells, both before and after cultivation identified using multimodal neighbors by weighting a combination of ATAC and RNA data. Cells are labeled for cell types (**a**) and before or after cultivation (**b**). **c**, Heatmaps of expression levels of viral RNA and CADM1, a marker of HTLV-1 infected cells. **d-e**, WNN UMAP projection of only HTLV-1 infected cells, both before and after cultivation identified using multimodal neighbors by weighting a combination of ATAC and RNA data. Cells are labeled for cell types (**d**) and before or after cultivation (**e**). **f**, Heatmaps of expression levels of viral RNA and CADM1, a marker of HTLV-1 infected cells. **g**, Aggregated and normalized ATAC signals of burst and latent cell clusters in the proviral region. **h**, Violin plots of viral RNA expression in burst and latent cell clusters. **i**, Violin plots of infected cells and UMAP projections of all cells with a heatmap showing RNA expression level of transcription factors related with the OCR in burst and latent cell clusters. **j**, Heatmaps of RUNX1 motif activity and a correlation plot showing the relationship between the transcription level of HTLV-1 (horizontal axis) and the motif activity of RUNX1 (vertical axis). The blue line represents the regression line. Plot is labeled for burst and latent clusters.

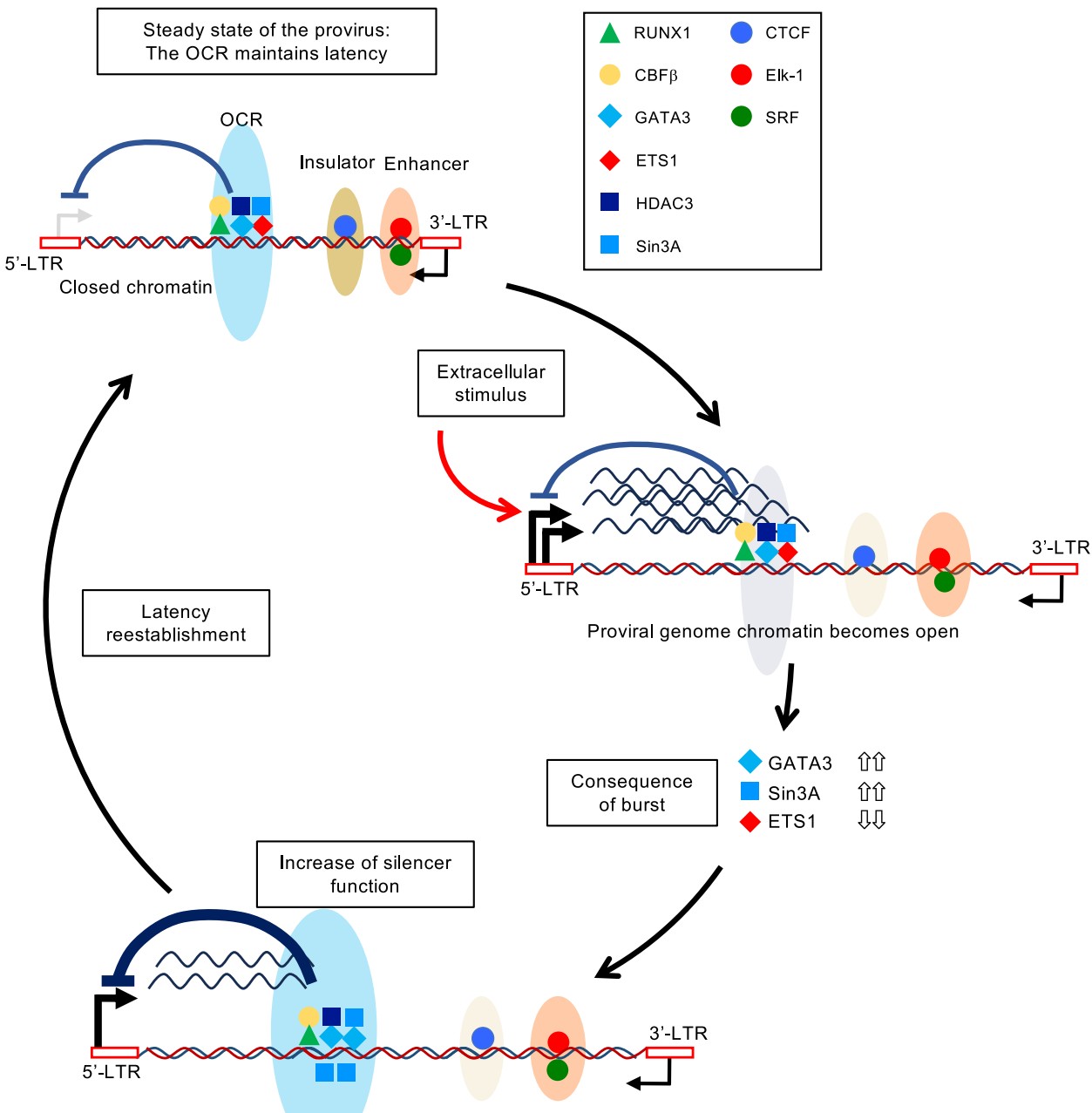

**Extended Data Fig. 10 | A scheme illustrating the molecular mechanisms underlying HTLV-1 latency.** At steady state, HTLV-1 sense expression is silenced, while antisense expression persists. Upon extracellular stimulation, promoter activity overrides silencing, triggering a transcriptional burst. Most cells are eliminated, but some revert to latency, sustaining viral persistence. Upregulation of Sin3A and GATA-3, or downregulation of EST-1, may reinforce silencing and promote latency re-establishment.

**nature** portfolio

# Reporting Summary

## Statistics

For all statistical analyses, confirm that the following items are present in the figure legend, table legend, main text, or Methods section.

| n/a | Confirmed | |
|---|---|---|
| ☐ | ☒ | The exact sample size (*n*) for each experimental group/condition, given as a discrete number and unit of measurement |
| ☐ | ☒ | A statement on whether measurements were taken from distinct samples or whether the same sample was measured repeatedly |
| ☐ | ☒ | The statistical test(s) used AND whether they are one- or two-sided<br>*Only common tests should be described solely by name; describe more complex techniques in the Methods section.* |
| ☒ | ☐ | A description of all covariates tested |
| ☒ | ☐ | A description of any assumptions or corrections, such as tests of normality and adjustment for multiple comparisons |
| ☐ | ☒ | A full description of the statistical parameters including central tendency (e.g. means) or other basic estimates (e.g. regression coefficient) AND variation (e.g. standard deviation) or associated estimates of uncertainty (e.g. confidence intervals) |
| ☐ | ☒ | For null hypothesis testing, the test statistic (e.g. *F*, *t*, *r*) with confidence intervals, effect sizes, degrees of freedom and *P* value noted<br>*Give P values as exact values whenever suitable.* |
| ☒ | ☐ | For Bayesian analysis, information on the choice of priors and Markov chain Monte Carlo settings |
| ☒ | ☐ | For hierarchical and complex designs, identification of the appropriate level for tests and full reporting of outcomes |
| ☒ | ☐ | Estimates of effect sizes (e.g. Cohen's *d*, Pearson's *r*), indicating how they were calculated |

*Our web collection on statistics for biologists contains articles on many of the points above.*

## Software and code

Policy information about availability of computer code

| Data collection | ATAC-seq:<br>ATAC libraries were sequenced on Illumina NextSeq 550 to obtain paired-end reads around 20 million reads using the following read length: read1_37 bp; read2_37 bp.<br>ChIP-seq:<br>ChIP DNA libraries after DNA capture were sequenced on Illumina MiSeq to obtain paired-end reads around 1 million reads using the following read length: read1_75 bp; read2_75 bp.<br>Sequencing for Cut and Run:<br>Cut and Run DNA libraries with DNA capture were sequenced on Illumina MiSeq to obtain paired-end reads around 1 million reads using the following read length: read1_75 bp; read2_75 bp. |
|---|---|
| Data analysis | ☐ TFBIND – The prediction tool of transcription factors binding site (https://tfbind.hgc.jp/)<br>☐ GraphPad Prism 7 – The validation of Statistical significance<br>☐ FlowJo (version 9.9.6) – FACS analysis<br>☐ FastQC (version 0.10.0) – Checking quality of fastq file (https://www.bioinformatics.babraham.ac.uk/projects/fastqc/)<br>☐ cutadapt (version 1.18) – Removing the adapter sequences (https://cutadapt.readthedocs.io/en/stable/index.html)<br>☐ PRINSEQ (version 0.20.4) – Cleaning the sequence data (http://prinseq.sourceforge.net/)<br>☐ BWA (version 0.7.12) – Alignment for DNA-seq, MNase-seq, ChIP-seq (https://sourceforge.net/projects/bio-bwa)<br>☐ SAMTools (version 1.11) – Operating sam and bam files (http://www.htslib.org/download/)<br>☐ picard (version 2.0.1) – PCR replicate removal (https://broadinstitute.github.io/picard/)<br>☐ IGV (version 2.8.0) – Used to visualize mRNA-seq, MNase-seq and ChIP-seq data (https://software.broadinstitute.org/software/igv/download) |

Codes used for ChIP-seq and ATAC-seq data analysis have been deposited to Github (https://github.com/satoulab/Enhancer_HTLV_NatComm/tree/main/SCRIPTS/ChIP-seq). Codes for analyzing integration site for the wt39 and wt 51 are also available in Github (https://github.com/satoulab/Enhancer_HTLV_NatComm/tree/main/SCRIPTS/DNA-seq_including_IS_analysis).

For manuscripts utilizing custom algorithms or software that are central to the research but not yet described in published literature, software must be made available to editors and reviewers. We strongly encourage code deposition in a community repository (e.g. GitHub). See the Nature Portfolio guidelines for submitting code & software for further information.

## Data

Policy information about availability of data

All manuscripts must include a data availability statement. This statement should provide the following information, where applicable:
- Accession codes, unique identifiers, or web links for publicly available datasets
- A description of any restrictions on data availability
- For clinical datasets or third party data, please ensure that the statement adheres to our policy

Data supporting the findings reported in this study are available from the corresponding author upon request. Raw sequence files (fastq) for ATAC-seq, ChIP-seq and genomic DNA have been deposited to SRA under the bioproject: PRJNA1236037.

## Research involving human participants, their data, or biological material

Policy information about studies with human participants or human data. See also policy information about sex, gender (identity/presentation), and sexual orientation and race, ethnicity and racism.

| | |
|---|---|
| Reporting on sex and gender | This information is provided in Supplemental Table 1, Supplemental Table 2 and Supplemental Table 6. |
| Reporting on race, ethnicity, or other socially relevant groupings | This manuscript did not include the research for race, ethnicity or other socially constructed categories. |
| Population characteristics | ACs and smoldering and chronic ATL patients were male and female, ages 42-78(Median:67). Each sample has high PVL 2.4-35.8(Median:14.5). |
| Recruitment | ATL samples were obtained from patients with high PVL. Some patients supplied the samples of different time-points. We recruited all participants as volunteers without any biases and written informed consent was obtained from each participant. |
| Ethics oversight | This study was approved by the Kumamoto University Institutional Review Board (approval number 248 and 263) and carried out in accordance with the guidelines proposed in the Declaration of Helsinki. Informed written consent was obtained from all subjects in this study. |

Note that full information on the approval of the study protocol must also be provided in the manuscript.

# Field-specific reporting

Please select the one below that is the best fit for your research. If you are not sure, read the appropriate sections before making your selection.

☒ Life sciences  ☐ Behavioural & social sciences  ☐ Ecological, evolutionary & environmental sciences

For a reference copy of the document with all sections, see nature.com/documents/nr-reporting-summary-flat.pdf

# Life sciences study design

All studies must disclose on these points even when the disclosure is negative.

| | |
|---|---|
| Sample size | Sample sizes were chosen to provide sufficient confidence to validate the tendency of chromatin-openess, the viral and host transcription level, and TF binding in HTLV-1 infected cell lines and patient samples. We did not do any calculation for sample size. We analyzed as many samples as possible in terms of clinical sample and research budget availability. |
| Data exclusions | No data were excluded from the manuscript. |
| Replication | All results shown in the manuscript were validated by duplicates or triplicates |
| Randomization | Randomization is not required for this study. |
| Blinding | Blinding is not required for this study. |

# Reporting for specific materials, systems and methods

We require information from authors about some types of materials, experimental systems and methods used in many studies. Here, indicate whether each material, system or method listed is relevant to your study. If you are not sure if a list item applies to your research, read the appropriate section before selecting a response.

## Materials & experimental systems

| n/a | Involved in the study |
|---|---|
| ☐ | ☒ Antibodies |
| ☐ | ☒ Eukaryotic cell lines |
| ☒ | ☐ Palaeontology and archaeology |
| ☒ | ☐ Animals and other organisms |
| ☒ | ☐ Clinical data |
| ☒ | ☐ Dual use research of concern |
| ☒ | ☐ Plants |

## Methods

| n/a | Involved in the study |
|---|---|
| ☐ | ☒ ChIP-seq |
| ☐ | ☒ Flow cytometry |
| ☒ | ☐ MRI-based neuroimaging |

## Antibodies

| | |
|---|---|
| Antibodies used | anti-RUNX1 (abcam ab23980), anti-CBFb (abcam ab195411), anti-GATA3 (CST 5852P), anti-HDAC3 (Invitrogen PA5-85378), anti-ETS1 (CST 14069S) and anti Sin3A (CST 7691); 1:250 dilution for conventional ChIP. For Cut and Run, the antibodies are used in 1:50 dilution.<br>FITC-labeled anti-Tax mAb (Lt4: kindly gifted by Yuetsu Tanaka (University of the Ryukyus)<br>mouse anti-RUNX1 (A-2; Santa Cruz Biotechnology); 1:2000 dilution<br>Rabbit anti-beta actin (13E5; Cell Signaling Technology); 1:5000 dilution<br>horseradish peroxidase (HRP)-conjugated secondary antibodies donkey anti-rabbit IgG-HRP (Jackson ImmunoResearch) ;1:10000 dilution<br>horseradish peroxidase (HRP)-conjugated secondary antibodies donkey donkey anti-mouse IgG-HRP (Jackson ImmunoResearch);1:10000 dilution<br>PE-labeled anti-RUNX1 antibody (RXDMC, Invitrogen) ; 1:20 dilution<br>PE-labeled anti-NGFR mAb (ME20.4, BioLegend) ; 1:30 dilution<br>FITC-labeled anti-HIV p24 mAb (Beckman);1:500 dilution<br>biotin-labeled anti CD8 mAb (RPA-T8; Biolegend);1:30 dilution<br>biotin-labeled CD45RA mAb (HI100; Biolegend) ;1:30 dilution<br>PerCP/Cy5.5-labeled anti-CD4 mAb (OKT4, BioLegend);1:30 dilution<br>anti-H3K4me3; 1:250 dilution<br>anti-H3K9Ac; 1:250 dilution<br>PE-labeled anti-EGFR mAb (AY13, BioLegend); 1:30 dilution<br>BV510-labeled anti CD8 mAb (RPA-T8; Biolegend) 1:30 dilution<br>APC-labeled anti EGFR mAb (AY13; Biolegend) 1:30 dilution<br>anti CADM1 mAb (3E1; MBL);1:1000 dilution<br>Alexa Fluor647-labeled Goat Anti-Chicken IgY H&L ab(Abcam);1:2000 dilution<br>BV421-labeled anti CD7 mAb (M-T701; BD Biosciences) 1:30 dilution<br>APC-labeled anti CD45RO mAb (UCHL1; Biolegend) 1:30 dilution |
| Validation | Further validation report could be found on the the supplier website.<br>anti-RUNX1 (abcam ab23980): https://www.abcam.co.jp/products/primary-antibodies/runx1--aml1-antibody-ab23980.html<br>anti-CBFb (abcam ab195411):https://www.abcam.co.jp/products/primary-antibodies/cbfb-antibody-chip-grade-ab195411.html<br>anti-GATA3 (CST 5852P):https://www.cellsignal.jp/products/primary-antibodies/gata-3-d13c9-xp-rabbit-mab/5852<br>anti-HDAC3 (Invitrogen PA5-85378):https://www.thermofisher.com/antibody/product/HDAC3-Antibody-Polyclonal/PA5-85378<br>anti-ETS1 (CST 14069S) :https://www.cellsignal.jp/products/primary-antibodies/ets-1-d8o8a-rabbit-mab/14069<br>Sin3A (CST 7691):https://www.cellsignal.jp/products/primary-antibodies/sin3a-d1b7-rabbit-mab/7691<br>mouse anti-RUNX1 (A-2; Santa Cruz Biotechnology):https://www.scbt.com/p/runx1-antibody-a-2<br>Rabbit anti-beta actin (13E5; Cell Signaling Technology):https://www.cellsignal.jp/products/primary-antibodies/b-actin-13e5-rabbit-mab/4970<br>horseradish peroxidase (HRP)-conjugated secondary antibodies donkey anti-rabbit IgG-HRP (Jackson ImmunoResearch):https://www.jacksonimmuno.com/catalog/products/711-035-152<br>horseradish peroxidase (HRP)-conjugated secondary antibodies donkey donkey anti-mouse IgG-HRP (Jackson ImmunoResearch):https://www.jacksonimmuno.com/catalog/products/715-035-150<br>PE-labeled anti-RUNX1 antibody (RXDMC, Invitrogen) :https://www.thermofisher.com/antibody/product/RUNX1-Antibody-clone-RXDMC-Monoclonal/12-9816-80<br>PE-labeled anti-NGFR mAb (ME20.4, BioLegend) :https://www.biolegend.com/ja-jp/products/pe-anti-human-cd271-ngfr-antibody-6428?GroupID=GROUP28<br>FITC-labeled anti-HIV p24 mAb (Beckman):https://www.beckman.com/reagents/coulter-flow-cytometry/antibodies-and-kits/single-color-antibodies/hiv-1-core-antigen/6604665<br>biotin-labeled anti CD8 mAb (RPA-T8; Biolegend):https://www.biolegend.com/ja-jp/clone-search/biotin-anti-human-cd8a-antibody-833?GroupID=BLG5903<br>biotin-labeled CD45RA mAb (HI100; Biolegend) :https://www.biolegend.com/ja-jp/products/biotin-anti-human-cd45ra-antibody-685<br>PerCP/Cy5.5-labeled anti-CD4 mAb (OKT4, BioLegend):https://www.biolegend.com/ja-jp/products/percp-cyanine5-5-anti-human-cd4-antibody-5011<br>anti-H3K4me3; https://www.cellsignal.jp/products/primary-antibodies/tri-methyl-histone-h3-lys4-c42d8-rabbit-mab/9751<br>anti-H3K9Ac; https://www.merckmillipore.com/JP/ja/product/Anti-acetyl-Histone-H3-Lys9-Antibody,MM_NF-06-942<br>PE-labeled anti-EGFR mAb (AY13, BioLegend); https://www.biolegend.com/ja-jp/products/pe-anti-human-egfr-antibody-7432<br>BV510-labeled anti CD8 mAb (RPA-T8; Biolegend);https://www.biolegend.com/ja-jp/products/brilliant-violet-510-anti-human-cd8a- |

antibody-8000
APC-labeled anti EGFR mAb (AY13; Biolegend) https://www.biolegend.com/ja-jp/products/apc-anti-human-egfr-antibody-7714
anti CADM1 mAb (3E1; MBL) https://ruo.mbl.co.jp/bio/dtl/A/index.html?pcd=CM004-3
Alexa Fluor647-labeled Goat Anti-Chicken IgY H&L ab(Abcam) https://www.abcam.co.jp/products/secondary-antibodies/goat-chicken-igy-hl-alexa-fluor-647-ab150171.html
BV421-labeled anti CD7 mAb (M-T701; BD Biosciences); https://www.bdbiosciences.com/en-dk/products/reagents/flow-cytometry-reagents/research-reagents/single-color-antibodies-ruo/bv421-mouse-anti-human-cd7.562635
APC-labeled anti CD45RO mAb (UCHL1; Biolegend);https://www.biolegend.com/ja-jp/products/apc-anti-human-cd45ro-antibody-856

# Eukaryotic cell lines

Policy information about cell lines and Sex and Gender in Research

| Cell line source(s) | Jurkat cells were obtained from ATCC (TIB-152). |
|---|---|
| | 293T cells were obtained from ATCC (CRL-3216). |
| | Molt4 cells were obtained from ATCC (CRL-1582). |
| | Kit225(+) cells were obtained from ATCC (CRL-1990). |
| | 293T cells were obtained from ATCC (CRL-3216). |
| | PG13 cells were obtained from ATCC (CRL-10686). |
| | HeLa cells were obtained from ATCC (CCL-2). |
| | K562 cells were obtained from ATCC (CCL-243). |
| | MoT cells were obtained from ATCC (CRL-8066) |
| | Si-2 cells were obtained from Japanese Collection of Research Bioresources Cell Bank (JCRB1321) |
| | FLK-BLV was kindly provided by Yoko Aida (The University of Tokyo) |
| | Plate-GP cells were obtained from Takara Bio Inc. |
| | BCBL1, BC-2, KMS-12-PE and YT-1 cell lines were kindly gifted by Seiji Okada (Kumamoto university). |
| | iPS-ML cells were kindly gifted by Shinya Suzu (Kumamoto university). |
| | CEM and THP1 cell lines were obtained from Hiroaki Takeuchi (Tokyo Medical and Dental University). |
| | MT1 cells were obtained from Dr. Michiyuki Maeda (Maeda M. et al., J Exp Med. 1985). |
| | TBX-4B cells were obtained from Prof. Charles R.M. Bangham (Cook L.B. et al., Blood. 2014). |
| | JET cells were obtained from Prof. Jun-ichi Fujisawa (Furuta R. et al., PLos Pathog. 2017). |
| | wt39 and wt51 cells were established from JET cells infected with HTLV-1 molecular clone using limiting dilution (Matsuo M. et al., Nature communications. 2021). |
| | HTLV-1 molecular clone, J-Lat cells (9.2 and 10.6) were obtained from the National Institutes of Health AIDs reagent Program. |
| | TCR deficient Jurkat cells were kindly gifted by Takamasa Ueno (Kumamoto university) |
| Authentication | None of the cell lines used were authenticated. |
| Mycoplasma contamination | All cell lines tested negative for mycoplasma contamination. |
| Commonly misidentified lines (See ICLAC register) | None of the cell lines used in this study are listed in this database. |

# Plants

| Seed stocks | n/a |
|---|---|
| Novel plant genotypes | n/a |
| Authentication | n/a |

# ChIP-seq

## Data deposition

☒ Confirm that both raw and final processed data have been deposited in a public database such as GEO.

☐ Confirm that you have deposited or provided access to graph files (e.g. BED files) for the called peaks.

| Data access links *May remain private before publication.* | Data supporting the findings reported in this study are available from the corresponding author upon request. Raw sequence files (fastq) for ChIP-seq(SUB15170571), ATAC-seq and genomic DNA sequences(SUB15174788) have been deposited to SRA under the bioproject: PRJNA1236037. |
|---|---|
| Files in database submission | Raw fastq files and processed bam files from ChIP-seq data are following: [Fastq:SRR32706051] [bam:SRR32764928] 21_wt39_RUNX1_ChIP_replicate_01 [Fastq:SRR32706051] [bam:SRR32764927] 23_wt51_RUNX1_ChIP_replicate_01 |

[Fastq:SRR32706017] [bam:SRR32764916]  25_wt39_CBFb_ChIP_replicate_01
[Fastq:SRR32706013] [bam:SRR32764905]  27_wt51_CBFb_ChIP_replicate_01
[Fastq:SRR32706011] [bam:SRR32764904]  29_wt39_GATA3_ChIP_replicate_01
[Fastq:SRR32706049] [bam:SRR32764903]  31_wt51_GATA3_ChIP_replicate_01
[Fastq:SRR32706047] [bam:SRR32764902]  33_wt39_ETS1_ChIP_replicate_01
[Fastq:SRR32706045] [bam:SRR32764901]  35_wt51_ETS1_ChIP_replicate_01
[Fastq:SRR32706043] [bam:SRR32764900]  37_wt39_HDAC3_ChIP_replicate_01
[Fastq:SRR32706041] [bam:SRR32764899]  39_wt51_HDAC3_ChIP_replicate_01
[Fastq:SRR32706038] [bam:SRR32764926]  41_wt39_Sin3A_ChIP_replicate_01
[Fastq:SRR32706036] [bam:SRR32764925]  42_wt51_Sin3A_ChIP_replicate_01
[Fastq:SRR32706034] [bam:SRR32764924]  43_AI-5_RUNX1_ChIP_replicate_01
[Fastq:SRR32706033] [bam:SRR32764923]  44_AI-9_RUNX1_ChIP_replicate_01
[Fastq:SRR32706032] [bam:SRR32764922]  45_AI-5_CBFb_ChIP_replicate_01
[Fastq:SRR32706031] [bam:SRR32764921]  46_AI-9_CBFb_ChIP_replicate_01
[Fastq:SRR32706030] [bam:SRR32764920]  47_AI-5_GATA3_ChIP_replicate_01
[Fastq:SRR32706029] [bam:SRR32764919]  48_AI-9_GATA3_ChIP_replicate_01
[Fastq:SRR32706027] [bam:SRR32764918]  49_AI-5_ETS1_ChIP_replicate_01
[Fastq:SRR32706026] [bam:SRR32764917]  50_AI-9_ETS1_ChIP_replicate_01
[Fastq:SRR32706025] [bam:SRR32764915]  51_AI-5_HDAC3_ChIP_replicate_01
[Fastq:SRR32706024] [bam:SRR32764914]  52_AI-9_HDAC3_ChIP_replicate_01
[Fastq:SRR32706023] [bam:SRR32764913]  53_AI-5_Sin3A_ChIP_replicate_01
[Fastq:SRR32706022] [bam:SRR32764912]  54_AI-9_Sin3A_ChIP_replicate_01
[Fastq:SRR32706021] [bam:SRR32764911]  55_JEX_wt_bulk_RUNX1_ChIP_replicate_01
[Fastq:SRR32706020] [bam:SRR32764910]  56_JEX_s-mut_bulk_RUNX1_ChIP_replicate_01
[Fastq:SRR32706019] [bam:SRR32764909]  57_JEX_wt_bulk_CBFb_ChIP_replicate_01
[Fastq:SRR32706018] [bam:SRR32764908]  58_JEX_s-mut_bulk_CBFb_ChIP_replicate_01
[Fastq:SRR32706016] [bam:SRR32764907]  59_JEX_wt_bulk_HDAC3_ChIP_replicate_01
[Fastq:SRR32706015] [bam:SRR32764906]  60_JEX_s-mut_bulk_HDAC3_ChIP_replicate_01
[Fastq:SRR32706050]      22_wt39_RUNX1_ChIP_replicate_02
[Fastq:SRR32706028]      24_wt51_RUNX1_ChIP_replicate_02
[Fastq:SRR32706014]      26_wt39_CBFb_ChIP_replicate_02
[Fastq:SRR32706012]      28_wt51_CBFb_ChIP_replicate_02
[Fastq:SRR32706010]      30_wt39_GATA3_ChIP_replicate_02
[Fastq:SRR32706048]      32_wt51_GATA3_ChIP_replicate_02
[Fastq:SRR32706046]      34_wt39_ETS1_ChIP_replicate_02
[Fastq:SRR32706044]      36_wt51_ETS1_ChIP_replicate_02
[Fastq:SRR32706042]      38_wt39_HDAC3_ChIP_replicate_02
[Fastq:SRR32706040]      40_wt51_HDAC3_ChIP_replicate_02
[Fastq:SRR32706037]      61_wt39_Sin3A_ChIP_replicate_02
[Fastq:SRR32706035]      62_wt51_Sin3A_ChIP_replicate_02

**Genome browser session**
(e.g. UCSC)

*Provide a link to an anonymized genome browser session for "Initial submission" and "Revised version" documents only, to enable peer review.  Write "no longer applicable" for "Final submission" documents.*

## Methodology

**Replicates**

All results using cell line sample shown in the manuscript were validated by duplicate.
ChIP-seq data using patients samples were performed by single assay because of the limitation with few cell number.

**Sequencing depth**

In samples with DNA-capture, libraries were sequenced by Illumina MiSeq to obtain paired-end reads around 1 million reads using the following read length: read1_75 bp; read2_75 bp.

**Antibodies**

anti-RUNX1 (abcam ab23980), anti-CBFb (abcam ab195411), anti-GATA3 (CST 5852P), anti-HDAC3 (Invitrogen PA5-85378), anti-ETS1 (CST 14069S) and anti Sin3A (CST 7691); 1:250 dilution for conventional ChIP. For Cut and Run, the antibodies are used in 1:50 dilution.

**Peak calling parameters**

No peak calling was performed.

**Data quality**

The quality of raw data fastq files were determined with FastQC and the reads having phred score >20 were filtered for mapping.

**Software**

☐ FastQC (version 0.10.0) – Checking quality of fastq file (https://www.bioinformatics.babraham.ac.uk/projects/fastqc/)
☐ cutadapt (version 1.18) – Removing the adapter sequences (https://cutadapt.readthedocs.io/en/stable/index.html)
☐ PRINSEQ (version 0.20.4) – Cleaning the sequence data (http://prinseq.sourceforge.net/)
☐ BWA (version 0.7.12)  – Alignment for DNA-seq, MNase-seq, ChIP-seq (https://sourceforge.net/projects/bio-bwa)
☐ SAMTools (version 1.11) – Operating sam and bam files (http://www.htslib.org/download/)
☐ picard (version 2.0.1)  – PCR replicate removal (https://broadinstitute.github.io/picard/)
☐ IGV (version 2.8.0) – Used to visualize mRNA-seq, MNase-seq and ChIP-seq data (https://software.broadinstitute.org/software/igv/download)

# Flow Cytometry

## Plots

Confirm that:

☒ The axis labels state the marker and fluorochrome used (e.g. CD4-FITC).

☒ The axis scales are clearly visible. Include numbers along axes only for bottom left plot of group (a 'group' is an analysis of identical markers).

☒ All plots are contour plots with outliers or pseudocolor plots.

☒ A numerical value for number of cells or percentage (with statistics) is provided.

## Methodology

| | |
|---|---|
| Sample preparation | shRNA-RUNX1-transduced Jurkat cells were stained by PE-labeled anti-RUNX1 antibody (RXDMC, Invitrogen). JET cell clones (wt39 and wt51) infected with HTLV-1 molecular clone, J-Lat cells (9.2 and 10.6) and PBMCs from ACs and ATL patients were treated with Ro5-3335 for 24 hours. Enhanced 5'LTR reactivation in the JET cell clones and J-Lat cells was measured by expression of tdTomato and GFP, respectively. Ro5-3335-treated PBMCs were stained with PerCP/Cy5.5-labeled anti-CD4 mAb (OKT4, BioLegend) and FITC-labeled anti-Tax mAb. RUNX1-transduced TBX4B and MT-1 cells were stained with FITC-labeled anti-Tax mAb (Lt4) for 30 min at room temperature.  rHIV-infected Jurkat and Molt4 cells were detected upon staining with LIVE/DEAD Fixable Near-IR Cell Stain Kit (Thermo Fisher Scientific) and FITC-labeled anti-HIV p24 mAb (Beckman). PBMCs from three ATL patients were stained with anti CADM1 mAb and secondary antibody (Alexa Fluor647-labeled Goat Anti-Chicken IgY H&L ab) and BV421-labeled anti CD7 mAb (M-T701; BD Biosciences), APC-labeled anti CD45RO mAb (UCHL1; Biolegend), and FITC-labeled anti-Tax mAb (Lt4). |
| Instrument | Processed cells were detected by flow cytometry using BD FACSVerse  and Sony Biotechnology SH800S Cell Sorter |
| Software | Flow cytometry data was analyzed using FlowJo (version 9.9.6). |
| Cell population abundance | not applicable |
| Gating strategy | Primary viable cells were gated by forward scatter (FSC) vs. side scatter (SSC). After gating, deadcells were stained by LIVE/DEAD Fixable Near-IR Dead Cell Stain Kit (Thermo Fisher Scientific) and then excluded as the positive cells. |

☒ Tick this box to confirm that a figure exemplifying the gating strategy is provided in the Supplementary Information.

