## [Peer Review File · Nature Microbiology]

Intragenic viral silencer element regulates HTLV-1 latency via RUNX complex recruitment

Corresponding Author: Professor Yorifumi Satou

Version 0:

Reviewer comments:

Reviewer #1

(Remarks to the Author)

The authors describe the discovery of a specific regulatory region in the center of the HTLV-1 genome that can suppress viral transcription from the 5-LTR promoter through recruitment of the transcription factor RUNX1. The work addresses an important and interesting research question, and is technically elegant with a range of complementary assays and experiments. I have the following suggestions:

1. The authors argue that the 5-LTR promoter activity does not differ between HIV and HTLV-1 infected cells (Figure 1b), however, the OCR is only present in HTLV. This raises concerns about the functional significance of the OCR for silencing the 5-LTR promoter of HTLV.
2. Figure 1e suggests that the OCR can suppress the HTLV 5-LTR promoter, however, this is only shown in reporter cell lines. The experiments described in Figure 1e with reporter cell lines are not very well described; specifically, the design of the reporter cell assay should be described more specifically, and additional data showing the validation of this assay should be shown. Experiments showing a direct effect of the OCR on 5-LTR promoter activity in primary cells should be considered.
3. Figure 3c shows that binding of ETS1 to the OCR can stimulate HTLV transcription; therefore, it appears that the OCR/RSR can activate HTLV transcription in specific settings, specifically when ETS1 is highly expressed. Can the authors provide expression intensities of ETS1 and RUNX1 in HTLV-1 infected PBMC from study participants? How do ETS and RUNX1 expression change during the course of infection?
4. In Figure 3, the authors show that a RUNX1 inhibitor can increase HTLV gene expression and enhance the immunological visibility of infected cells. Can HTLV-specific T cells better kill virally-infected cells after treatment with the RUNX1 inhibitor?
5. In Figure 4i, the authors show that ETS1 expression is reduced in burst HTLV-1 infected cells; however, ETS1 binding to the OCR enhances viral gene expression per the authors' prior data. How do the authors explain this?
6. Can the authors be more specific why the OCR seems to mostly affect the 5-LTR promoter and not the 3-LTR promoter in HTLV-1?
7. In Figure 5, the authors suggest that the OCR integrated at the nef location in HIV can suppress the 5-LTR activity of HIV. It is not clear how this is mechanistically possible, since nef and the 5-LTR are separated by a considerable genetic distance (around 7KB).
8. From a higher level perspective, the data suggest that the HTLV genome contains genetic elements that can regulate proviral transcription and latency in a more autonomous fashion, while HIV lacks such features and largely depends on host genomic features to modulate viral gene expression/repression (Einkauf et al, Cell 2022). Consistent with such a view, selection of HIV proviruses in specific chromosomal regions seems to be more obvious compared to HTLV, although some recent data suggest selective persistence of HTLV in specific chromatin regions as well (Melamed et al, Science Advances 2023). The authors may wish to discuss this in more detail, and cite the relevant literature.

Reviewer #2

(Remarks to the Author)

Reviewer #3

(Remarks to the Author)

The authors identify a novel open chromatin region in the HTLV-1 provirus. Using in vitro experiments, they find that the region functions as a transcriptional silencer for the HTLV-1 5'-LTR promoter. They show that the transcription factor RUNX1 (and some co-factors) binds to this OCR (named RSR = retrovirus silencer region) and plays a role in the silencer function. Mutant viruses that have impaired RUNX1 binding capacity show higher proviral expression, virus production and immunogenicity in vitro. They insert the silencer within a cloned HIV-1 provirus and this decreases proviral expression, virus production and cytopathic effect.

This is a very elegant study that describes the identification of a novel regulatory region within HTLV-1. It is an important discovery particularly in the field of HTLV-1 which causes devastating diseases, ATL and HAM/TSP. The in vitro experiments have been performed with great attention to every single detail, the single-cell RNA-seq/ATAC-seq work is remarkable, however the authors further claim their findings pertain to “retroviruses” in general and that the absence of this silencer alone can explain the productive phenotype of HIV-1. Also, there is little, if none, evidence from in vivo observations.

General comments:

Overall generalization of the authors' findings to “retrovirus” (also in the title) while this was described for HTLV-1, not for any other retrovirus to our knowledge.

OCR is renamed RSR by the authors = retrovirus silencer region which points to a broad mechanism used by (human) retroviruses in general. Our understanding, according to the authors' findings, is that this RSR is specific of HTLV-1. The authors mention in the discussion and elsewhere that the silencer-related mechanism they uncovered is not used by endogenous retroviruses and other exogenous retroviruses (murine leukemia virus?).

HIV: There are many similarities between HTLV-1 and HIV-1, however, there are many differences as well, and simplifying the mechanisms by which HTLV-1 causes cancer (latency) and HIV-1 immunodeficiency (expression, viremia, cell killing) to the presence or absence of a silencer is not a reasonable claim. HTLV-1 is a delta retrovirus. HIV-1 is a lentivirus. Although we agree these viruses share a number of structural features, there are also many differences between these viruses which contribute to their distinct biological behavior. Tax and Tat are very different in structure, and HIV-1 has additional ORFs (vif, vpr, nef ...), the LTRs are significantly different, and there is little consensus regarding antisense expression in HIV-1, many features the authors do not comment on. In the paper, both viruses are “simplified” to Tax and Tat. We also miss a discussion regarding the antisense transcripts (RNA and/or protein for HTLV-1, HBZ) and how they fit within the story of the novel described silencer region.

The HIV-1 latent reservoir is the major problem in the HIV-1 clinics nowadays. Upon interruption of anti retroviral (tri-)therapy, the virus rebounds. Thus, HIV-1 is a champion in hiding from the immune system in these reservoirs. The authors mention productive infection, viral expression, but they do not address this essential point.

The authors could verify their findings for other delta-retroviruses like STLV-1, the type-C HTLV-1 viruses specifically found in Australia, and HTLV-2? Is this silencer found in BLV, another delta-retrovirus?

Conservation of the OCR region between strains? This is shown in extended data Fig 2 f. Were these samples obtained from individuals living in Japan or did the authors look in other strains, originating from US/South America/Caribbean, Australian?

The authors claim that defective HTLV-1 proviruses are an exception. There is evidence from the literature about the frequent occurrence of 5'LTR or 5'LTR-extended deletion HTLV-1 proviruses (reported in several publications, significant proportions). Is this region systematically retained in the absence of a 5'LTR or is it part of the deletion(s)? The same question may be asked for tax-mutated HTLV-1 proviruses. Is this silencer still required when Tax is dysfunctional?

The in vitro work is outstanding (work in cell lines, transfection experiments). When primary cells are used (ATLs AI5, AI9, smoldering ATLs, ACs), there is little or no information about these samples, both in terms of PVL (“high PVL” in all samples is indicated in the additional information file, Is this the case in ACs, smoldering ATL cases used for single cell work and ATLs earlier in the paper), the clonal situation of the population and proviral structure, etc. This should be clarified for improved understanding.

Fig 1 a: RNA copies in plasma, this is OK, but what about the number of infected T-cells: copies proviral DNA/T-cell, is this correlated with RNA?

For HIV-1 individuals (58), how were they selected?

For HTLV-1 individuals (17), the methods say “from ACs”, there is a need to have an idea of the number of infected T-cells in these individuals which may or may not influence viral RNA levels

Fig 1 c: w39 and w51 are Jurkat (JET) cells infected and cloned. Is there a single provirus or multiple? Why were these clones specifically selected? Any information regarding the genomic location of the provirus or is this feature not relevant?

Fig 1c: cloned HTLV-1 infected JET cells + two ATLs (ex vivo), for HIV-1 the authors use cell lines with “latent” HIV-1. How is this latency explained, in vitro? The authors may want to add primary cells of HIV-1 infected people for improvement.

Prediction

Prediction of transcription factor binding to the RSR in methods, requires more detailed information (i.e. in supplemental methods section). This refers to SNPs in HTLV-1 infected individuals described by Katsuya (2019). Can the authors include HTLV-1 proviral sequences from different origins, not just a single geographic region and a single paper.

Validation of transcription factor binding prediction with ATAC-seq motif enrichment analysis: elusive description of the method, please clarify or improve the sentences (Peak calling was performed from bam file of ATAC-seq using MACS2 where first bam file converted to BEDPE and then callpeak (options: -f BEDPE --nomodel --shift -37 --extsize 73 -g hs -B --keep-dup all).

Sometimes RUNX is used, at other places RUNX1, be consistent in denominations or RUNX family if all

About RUNX1: can the authors more extensively comment on the host-related context in which RUNX1 DNA binding is involved. Where are such motifs located in the host?

Figure 2:

Since these TFs localized to the 5'-LTR as well (Fig. 2a, b), we also analyzed the effect of these TFs on the 5'-LTR itself. These TFs affected promoter activity of the 5'-LTR; however, ...

Luc assays: test the effect on the 3'-LTR promoter activity, IGV shows peaks on both 5' and 3' LTR, given the sequence is similar, and we agree this doesn't mean the 3'LTR is involved. But can the authors exclude a 3'LTR origin?

Order of sub-figures a, b, ... f, g, ... is confusing

"defective" 5'-LTR suppression: do the authors mean "impaired"? g) viral gene expression is "suppressed", it is decreased, not suppressed

Figure 3:

"(b) Experimental workflow illustrates the establishment of stable reporter cells (JET cells) infected with wt- and s-mut-HTLV-1. HTLV-1-infected JET cells were sorted using tdTomato driven by Tax responsive elements as infection marker." Any idea of the pattern of these cell lines in terms of provirus, is it integrated, are there episomic forms, many integrations or just a single? Also important for figure 3g (RUNX inhibitor)

(f) s-mut-HTLV-1 JET cells show expansion of PVL during long-term culture after tdTomato sorting: did the authors check this for both cell lines (wt39 and wt51) and was this done in replicate?

Single-cell data/figure 4 + extended data figures.

Transcriptional burst (general): "This reactivation can cause infectious spread, allowing the virus to establish de novo infection from infected to uninfected individuals". Has this been effectively demonstrated in vivo? Is there a consensus in the field agreeing on the fact that besides clonal expansion there is also de novo infection at later stages?

"To this end, we cultivated primary CD4+ T cells from three indolent type of ATL cases ex vivo overnight to induce transcriptional burst and performed single cell (sc) multiome ...":

Why have smoldering ATLs been chosen for this experiment (a5, a6, a9)? What is the rationale for using ATLs and not infected non-transformed cell populations, and why specifically these three patients, how were they selected? Information regarding PVL is not available unless we missed it. Is this consistent with the single cell based detection of infected cells? Do all cells belong to the same malignant clone, defined as carrying a single integration site as expected in ATL samples? If this is the case, will this bias the findings and conclusions? Could the "burst" cluster and the "latent" cluster be determined by the genomic environment?

"Infected cells were identified by their transcriptional characteristics (Fig. 4c, Extended Data Fig. 4c and 5c; see Method for more in details), and we further refined cell clustering using only the infected cells."

"Method: a total of 15,567 cells before ex vivo cultivation (8,486 a5, 3,682 a6, 3,399 a9) and 22577 after ex vivo cultivation (8,375 a5, 10,160 a6, 4,042 a9) were retained for downstream analysis. Then: A total of 6,344 cells before ex vivo cultivation (3,764 a5, 620 a6, 1,960 a9) and 9275 cells after ex vivo cultivation (4,172 a5, 2,464 a6, 2,639 a9) were defined as infected cells. "

Is this consistent, what is the relative contribution to the detection of virus-infected cells by each of the markers used: sense expression, antisense expression, and ATAC-seq peak calling (virus) before cultivation? Do they all three match? And how does each of them relate to the same markers after cultivation?

In latent cells, the antisense is barely found (Fig 4 h, and extended) while in burst cells it is expressed. Is antisense "silencing" expected in latent cells and how does this relate to 3'LTR activity?

To what extent is the host transcriptome of a cell modified upon cell culture, in both uninfected and infected cells? Is it related to burst or latency, and do sense and antisense transcripts (and their corresponding proteins + which proteins amongst all virus-produced) have a direct effect?

Could the authors better explain and provide a visual representation of how they use ATAC-seq marks in single cell ATAC-seq to identify HTLV-1 infected cells?

What does it look like if PCA and UMPAs are generated separately for ATAC-seq and scRNAseq?

CTL response:

"We performed ELISpot assay and found that infected cells with the s-mut virus showed higher immunogenicity than those with the wild-type virus (Fig. 3i and Extended Data Fig. 4g, h). The result establishes the immunological significance of the RSR by controlling viral antigen Tax."

The in vitro experiment is elegant and does effectively suggest RSR control of Tax expression, which was shown by the experiments described earlier in the manuscript. That the increase in Tax expression has an effect on the viral immune response is known since many years. Tax is a dominant CTL-inducing viral product. A positive ELISPOT assay is thus expected in any system that enhances Tax expression. It does not confirm that RSR contributes to mechanisms underlying the modulation of immune response in vivo.

Other comments:

(Introduction) "HTLV-1 uniquely undergoes a self-induced latency, leading to a spontaneous reduction of viremia, obviating the

need of anti-retroviral drugs.” Antiretroviral therapy (ZDV IFN-alpha) is used to treat ATL patients and this treatment is increasingly considered worldwide. Antiretroviral drugs are in the pipeline as candidates for preemptive intervention.

Methods, recombinant HIV: To measure integrated rHIV-1 after 24 hours, genomic DNA in the Jurkat cells was extracted ... Is it possible to discriminate between integrated and episomal forms (double stranded unintegrated provirus, and do the authors think integration is fully completed after 24 h.

Language: could be improved, some sentences are hard to understand and several errors can be easily fixed (examples below):

Line 236 a maker of HTLV-1

Line 293 aligned to HTLV-1

Line 461 as shown the above section

Reviewer #4

(Remarks to the Author)

In this manuscript, the authors identified a key regulatory element that represses sense transcription of the HTLV-1 provirus, demonstrating that RUNX1 is located at the RSR and plays a pivotal role in silencer function. They also found that insertion of the RSR into recombinant HIV-1 significantly reduced viral expression, virus production, and cytopathic effects. The authors' group recently showed that HTLV-1 antisense transcription is constitutively activated through the role of viral enhancers. This study further extends that previous work, demonstrating that HTLV-1 sense transcription is repressed through RSR and RUNX1. Overall, the new findings about RSR and RUNX1 have potential implications for future research and antiretroviral drug development. However, here are some suggestions to review and address for publication.

Major issues:

1) The author demonstrated RSR repress the transcriptional activity of 5'LTR through the RUNX1 and co-factors. However, the molecular mechanism How the distal RSR affect the transcriptional activity of the 5'LTR is lack. I recommend using 3C-based experiments to determine the topological association between the 5'LTR and RSR in JET cells infected with HTLV-1-wt and -s-mut-RSR using 3C-based experiments.

In 2016, Satou et al. reported the epigenetic properties of the HTLV-1 provirus. Is the distal regulation of RSR in 5'LTR activation associated with epigenetic properties? We recommend identifying epigenetic features such as histone modifications and DNA methylation in JET cells infected with HTLV-1-wt and -s-mut-RSR to elucidate the mechanism of distal regulation of RSR.

2) RSR(4349-4531) appears to be binding site for multiple transcription factors, including repressor RUNX1 and activator ETS1. The authors focus primarily on the role of RUNX1, as RSR appears to play a repressive role in the activation of 5'-LTR, whereas ETS1 activates the 5'LTR in a dose-dependent manner. Experiments exploring the interplay between RUNX1 and ETS1 is recommended, as RUNX1 and ETS1 have opposing effects on the same RSR.

Minor issues:

1) In figure 1e: Why does OCR only affect transcription of 5'LTRs and not 3'LTRs? The author should mention or discuss this in the text.

2) Line #145: Where are the data for following statement “293T cells that have no endogenous expression of these TFs”?

3) In figure 2c and 2e: In figure 2c, the addition of GATA3 did not significantly change the relative luciferase activity, but in figure 2e, the relative luciferase activity was clearly reduced in the mutations in the GATA3 binding sites. How can the author explain this discrepancy?

4) Line #152: A period is missing from the sentence.

5) In figure 2e and 3a: In figure 2e, detailed sequence information for all mutations is required. In figure 3a, the data of luciferase activity assay is redundant with figure 2e.

6) The lack of explanation of the data can be misleading to readers.

- Figure 2h: What is the NGFR in figure 2h?

- Figure 4a-f: The explanation of data in the main text and figure legends are so implicated. Also, the extended data Figure 4C does not link to transcriptional characteristics as mentioned in line #229.

- Figure 4j and Line #249: What is the TF motif activity? The description of the schematic model in figure 4k is also missing in the main text and figure legend.

Reviewer #5

(Remarks to the Author)

Satou and coworkers have compared HIV and HTLV-1, the two pathogenic retroviruses in humans. They focused their attention on understanding why HTLV-1 is not active in replicating itself, causing long years of latency whereas HIV is very active in replication and the onset and development of the consequent disease is quick. They identified a formerly unknown element in the HTLV-1 that may account for this difference. This work is novel solid and likely opens up a new area of research, with special emphasis on the involvement of RUNX-1 in the pathophysiology of HTLV-1.

While the study is very intriguing, there are a few points that would make the statements/conclusion of this paper not convincing/relevant enough.

Major issues;

1. Are there any mutations of HTLV-1 in the RSR region known?
2. If such mutation exists, the mutated HTLV-1 might cause higher viremia, and moreover, a juvenile or fulminant form of T-cell leukemia. There have been some reports of early onsets of ATL/HAM (Oliveira et al., J Trop Pediatr 2018 64: 151). Any insights on the connection?
3. The virological nature of the s-mut HTLV-1 would be interesting. Whether its capacity to create T-cell leukemia is augmented or not can be perhaps tested infecting this virus in vivo to normal human T-cells in humanized mice.
4. In addition to HTLV-1 mutations in the RSR region (noted above), there are individuals with RUNX1 mutations. Have there been known cases in which such individuals have contracted HTLV-1? Do they manifest fulminant T-cell leukemia?

Minor comments.

5. In the abstract (line 43) and introduction, repeatedly it is mentioned that the HTLV-1 infection leads to the expansion of infected cells over the uninfected cells and causes leukemia. However, the disease manifestation of HTLV-1 is not limited to malignancy. In fact, the combined percentages of HAM/TSP, uveitis and other immunological in which "inflammatory" nature of the infected cells would account for pathogenic involvements. Thus, this sentence is too simplified and misleading. It needs to be modified. For example, the potential increased proliferation would be seen in asymptomatic carriers as well – have there been studies indicating if proliferative propensity would indicate an increase in the later leukemic occurrence?
6. The authors chose RUNX1, ETS1 and GATA3 because their expression levels are higher than other family members in normal CD4 T-cells. Why? If HTLV-1 infection would elevate the levels of any TFs, would those not be more plausible candidates?
7. In line 205, it should read "freshly isolated PBMCs from HTLV-1 infected individuals", otherwise misleading.
8. How specific is Ro5-3335 inhibitor? Are other cellular factors/pathways known to be inhibited by this compound? If the specificity is not guaranteed, the effects observed here can be because of an inhibition of unknown pathways, not by the RUNX1/CBF inhibition.
9. Perhaps in humans (in vivo), mutations that would negate the latency (and/or expose highly immunogenic antigens) would be detrimental for the survival of the virus, but the cell culture system has no such restrictions. It would be interesting to examine any HTLV-1 infected cell lines for similar mutations. Perhaps these mutations may have more positive impacts on the host cell biology?

Decision Letter:

5th June 2024

Dear Yorifumi,

Thank you for your patience while your manuscript "Intragenic silencer regulates human retrovirus latency by recruiting RUNX1" was under peer-review at Nature Microbiology. Thank you also for your previous message. It would have been nice to meet you in person at the London office, however I am based in the Berlin office, so unfortunately I am not able to meet you there. Nonetheless, your study has now been seen by 5 referees, whose expertise and comments you will find at the end of this email. Please note that reviewer #2 and #5 submitted a joint review. Although they find your work of some potential interest, they have raised a number of concerns that will need to be addressed before we can consider publication of the work in Nature Microbiology.

In particular, reviewers #1 and #4 request more mechanistic insight into the regulatory silencer and enhancer elements (RUNX1 vs. ETS-1). Reviewer #4 has suggested specific assays to investigate this. The other reviewers have called into question the in vivo relevance of your findings. Reviewer #3 has requested to show whether the OCR also relevant for other delta-retroviruses, as well as other HTLV-1 strains from different geographical locations. This reviewer requested to clarify the differences between HIV-1 and HTLV-1. Reviewer #2 and #5 (joint review) wish to be informed on whether effects of mutations in the OCR or RUNX1 have been reported in people infected with HTLV-1 and suggested to investigate your findings in a humanized mouse model.

Should further experimental data allow you to address these criticisms, we would be happy to look at a revised manuscript.

We remain interested and enthusiastic about your manuscript, and realise that reviewers have made multiple requests. We can offer to read and discuss a revision plan, in order to ensure that the revisions made are sufficient without demanding experiments that aren't necessary to support the main claims of your manuscript. Please consider these reports and send us a rebuttal indicating what you can and cannot do in revision and we will then work with you to finalise the revision plan.

Please include a data availability statement as a separate section after Methods but before references, under the heading "Data

Availability". This section should inform readers about the availability of the data used to support the conclusions of your study. This information includes accession codes to public repositories (data banks for protein, DNA or RNA sequences, microarray, proteomics data etc...), references to source data published alongside the paper, unique identifiers such as URLs to data repository entries, or data set DOIs, and any other statement about data availability. At a minimum, you should include the following statement: "The data that support the findings of this study are available from the corresponding author upon request", mentioning any restrictions on availability. If DOIs are provided, we also strongly encourage including these in the Reference list (authors, title, publisher (repository name), identifier, year). For more guidance on how to write this section please see: <http://www.nature.com/authors/policies/data/data-availability-statements-data-citations.pdf>

* If you have not done so already we suggest that you begin to revise your manuscript so that it conforms to our Article format instructions at <http://www.nature.com/nmicrobiol/info/final-submission>. Refer also to any guidelines provided in this letter.

When submitting the revised version of your manuscript, please pay close attention to our [href="https://www.nature.com/nature-portfolio/editorial-policies/image-integrity">Digital Image Integrity Guidelines. and to the following points below:](https://www.nature.com/nature-portfolio/editorial-policies/image-integrity)

Link Redacted

Note: This url links to your confidential homepage and associated information about manuscripts you may have submitted or be reviewing for us. If you wish to forward this e-mail to co-authors, please delete this link to your homepage first.

Nature Microbiology is committed to improving transparency in authorship. As part of our efforts in this direction, we are now requesting that all authors identified as 'corresponding author' on published papers create and link their Open Researcher and Contributor Identifier (ORCID) with their account on the Manuscript Tracking System (MTS), prior to acceptance. This applies to primary research papers only. ORCID helps the scientific community achieve unambiguous attribution of all scholarly contributions. You can create and link your ORCID from the home page of the MTS by clicking on 'Modify my Springer Nature account'. For more information please visit [please visit www.springernature.com/orcid](http://www.springernature.com/orcid).

If you wish to submit a suitably revised manuscript we would hope to receive it within 6 months. If you cannot send it within this time, please let us know. We will be happy to consider your revision, even if a similar study has been accepted for publication at Nature Microbiology or published elsewhere (up to a maximum of 6 months).

Yours sincerely,

Reviewer Expertise:

- Referee #1: HIV-1, Retrovirus persistence
- Referee #2: HIV-1, HTLV-1, Provirus (joint review with referee #5)
- Referee #3: Genomics, HTLV-1, BLV, Viral Oncogenesis
- Referee #4: Virology, epigenetics
- Referee #5: HTLV:1

Reviewer Comments:

Reviewer #1 (Remarks to the Author):

The authors describe the discovery of a specific regulatory region in the center of the HTLV-1 genome that can suppress viral transcription from the 5-LTR promoter through recruitment of the transcription factor RUNX1. The work addresses an important and interesting research question, and is technically elegant with a range of complementary assays and experiments. I have the following suggestions:

1. The authors argue that the 5-LTR promoter activity does not differ between HIV and HTLV-infected cells (Figure 1b), however, the OCR is only present in HTLV. This raises concerns about the functional significance of the OCR for silencing the 5-LTR promoter of HTLV.
2. Figure 1e suggests that the OCR can suppress the HTLV 5-LTR promoter, however, this is only shown in reporter cell lines. The experiments described in Figure 1e with reporter cell lines are not very well described; specifically, the design of the reporter cell assay should be described more specifically, and additional data showing the validation of this assay should be shown. Experiments showing a direct effect of the OCR on 5-LTR promoter activity in primary cells should be considered.
3. Figure 3c shows that binding of ETS1 to the OCR can stimulate HTLV transcription; therefore, it appears that the OCR/RSR can activate HTLV transcription in specific settings, specifically when ETS1 is highly expressed. Can the authors provide expression intensities of ETS1 and RUNX1 in HTLV-infected PBMC from study participants? How do ETS and RUNX1 expression change during the course of infection?
4. In Figure 3, the authors show that a RUNX1 inhibitor can increase HTLV gene expression and enhance the immunological visibility of infected cells. Can HTLV-specific T cells better kill virally-infected cells after treatment with the RUNX1 inhibitor?
5. In Figure 4i, the authors show that ETS1 expression is reduced in burst HTLV-infected cells; however, ETS1 binding to the OCR enhances viral gene expression per the authors' prior data. How do the authors explain this?
6. Can the authors be more specific why the OCR seems to mostly affect the 5-LTR promoter and not the 3-LTR promoter in HTLV-1?
7. In Figure 5, the authors suggest that the OCR integrated at the nef location in HIV can suppress the 5-LTR activity of HIV. It is not clear how this is mechanistically possible, since nef and the 5-LTR are separated by a considerable genetic distance (around 7KB).
8. From a higher level perspective, the data suggest that the HTLV genome contains genetic elements that can regulate proviral transcription and latency in a more autonomous fashion, while HIV lacks such features and largely depends on host genomic features to modulate viral gene expression/repression (Einkauf et al, Cell 2022). Consistent with such a view, selection of HIV proviruses in specific chromosomal regions seems to be more obvious compared to HTLV, although some recent data suggest selective persistence of HTLV in specific chromatin regions as well (Melamed et al, Science Advances 2023). The authors may wish to discuss this in more detail, and cite the relevant literature.

Reviewer #2: see comments of reviewer #5

Reviewer #3 (Remarks to the Author):

The authors identify a novel open chromatin region in the HTLV-1 provirus. Using in vitro experiments, they find that the region functions as a transcriptional silencer for the HTLV-1 5'-LTR promoter. They show that the transcription factor RUNX1 (and some co-factors) binds to this OCR (named RSR = retrovirus silencer region) and plays a role in the silencer function. Mutant viruses that have impaired RUNX1 binding capacity show higher proviral expression, virus production and immunogenicity in vitro. They insert the silencer within a cloned HIV-1 provirus and this decreases proviral expression, virus production and cytopathic effect.

This is a very elegant study that describes the identification of a novel regulatory region within HTLV-1. It is an important discovery particularly in the field of HTLV-1 which causes devastating diseases, ATL and HAM/TSP. The in vitro experiments have been performed with great attention to every single detail, the single-cell RNA-seq/ATAC-seq work is remarkable, however the authors further claim their findings pertain to "retroviruses" in general and that the absence of this silencer alone can explain the productive phenotype of HIV-1. Also, there is little, if none, evidence from in vivo observations.

General comments:

Overall generalization of the authors' findings to "retrovirus" (also in the title) while this was described for HTLV-1, not for any other retrovirus to our knowledge.

OCR is renamed RSR by the authors = retrovirus silencer region which points to a broad mechanism used by (human) retroviruses in general. Our understanding, according to the authors' findings, is that this RSR is specific of HTLV-1. The authors mention in the discussion and elsewhere that the silencer-related mechanism they uncovered is not used by endogenous retroviruses and other exogenous retroviruses (murine leukemia virus?).

HIV: There are many similarities between HTLV-1 and HIV-1, however, there are many differences as well, and simplifying the mechanisms by which HTLV-1 causes cancer (latency) and HIV-1 immunodeficiency (expression, viremia, cell killing) to the presence or absence of a silencer is not a reasonable claim. HTLV-1 is a delta retrovirus. HIV-1 is a lentivirus. Although we agree these viruses share a number of structural features, there are also many differences between these viruses which contribute to their distinct biological behavior. Tax and Tat are very different in structure, and HIV-1 has additional ORFs (vif, vpr, nef ...), the LTRs are significantly different, and there is little consensus regarding antisense expression in HIV-1, many features the authors do not comment on. In the paper, both viruses are "simplified" to Tax and Tat. We also miss a discussion regarding the antisense transcripts (RNA and/or protein for HTLV-1, HBZ) and how they fit within the story of the novel described silencer region.

The HIV-1 latent reservoir is the major problem in the HIV-1 clinics nowadays. Upon interruption of anti retroviral (tri-)therapy, the virus rebounds. Thus, HIV-1 is a champion in hiding from the immune system in these reservoirs. The authors mention productive infection, viral expression, but they do not address this essential point.

The authors could verify their findings for other delta-retroviruses like STLV-1, the type-C HTLV-1 viruses specifically found in Australia, and HTLV-2? Is this silencer found in BLV, another delta-retrovirus?

Conservation of the OCR region between strains? This is shown in extended data Fig 2 f. Were these samples obtained from individuals living in Japan or did the authors look in other strains, originating from US/South America/Caribbean, Australian?

The authors claim that defective HTLV-1 proviruses are an exception. There is evidence from the literature about the frequent occurrence of 5'LTR or 5'LTR-extended deletion HTLV-1 proviruses (reported in several publications, significant proportions). Is this region systematically retained in the absence of a 5'LTR or is it part of the deletion(s)? The same question may be asked for tax-mutated HTLV-1 proviruses. Is this silencer still required when Tax is dysfunctional?

The in vitro work is outstanding (work in cell lines, transfection experiments). When primary cells are used (ATLs AI5, AI9, smoldering ATLs, ACs), there is little or no information about these samples, both in terms of PVL ("high PVL" in all samples is indicated in the additional information file, Is this the case in ACs, smoldering ATL cases used for single cell work and ATLs earlier in the paper), the clonal situation of the population and proviral structure, etc. This should be clarified for improved understanding.

Fig 1 a: RNA copies in plasma, this is OK, but what about the number of infected T-cells: copies proviral DNA/T-cell, is this correlated with RNA?

For HIV-1 individuals (58), how were they selected?

For HTLV-1 individuals (17), the methods say "from ACs", there is a need to have an idea of the number of infected T-cells in these individuals which may or may not influence viral RNA levels

Fig 1 c: w39 and w51 are Jurkat (JET) cells infected and cloned. Is there a single provirus or multiple? Why were these clones specifically selected? Any information regarding the genomic location of the provirus or is this feature not relevant?

Fig 1c: cloned HTLV-1 infected JET cells + two ATLs (ex vivo), for HIV-1 the authors use cell lines with "latent" HIV-1. How is this latency explained, in vitro? The authors may want to add primary cells of HIV-1 infected people for improvement.

Prediction

Prediction of transcription factor binding to the RSR in methods, requires more detailed information (i.e. in supplemental methods section). This refers to SNPs in HTLV-1 infected individuals described by Katsuya (2019). Can the authors include HTLV-1 proviral sequences from different origins, not just a single geographic region and a single paper.

Validation of transcription factor binding prediction with ATAC-seq motif enrichment analysis: elusive description of the method, please clarify or improve the sentences (Peak calling was performed from bam file of ATAC-seq using MACS2 where first bam file converted to BEDPE and then callpeak (options: -f BEDPE --nomodel --shift -37 --extsize 73 -g hs -B --keep-dup all)).

Sometimes RUNX is used, at other places RUNX1, be consistent in denominations or RUNX family if all

About RUNX1: can the authors more extensively comment on the host-related context in which RUNX1 DNA binding is involved. Where are such motifs located in the host?

Figure 2:

Since these TFs localized to the 5'-LTR as well (Fig. 2a, b), we also analyzed the effect of these TFs on the 5'-LTR itself. These TFs affected promoter activity of the 5'-LTR; however, ...

Luc assays: test the effect on the 3'-LTR promoter activity, IGV shows peaks on both 5' and 3' LTR, given the sequence is similar, and we agree this doesn't mean the 3'LTR is involved. But can the authors exclude a 3'LTR origin?

Order of sub-figures a, b, ... f, g, ... is confusing

"defective" 5'-LTR suppression: do the authors mean "impaired"? g) viral gene expression is "suppressed", it is decreased, not suppressed

Figure 3:

"(b) Experimental workflow illustrates the establishment of stable reporter cells (JET cells) infected with wt- and s-mut-HTLV-1. HTLV-1-infected JET cells were sorted using tdTomato driven by Tax responsive elements as infection marker:" Any idea of the pattern of these cell lines in terms of provirus, is it integrated, are there episomic forms, many integrations or just a single? Also important for figure 3g (RUNX inhibitor)

(f) s-mut-HTLV-1 JET cells show expansion of PVL during long-term culture after tdTomato sorting: did the authors check this for both cell lines (wt39 and wt51) and was this done in replicate?

Single-cell data/figure 4 + extended data figures.

Transcriptional burst (general): "This reactivation can cause infectious spread, allowing the virus to establish de novo infection from infected to uninfected individuals". Has this been effectively demonstrated in vivo? Is there a consensus in the field agreeing on the fact that besides clonal expansion there is also de novo infection at later stages?

“To this end, we cultivated primary CD4+ T cells from three indolent type of ATL cases ex vivo overnight to induce transcriptional burst and performed single cell (sc) multiome ...”:

Why have smoldering ATLs been chosen for this experiment (a5, a6, a9)? What is the rationale for using ATLs and not infected non-transformed cell populations, and why specifically these three patients, how were they selected? Information regarding PVL is not available unless we missed it. Is this consistent with the single cell based detection of infected cells? Do all cells belong to the same malignant clone, defined as carrying a single integration site as expected in ATL samples? If this is the case, will this bias the findings and conclusions? Could the “burst” cluster and the “latent” cluster be determined by the genomic environment?

“Infected cells were identified by their transcriptional characteristics (Fig. 4c, Extended Data Fig. 4c and 5c; see Method for more in details), and we further refined cell clustering using only the infected cells.”

“Method: a total of 15,567 cells before ex vivo cultivation (8,486 a5, 3,682 a6, 3,399 a9) and 22577 after ex vivo cultivation (8,375 a5, 10,160 a6, 4,042 a9) were retained for downstream analysis. Then: A total of 6,344 cells before ex vivo cultivation (3,764 a5, 620 a6, 1,960 a9) and 9275 cells after ex vivo cultivation (4,172 a5, 2,464 a6, 2,639 a9) were defined as infected cells. “

Is this consistent, what is the relative contribution to the detection of virus-infected cells by each of the markers used: sense expression, antisense expression, and ATAC-seq peak calling (virus) before cultivation? Do they all three match? And how does each of them relate to the same markers after cultivation?

In latent cells, the antisense is barely found (Fig 4 h, and extended) while in burst cells it is expressed. Is antisense “silencing” expected in latent cells and how does this relate to 3’LTR activity?

To what extent is the host transcriptome of a cell modified upon cell culture, in both uninfected and infected cells? Is it related to burst or latency, and do sense and antisense transcripts (and their corresponding proteins + which proteins amongst all virus-produced) have a direct effect?

Could the authors better explain and provide a visual representation of how they use ATAC-seq marks in single cell ATAC-seq to identify HTLV-1 infected cells?

What does it look like if PCA and UMPAs are generated separately for ATAC-seq and scRNAseq?

CTL response:

“We performed ELISpot assay and found that infected cells with the s-mut virus showed higher immunogenicity than those with the wild-type virus (Fig. 3i and Extended Data Fig. 4g, h). The result establishes the immunological significance of the RSR by controlling viral antigen Tax.”

The in vitro experiment is elegant and does effectively suggest RSR control of Tax expression, which was shown by the experiments described earlier in the manuscript. That the increase in Tax expression has an effect on the viral immune response is known since many years. Tax is a dominant CTL-inducing viral product. A positive ELISPOT assay is thus expected in any system that enhances Tax expression. It does not confirm that RSR contributes to mechanisms underlying the modulation of immune response in vivo.

Other comments:

(Introduction) “HTLV-1 uniquely undergoes a self-induced latency, leading to a spontaneous reduction of viremia, obviating the need of anti-retroviral drugs.” Antiretroviral therapy (ZDV IFN-alpha) is used to treat ATL patients and this treatment is increasingly considered worldwide. Antiretroviral drugs are in the pipeline as candidates for preemptive intervention.

Methods, recombinant HIV: To measure integrated rHIV-1 after 24 hours, genomic DNA in the Jurkat cells was extracted ... Is it possible to discriminate between integrated and episomal forms (double stranded unintegrated provirus, and do the authors think integration is fully completed after 24 h.

Language: could be improved, some sentences are hard to understand and several errors can be easily fixed (examples below):

Line 236 a maker of HTLV-1

Line 293 aligned to HITL-1

Line 461 as shown the above section

Reviewer #4 (Remarks to the Author):

In this manuscript, the authors identified a key regulatory element that represses sense transcription of the HTLV-1 provirus, demonstrating that RUNX1 is located at the RSR and plays a pivotal role in silencer function. They also found that insertion of the RSR into recombinant HIV-1 significantly reduced viral expression, virus production, and cytopathic effects. The authors' group recently showed that HTLV-1 antisense transcription is constitutively activated through the role of viral enhancers. This study further extends that previous work, demonstrating that HTLV-1 sense transcription is repressed through RSR and RUNX1. Overall, the new findings about RSR and RUNX1 have potential implications for future research and antiretroviral drug development. However, here are some suggestions to review and address for publication.

Major issues:

1) The author demonstrated RSR repress the transcriptional activity of 5’LTR through the RUNX1 and co-factors. However, the molecular mechanism How the distal RSR affect the transcriptional activity of the 5’LTR is lack. I recommend using 3C-based experiments to determine the topological association between the 5’LTR and RSR in JET cells infected with HTLV-1-wt and -s-

mut-RSR using 3C-based experiments.

In 2016, Satou et al. reported the epigenetic properties of the HTLV-1 provirus. Is the distal regulation of RSR in 5'LTR activation associated with epigenetic properties? We recommend identifying epigenetic features such as histone modifications and DNA methylation in JET cells infected with HTLV-1-wt and -s-mut-RSR to elucidate the mechanism of distal regulation of RSR.

2) RSR(4349-4531) appears to be binding site for multiple transcription factors, including repressor RUNX1 and activator ETS1. The authors focus primarily on the role of RUNX1, as RSR appears to play a repressive role in the activation of 5'-LTR, whereas ETS1 activates the 5'LTR in a dose-dependent manner. Experiments exploring the interplay between RUNX1 and ETS1 is recommended, as RUNX1 and ETS1 have opposing effects on the same RSR.

Minor issues:

- 1) In figure 1e: Why does OCR only affect transcription of 5'LTRs and not 3'LTRs? The author should mention or discuss this in the text.
- 2) Line #145: Where are the data for following statement "293T cells that have no endogenous expression of these TFs"?
- 3) In figure 2c and 2e: In figure 2c, the addition of GATA3 did not significantly change the relative luciferase activity, but in figure 2e, the relative luciferase activity was clearly reduced in the mutations in the GATA3 binding sites. How can the author explain this discrepancy?
- 4) Line #152: A period is missing from the sentence.
- 5) In figure 2e and 3a: In figure 2e, detailed sequence information for all mutations is required. In figure 3a, the data of luciferase activity assay is redundant with figure 2e.
- 6) The lack of explanation of the data can be misleading to readers.
 - Figure 2h: What is the NGFR in figure 2h?
 - Figure 4a-f: The explanation of data in the main text and figure legends are so implicated. Also, the extended data Figure 4C does not link to transcriptional characteristics as mentioned in line #229.
 - Figure 4j and Line #249: What is the TF motif activity? The description of the schematic model in figure 4k is also missing in the main text and figure legend.

Reviewer #5 (Remarks to the Author):

Satou and coworkers have compared HIV and HTLV-1, the two pathogenic retroviruses in humans. They focused their attention on understanding why HTLV-1 is not active in replicating itself, causing long years of latency whereas HIV is very active in replication and the onset and development of the consequent disease is quick. They identified a formerly unknown element in the HTLV-1 that may account for this difference. This work is novel solid and likely opens up a new area of research, with special emphasis on the involvement of RUNX-1 in the pathophysiology of HTLV-1.

While the study is very intriguing, there are a few points that would make the statements/conclusion of this paper not convincing/relevant enough.

Major issues;

1. Are there any mutations of HTLV-1 in the RSR region known?
2. If such mutation exists, the mutated HTLV-1 might cause higher viremia, and moreover, a juvenile or fulminant form of T-cell leukemia. There have been some reports of early onsets of ATL/HAM (Oliveira et al., J Trop Pediatr 2018 64: 151). Any insights on the connection?
3. The virological nature of the s-mut HTLV-1 would be interesting. Whether its capacity to create T-cell leukemia is augmented or not can be perhaps tested infecting this virus in vivo to normal human T-cells in humanized mice.
4. In addition to HTLV-1 mutations in the RSR region (noted above), there are individuals with RUNX1 mutations. Have there been known cases in which such individuals have contracted HTLV-1? Do they manifest fulminant T-cell leukemia?

Minor comments.

5. In the abstract (line 43) and introduction, repeatedly it is mentioned that the HTLV-1 infection leads to the expansion of infected cells over the uninfected cells and causes leukemia. However, the disease manifestation of HTLV-1 is not limited to malignancy. In fact, the combined percentages of HAM/TSP, uveitis and other immunological in which "inflammatory" nature of the infected cells would account for pathogenic involvements. Thus, this sentence is too simplified and misleading. It needs to be modified. For example, the potential increased proliferation would be seen in asymptomatic carriers as well – have there been studies indicating if proliferative propensity would indicate an increase in the later leukemic occurrence?
6. The authors chose RUNX1, ETS1 and GATA3 because their expression levels are higher than other family members in normal CD4 T-cells. Why? If HTLV-1 infection would elevate the levels of any TFs, would those not be more plausible candidates?
7. In line 205, it should read "freshly isolated PBMCs from HTLV-1 infected individuals", otherwise misleading.
8. How specific is Ro5-3335 inhibitor? Are other cellular factors/pathways known to be inhibited by this compound? If the specificity is not guaranteed, the effects observed here can be because of an inhibition of unknown pathways, not by the RUNX1/CBF inhibition.
9. Perhaps in humans (in vivo), mutations that would negate the latency (and/or expose highly immunogenic antigens) would be detrimental for the survival of the virus, but the cell culture system has no such restrictions. It would be interesting to examine any

HTLV-1 infected cell lines for similar mutations. Perhaps these mutations may have more positive impacts on the host cell biology?

Version 1:

Reviewer comments:

Reviewer #1

(Remarks to the Author)

In this manuscript the authors identified an open chromatin region (OCR) within HTLV1 genome, responsible for transcriptional silencing of the provirus. OCR transcriptional silencing is mediated by direct recruitment of RUNX1/CBFb/HDAC3. Remarkably, mutation of the RUNX1 binding site within HTLV-1 OCR region results in enhanced viral replication and spreading in vitro. The data is convincing, and support the conclusions. The work is of great importance and will represent a forward step towards our understanding of HTLV-1 latency and associated pathology.

The authors' response to reviewer 1's comment regarding HIV-1 is accurate and satisfying. However, showing that the OCR function as transcriptional silencer of HIV-1 LTR in reporter assay and HIV-1 in replication assay does not make it a general human retrovirus silencer (HRS) region. If the authors want to keep such statement, additional experiments are required. Notably, extending the experiment shown in figure 1f-g and figure 2e to non-retroviral promoters (such as CMV and SV40) and to cellular promoters (TATAA containing and TATAA-less). In my opinion, the manuscript is strong enough to merit publication in Nature microbiology even without such statement.

Reviewer #2

(Remarks to the Author)

- This compelling report from Sugata et al. details an insightful analysis of the control mechanisms that dictate the intriguing dominance of latent or non-expressive nature of the transmissible virus particles from the HTLV-1 from the positive sense DNA of the integrated provirus while maintaining with a low pulse of a negative sense gene products that ensures ongoing proliferation of infected cells until a moment when a burst the Tax protein is expressed that greatly enhances an outpouring of infectious virion. The authors contrast this with the other better known human retrovirus, HIV-1, that dominantly maintains a high-level cytopathic expression of transmissible virion and only rarely establishes a latent state of infection. The work presented characterises a region of HTLV-1 proviral DNA that interacts with cellular transcription factors of the RUNX1 family and associating co-factors in a far-ranging study detailing the molecular control elements of HTLV-1 transcriptional latency. They present a very well designed and executed study and deliver an important finding that strongly reveals new insights into the control of both the HTLV-1 and HIV-1 infections.

The authors have generally addressed most reviewers' concerns, and the most significant shortfall is not sufficiently addressing the following concern: "In latent cells, the antisense is barely found (Fig 4 h, and extended) while in burst cells it is expressed. Is antisense "silencing" expected in latent cells and how does this relate to 3'LTR activity?" Authors have referred to 10X documentation, which suggests antisense reads can be overestimated. If strandedness can't be accurately assigned by scRNA-seq, it should not be presented on the figures. This is not a limitation as mentioned in the methods, but rather it is misleading. As pointed out, the near equal levels of antisense and sense transcription have implications for viral persistence, and if this is a technical artefact, it shouldn't be presented. Instead, it should be shown as viral expression (without quantifying sense and antisense). This is further supported by the qPCR results in Fig. 3d, which are inconsistent with scRNA-seq data presented, showing that changes to tax and hbz expression aren't equivalent when the OCR is disrupted.

However, there are a number of additional less direct concerns from the original reviewers that I recommend for further attention and clarification by the authors.

- The opening paragraph should be reframed. It is a false equivalence to compare regulatory mechanisms for endogenous (ERV) and exogenous retroviruses in this section, given the distinct evolutionary histories (resulting in the so-called "arms race" between ERVs and KZFPs), developmental regulatory pathway (endogenous retroviruses are subject to silencing that is established upon epigenetic reprogramming in early development, which can be stably maintained, whereas exogenous silencing needs de novo establishment), and genetic diversity. Authors should highlight these differences, which underpin why very different mechanisms are employed to control endogenous and exogenous retroviral expression. It also highlights the importance of the work presented in this manuscript, as authors describe a novel specific pathway for the control of exogenous retroviral expression.

- Given antiretroviral drugs for HTLV-1 are in the pipeline as candidates for pre-emptive intervention for infection, the statement "Obviating the need for anti-retroviral drugs" should be removed from the introduction.

- Can the authors clarify what the difference is between the 5' and 3' LTRs in the reporter assays apart from their orientation for the direction of transcription? Authors state that in NGS the 3' and 5' LTRs cannot be distinguished due to their identical

sequences, so how was this resolved for a reporter assay? As the data unfolds it becomes unclear how the position and orientation of the OCR impacts on the flexibility for transplanting the ascribed functions into other settings.

- RUNX1 binds to a suite of co-factors to effect transcription, either positively or negatively. Why specifically were SIN3A and HDAC3 selected for further characterisation, over other cofactors such as TLE, HDAC1, etc.. for example? This should be stated in text.

- Can the authors comment on difference observed between Jurkat cell lines and ATL samples for binding of repressive complexes at the OCR in Fig. 2a? Specifically, how this relates to transcription from the provirus in each context? Can they explain the difference observed? This should be discussed in text.

- RUNX1 behaviour as repressor or activator is regulated by PTMs, which isn't captured by correlation of repressive factor expression (Fig. 4i). Given the ubiquitous roles of cellular TFs and epigenetic modifiers, it's not clear what the relevance of correlating their expression would be to this model. This should be discussed in text, as it is likely more relevant than changes in expression of the transcriptional complex. Further, a much more informative representation of the data would be to create UMAP projections of all cells with a heatmap of expression levels for RUNX1, ETS1, GATA3, HDAC3, and SIN3A (as in Fig 4c) for all cells. This should be included as extended data.

- The model presented extended Fig. 10 is lacking a figure legend to explain authors' hypothetical dynamic model. Can the authors please provide one, detailing their model and caveats?

- RUNX1 is unique in that it has both activating and repressive functions, depending on co-factor binding and PTMs, making it an intriguing factor that could act as a molecular switch. We agree with authors, that there are many mechanistic insights that will be gained from future studies, which will be important. However, the following points should be addressed in the model and/or in text in the discussion.

1. Data show ETS1 seems to either compete with or modify RUNX1 repressive behaviour (Fig. 2C). ETS1 usually co-binds with RUNX1 to activate transcription. Changing stoichiometry of ETS1 binding is likely an important molecular switch for RUNX1 function. This is not reflected in the discussion.

2. Binding of RUNX1 and co-factors are inferred from population data from a very heterogenous population of cells, with regard to sense/antisense expression from the provirus. Binding dynamics and stoichiometry have not been resolved in cells with defined sense and antisense expression. This is an important consideration that should be mentioned in text.

3. RUNX1 behaviour as repressor or activator is regulated by PTMs, often catalysed by chromatin modifying complexes themselves. This should be mentioned in the model or discussion for future studies, particularly as this is likely to be a more important regulator than expression levels of co-factors.

- The manuscript would benefit from copy-editing before publication. For example:

Line 45:why HTLV-1 prefers... is an anthropomorphic term and should state..... how HTLV-1 evolved....., or perhaps ""it remains unknown how HTLV-1 natural selection has led to such a latency phenotype in contrast to HIV-1""

Line 77:....tropical, not tropic

Line 83:not "an equivalent", but an analogue

Line 126...."not being reported", but perhaps really....not known as...

Line 269..."reactive" should be "reactivate"

Reviewer #3

(Remarks to the Author)

I am generally satisfied with the authors' responses to my questions. They have adequately addressed most of my concerns and incorporated relevant changes in the revised manuscript, including the addition of new data in Figures (Fig. 1) and Extended Data Figures (several).

However, I have one remaining concern regarding the generalization in the title and the naming of the silencer region. The region should be specifically referred to as the HTLV-1 silencer region rather than the broader term human retroviral silencer region. Additionally, the title should explicitly mention HTLV-1, such as: 'Intragenic silencer regulates human T-cell leukemia virus-1 latency by recruiting RUNX1.'

Reviewer #4

(Remarks to the Author)

In the revised manuscript, the author addressed many of the reviewers' comments. The novel finding regarding OCR is particularly intriguing, as it is linked to the repression of the 5'-LTR.

Ext Fig 5c,d

In response to the reviewer's recommendation, the author conducted 3C analysis to investigate the association between OCR and the 5'-LTR. The results revealed that the wild-type OCR associates with the 5'-LTR, and this association is disrupted by mutations in the OCR sequence.

I wonder whether the association between OCR and the 3'-LTR is blocked by CTCF. It would be beneficial to examine this association and evaluate the role of CTCF in the process. Such insights would greatly enhance our understanding of the underlying mechanism of OCR-mediated gene regulation.

Ext Fig 5b.

Active histone markers are more broadly distributed in s-mut-HTLV-1 compared to wt-HTLV-1. I recommend checking the enrichment of histone markers specifically at the OCR position.

The enrichment of histone markers near the 3'-LTR may be higher in wt-HTLV-1 than in s-mut-HTLV-1. How can this result be explained?

Decision Letter:

23rd January 2025

Dear Yorifumi,

Thank you for your patience while your manuscript "Intragenic silencer regulates human retrovirus latency by recruiting RUNX1" was under peer-review at Nature Microbiology. As mentioned previously, we had to replace the original referees #1, #2 and #5. It has now been seen by 4 referees, whose expertise and comments you will find at the of this email. You will see from their comments below that while they find your work of interest, some important points are raised. We are very interested in the possibility of publishing your study in Nature Microbiology, but would like to consider your response to these concerns in the form of a revised manuscript before we make a final decision on publication.

In particular, you will see that referee #2 asks for clarifications about the strandness of the data, as well as textual revisions in the introduction, toning down statements in the results and discussion section, and adding more details in the methods. This referee also asks to provide a reanalysis of the RUNX1, ETS1, GATA3, HDAC3, and SIN3A expression and to present it as UMAP. Please also tone down statements that the OCR is an HRS, as requested by both referee #1 and #3 and adjust the title accordingly. While referee #4 requests to further investigate whether CTCF blocks interaction with OCR and the 3'-LTR, we would overrule the need for further experimental data and suggest to discuss this aspect in the discussion section as future direction, to address this point.

The rest referees' reports are clear and the remaining issues should be straightforward to address.

We would not require you to perform further experimentation, but would like you to address the referees comments by textual changes and reanalysis (replotting) of available data.

If you have not done so already please begin to revise your manuscript so that it conforms to our Article format instructions at <http://www.nature.com/nmicrobiol/info/final-submission/>

The usual length limit for a Nature Microbiology Article is six display items (figures or tables) and 3,500 words.

Some reduction could be achieved by focusing any introductory material and moving it to the start of your opening 'bold' paragraph, whose function is to outline the background to your work, describe in a sentence your new observations, and explain your main conclusions. The discussion should also be limited. Methods should be described in a separate section following the discussion, we do not place a word limit on Methods.

Nature Microbiology titles should give a sense of the main new findings of a manuscript, and should not contain punctuation. Please keep in mind that we strongly discourage active verbs in titles, and that they should ideally fit within 90 characters each (including spaces).

Please include a data availability statement as a separate section after Methods but before references, under the heading "Data Availability". This section should inform readers about the availability of the data used to support the conclusions of your study. This information includes accession codes to public repositories (data banks for protein, DNA or RNA sequences, microarray, proteomics data etc...), references to source data published alongside the paper, unique identifiers such as URLs to data repository entries, or data set DOIs, and any other statement about data availability. At a minimum, you should include the following statement: "The data that support the findings of this study are available from the corresponding author upon request", mentioning any restrictions on availability. If DOIs are provided, we also strongly encourage including these in the Reference list (authors, title, publisher (repository name), identifier, year). For more guidance on how to write this section please see: <http://www.nature.com/authors/policies/data/data-availability-statements-data-citations.pdf>

To improve the accessibility of your paper to readers from other research areas, please pay particular attention to the wording of the paper's opening bold paragraph, which serves both as an introduction and as a brief, non-technical summary in about 150 words. If, however, you require one or two extra sentences to explain your work clearly, please include them even if the paragraph is over-length as a result. The opening paragraph should not contain references. Because scientists from other sub-

disciplines will be interested in your results and their implications, it is important to explain essential but specialised terms concisely. We suggest you show your summary paragraph to colleagues in other fields to uncover any problematic concepts.

If your paper is accepted for publication, we will edit your display items electronically so they conform to our house style and will reproduce clearly in print. If necessary, we will re-size figures to fit single or double column width. If your figures contain several parts, the parts should form a neat rectangle when assembled. Choosing the right electronic format at this stage will speed up the processing of your paper and give the best possible results in print. We would like the figures to be supplied as vector files - EPS, PDF, AI or postscript (PS) file formats (not raster or bitmap files), preferably generated with vector-graphics software (Adobe Illustrator for example). Please try to ensure that all figures are non-flattened and fully editable. All images should be at least 300 dpi resolution (when figures are scaled to approximately the size that they are to be printed at) and in RGB colour format. Please do not submit Jpeg or flattened TIFF files. Please see also 'Guidelines for Electronic Submission of Figures' at the end of this letter for further detail.

Figure legends must provide a brief description of the figure and the symbols used, within 350 words, including definitions of any error bars employed in the figures.

When submitting the revised version of your manuscript, please pay close attention to our <https://www.nature.com/nature-research/editorial-policies/image-integrity> Digital Image Integrity Guidelines and to the following points below:

EXTENDED DATA FIGURES

Please include a statement before the acknowledgements naming the author to whom correspondence and requests for materials should be addressed.

Finally, we require authors to include a statement of their individual contributions to the paper -- such as experimental work, project planning, data analysis, etc. -- immediately after the acknowledgements. The statement should be short, and refer to authors by their initials. For details please see the Authorship section of our joint Editorial policies at http://www.nature.com/authors/editorial_policies/authorship.html

- * include a point-by-point response to any editorial suggestions and to our referees. Please include your response to the editorial suggestions in your cover letter, and please upload your response to the referees as a separate document.
- * ensure it complies with our format requirements for Letters as set out in our guide to authors at www.nature.com/nmicrobiol/info/gta/
- * state in a cover note the length of the text, methods and legends; the number of references; number and estimated final size of figures and tables
- * please include a version of your manuscript with the changes highlighted.
- * resubmit electronically if possible using the link below to access your home page:

Link Redacted

*This url links to your confidential homepage and associated information about manuscripts you may have submitted or be reviewing for us. If you wish to forward this e-mail to co-authors, please delete this link to your homepage first.

Please ensure that all correspondence is marked with your Nature Microbiology reference number in the subject line.

Nature Microbiology is committed to improving transparency in authorship. As part of our efforts in this direction, we are now requesting that all authors identified as 'corresponding author' on published papers create and link their Open Researcher and Contributor Identifier (ORCID) with their account on the Manuscript Tracking System (MTS), prior to acceptance. This applies to primary research papers only. ORCID helps the scientific community achieve unambiguous attribution of all scholarly contributions. You can create and link your ORCID from the home page of the MTS by clicking on 'Modify my Springer Nature

account'. For more information please visit www.springernature.com/orcid.

We hope to receive your revised paper within three weeks. If you cannot send it within this time, please let us know.

Yours sincerely,

Reviewer Expertise:

Referee #1: HIV, gene regulation
Referee #2: HTLV-1
Referee #3: HTLV-1, gene regulation
Referee #4: Epigenetics

Reviewers Comments:

Reviewer #1 (Remarks to the Author):

In this manuscript the authors identified an open chromatin region (OCR) within HTLV1 genome, responsible for transcriptional silencing of the provirus. OCR transcriptional silencing is mediated by direct recruitment of RUNX1/CBFb/HDAC3. Remarkably, mutation of the RUNX1 binding site within HTLV-1 OCR region results in enhanced viral replication and spreading in vitro. The data is convincing, and support the conclusions. The work is of great importance and will represent a forward step towards our understanding of HTLV-1 latency and associated pathology.

The authors' response to reviewer 1's comment regarding HIV-1 is accurate and satisfying. However, showing that the OCR function as transcriptional silencer of HIV-1 LTR in reporter assay and HIV-1 in replication assay does not make it a general human retrovirus silencer (HRS) region. If the authors want to keep such statement, additional experiments are required. Notably, extending the experiment shown in figure 1f-g and figure 2e to non-retroviral promoters (such as CMV and SV40) and to cellular promoters (TATAA containing and TATAA-less). In my opinion, the manuscript is strong enough to merit publication in Nature microbiology even without such statement.

Reviewer #2 (Remarks to the Author):

- This compelling report from Sugata et al. details an insightful analysis of the control mechanisms that dictate the intriguing dominance of latent or non-expressive nature of the transmissible virus particles from the HTLV-1 from the positive sense DNA of the integrated provirus while maintaining with a low pulse of a negative sense gene products that ensures ongoing proliferation of infected cells until a moment when a burst of the Tax protein is expressed that greatly enhances an outpouring of infectious virion. The authors contrast this with the other better known human retrovirus, HIV-1, that dominantly maintains a high-level cytopathic expression of transmissible virion and only rarely establishes a latent state of infection. The work presented characterises a region of HTLV-1 proviral DNA that interacts with cellular transcription factors of the RUNX1 family and associating co-factors in a far-ranging study detailing the molecular control elements of HTLV-1 transcriptional latency. They present a very well designed and executed study and deliver an important finding that strongly reveals new insights into the control of both the HTLV-1 and HIV-1 infections.

The authors have generally addressed most reviewers' concerns, and the most significant shortfall is not sufficiently addressing the following concern: "In latent cells, the antisense is barely found (Fig 4 h, and extended) while in burst cells it is expressed. Is antisense "silencing" expected in latent cells and how does this relate to 3'LTR activity?" Authors have referred to 10X documentation, which suggests antisense reads can be overestimated. If strandedness can't be accurately assigned by scRNA-seq, it should not be presented on the figures. This is not a limitation as mentioned in the methods, but rather it is misleading. As pointed out, the near equal levels of antisense and sense transcription have implications for viral persistence, and if this is a technical artefact, it shouldn't be presented. Instead, it should be shown as viral expression (without quantifying sense and antisense). This is further supported by the qPCR results in Fig. 3d, which are inconsistent with scRNA-seq data presented, showing that changes to tax and hbz expression aren't equivalent when the OCR is disrupted.

However, there are a number of additional less direct concerns from the original reviewers that I recommend for further attention and clarification by the authors.

- The opening paragraph should be reframed. It is a false equivalence to compare regulatory mechanisms for endogenous (ERV) and exogenous retroviruses in this section, given the distinct evolutionary histories (resulting in the so-called "arms race" between ERVs and KZFPs), developmental regulatory pathway (endogenous retroviruses are subject to silencing that is established upon epigenetic reprogramming in early development, which can be stably maintained, whereas exogenous silencing needs de novo establishment), and genetic diversity. Authors should highlight these differences, which underpin why very different mechanisms are employed to control endogenous and exogenous retroviral expression. It also highlights the importance of the work presented in this manuscript, as authors describe a novel specific pathway for the control of exogenous

retroviral expression.

- Given antiretroviral drugs for HTLV-1 are in the pipeline as candidates for pre-emptive intervention for infection, the statement "Obviating the need for anti-retroviral drugs" should be removed from the introduction.
- Can the authors clarify what the difference is between the 5' and 3' LTRs in the reporter assays apart from their orientation for the direction of transcription? Authors state that in NGS the 3' and 5' LTRs cannot be distinguished due to their identical sequences, so how was this resolved for a reporter assay? As the data unfolds it becomes unclear how the position and orientation of the OCR impacts on the flexibility for transplanting the ascribed functions into other settings.
- RUNX1 binds to a suite of co-factors to effect transcription, either positively or negatively. Why specifically were SIN3A and HDAC3 selected for further characterisation, over other cofactors such as TLE, HDAC1, etc.. for example? This should be stated in text.
- Can the authors comment on difference observed between Jurkat cell lines and ATL samples for binding of repressive complexes at the OCR in Fig. 2a? Specifically, how this relates to transcription from the provirus in each context? Can they explain the difference observed? This should be discussed in text.
- RUNX1 behaviour as repressor or activator is regulated by PTMs, which isn't captured by correlation of repressive factor expression (Fig. 4i). Given the ubiquitous roles of cellular TFs and epigenetic modifiers, it's not clear what the relevance of correlating their expression would be to this model. This should be discussed in text, as it is likely more relevant than changes in expression of the transcriptional complex. Further, a much more informative representation of the data would be to create UMAP projections of all cells with a heatmap of expression levels for RUNX1, ETS1, GATA3, HDAC3, and SIN3A (as in Fig 4c) for all cells. This should be included as extended data.
- The model presented extended Fig. 10 is lacking a figure legend to explain authors' hypothetical dynamic model. Can the authors please provide one, detailing their model and caveats?
- RUNX1 is unique in that it has both activating and repressive functions, depending on co-factor binding and PTMs, making it an intriguing factor that could act as a molecular switch. We agree with authors, that there are many mechanistic insights that will be gained from future studies, which will be important. However, the following points should be addressed in the model and/or in text in the discussion.
 1. Data show ETS1 seems to either compete with or modify RUNX1 repressive behaviour (Fig. 2C). ETS1 usually co-binds with RUNX1 to activate transcription. Changing stoichiometry of ETS1 binding is likely an important molecular switch for RUNX1 function. This is not reflected in the discussion.
 2. Binding of RUNX1 and co-factors are inferred from population data from a very heterogenous population of cells, with regard to sense/antisense expression from the provirus. Binding dynamics and stoichiometry have not been resolved in cells with defined sense and antisense expression. This is an important consideration that should be mentioned in text.
 3. RUNX1 behaviour as repressor or activator is regulated by PTMs, often catalysed by chromatin modifying complexes themselves. This should be mentioned in the model or discussion for future studies, particularly as this is likely to be a more important regulator than expression levels of co-factors.
- The manuscript would benefit from copy-editing before publication. For example:
 - Line 45:why HTLV-1 prefers... is an anthropomorphic term and should state..... how HTLV-1 evolved....., or perhaps ""it remains unknown how HTLV-1 natural selection has led to such a latency phenotype in contrast to HIV-1""
 - Line 77:....tropical, not tropic
 - Line 83:not "an equivalent", but an analogue
 - Line 126....."not being reported", but perhaps really....not known as...
 - Line 269..."reactive" should be "reactivate"

Reviewer #3 (Remarks to the Author):

I am generally satisfied with the authors' responses to my questions. They have adequately addressed most of my concerns and incorporated relevant changes in the revised manuscript, including the addition of new data in Figures (Fig. 1) and Extended Data Figures (several).

However, I have one remaining concern regarding the generalization in the title and the naming of the silencer region. The region should be specifically referred to as the HTLV-1 silencer region rather than the broader term human retroviral silencer region. Additionally, the title should explicitly mention HTLV-1, such as: 'Intragenic silencer regulates human T-cell leukemia virus-1 latency by recruiting RUNX1.'

Reviewer #4 (Remarks to the Author):

In the revised manuscript, the author addressed many of the reviewers' comments. The novel finding regarding OCR is particularly intriguing, as it is linked to the repression of the 5'-LTR.

Ext Fig 5c,d

In response to the reviewer's recommendation, the author conducted 3C analysis to investigate the association between OCR and the 5'-LTR. The results revealed that the wild-type OCR associates with the 5'-LTR, and this association is disrupted by mutations in the OCR sequence.

I wonder whether the association between OCR and the 3'-LTR is blocked by CTCF. It would be beneficial to examine this association and evaluate the role of CTCF in the process. Such insights would greatly enhance our understanding of the underlying mechanism of OCR-mediated gene regulation.

Ext Fig 5b.

Active histone markers are more broadly distributed in s-mut-HTLV-1 compared to wt-HTLV-1. I recommend checking the enrichment of histone markers specifically at the OCR position.

The enrichment of histone markers near the 3'-LTR may be higher in wt-HTLV-1 than in s-mut-HTLV-1. How can this result be explained?

Version 2:

Reviewer comments:

Reviewer #2

(Remarks to the Author)

The revisions in reply to the reviewer comments generally address the concerns well and improve the quality and impact of the paper. There are three small things to further consider:

1) The change to the title of the paper to address concerns from reviewer #1 and #3 in focusing the findings to HTLV-1 is welcome. In addition, the last sentence of the opening bolded paragraph should change to "...offering new insights into retroviral evolution..." to complete the focusing of this work to HTLV-1. However, the new version of the title loses the impact of the identification of the intragenic positional location. A further refinement to the title might reinstate "intragenic" into the new title that specifies the HTLV-1 retrovirus.

2) The explanation of the differences between the 5' and 3'LTRs that share identical sequence would be further improved at line 128 by adding one additional word (antisense)....."the 3'-LTR lacks an antisense TATA box...."

3) The correction to Extended Figure 5b is welcomed, but I believe the label on the extreme left column of each panel be "3'LTR", and not "5'LTR".

Otherwise, all the revisions made this excellent paper have improved the clarity and quality of this important study.

Reviewer #6

(Remarks to the Author)

The authors have spoken to the reviewer concerns and addressed them, significantly. The new additions to Figure 4i are low resolution, and should be replaced prior to publication.

With the additional clarifications and analyses, this work provides an important contribution towards understanding HTLV-1 regulation and latency, and addresses gaps in knowledge regarding host-viral interactions, and transcriptional regulation of the HTLV-1 provirus, and retroviral regulation more broadly.

Decision Letter:

Our ref: NMICROBIOL-24041167B

5th March 2025

Dear Dr. Satou,

Thank you for submitting your revised manuscript "Discovery of a viral silencer element regulating HTLV-1 latency via RUNX complex recruitment" (NMICROBIOL-24041167B). It has now been seen by the original referees and their comments are below. The reviewers find that the paper has improved in revision, and therefore we'll be happy in principle to publish it in Nature Microbiology, pending minor revisions to satisfy the referees' final requests and to comply with our editorial and formatting guidelines.

Thank you again for your interest in Nature Microbiology Please do not hesitate to contact me if you have any questions.

Sincerely,
Paula Jauregui, PhD
Senior Editor
Nature Microbiology

On behalf of

Reviewer #2 (Remarks to the Author):

The revisions in reply to the reviewer comments generally address the concerns well and improve the quality and impact of the paper. There are three small things to further consider:

1) The change to the title of the paper to address concerns from reviewer #1 and #3 in focusing the findings to HTLV-1 is welcome. In addition, the last sentence of the opening bolded paragraph should change to "...offering new insights into retroviral evolution..." to complete the focusing of this work to HTLV-1. However, the new version of the title loses the impact of the identification of the intragenic positional location. A further refinement to the title might reinstate 'intragenic' into the new title that specifies the HTLV-1 retrovirus.

2) The explanation of the differences between the 5' and 3'LTRs that share identical sequence would be further improved at line 128 by adding one additional word (antisense)...."the 3'-LTR lacks an antisense TATA box...."

3) The correction to Extended Figure 5b is welcomed, but I believe the label on the extreme left column of each panel be "3'LTR", and not "5'LTR".

Otherwise, all the revisions made this excellent paper have improved the clarity and quality of this important study.

Reviewer #6 (Remarks to the Author):

The authors have spoken to the reviewer concerns and addressed them, significantly . The new additions to Figure 4i are low resolution, and should be replaced prior to publication.

With the additional clarifications and analyses, this work provides an important contribution towards understanding HTLV-1 regulation and latency, and addresses gaps in knowledge regarding host-viral interactions, and transcriptional regulation of the HTLV-1 provirus, and retroviral regulation more broadly.

Version 3:

Decision Letter:

7th April 2025

Dear Yorifumi,

I am delighted to accept your Article "Intragenic viral silencer element regulates HTLV-1 latency via RUNX complex recruitment" for publication in Nature Microbiology. Thank you for having chosen to submit your work to us and many congratulations.

You may wish to make your media relations office aware of your accepted publication, in case they consider it appropriate to organize some internal or external publicity. Once your paper has been scheduled you will receive an email confirming the publication details. This is normally 3-4 working days in advance of publication. If you need additional notice of the date and time

of publication, please let the production team know when you receive the proof of your article to ensure there is sufficient time to coordinate. Further information on our embargo policies can be found here:

<https://www.nature.com/authors/policies/embargo.html>

Authors may need to take specific actions to achieve [compliance](https://www.springernature.com/gp/open-research/funding/policy-compliance-faqs) with funder and institutional open access mandates. If your research is supported by a funder that requires immediate open access (e.g. according to [Plan S principles](https://www.springernature.com/gp/open-research/plan-s-compliance)) then you should select the gold OA route, and we will direct you to the compliant route where possible. For authors selecting the subscription publication route, the journal's standard licensing terms will need to be accepted, including [self-archiving policies](https://www.nature.com/nature-portfolio/editorial-policies/self-archiving-and-license-to-publish). Those licensing terms will supersede any other terms that the author or any third party may assert apply to any version of the manuscript.

Congrats again to you and your co-authors! I am looking forward to seeing your paper published.

With kind regards,

P.S. Click on the following link if you would like to recommend Nature Microbiology to your librarian
<http://www.nature.com/subscriptions/recommend.html#forms>

** Visit the Springer Nature Editorial and Publishing website at http://editorial-jobs.springernature.com?utm_source=ejP_NMicro_email&utm_medium=ejP_NMicro_email&utm_campaign=ejP_NMicro for more information about our career opportunities. If you have any questions please click [here](mailto:editorial.publishing.jobs@springernature.com).

Reviewer #1 (Remarks to the Author):

The authors describe the discovery of a specific regulatory region in the center of the HTLV-1 genome that can suppress viral transcription from the 5-LTR promoter through recruitment of the transcription factor RUNX1. The work addresses an important and interesting research question, and is technically elegant with a range of complementary assays and experiments. I have the following suggestions:

Our reply: We appreciate that the reviewer found our work addresses an important and interesting research question. We revised the manuscript according to the comments from reviewers.

1. The authors argue that the 5-LTR promoter activity does not differ between HIV and HTLV-infected cells (Figure 1b), however, the OCR is only present in HTLV. This raises concerns about the functional significance of the OCR for silencing the 5-LTR promoter of HTLV.

Our reply: We have shown promoter assay about effect of the OCR on HTLV-1 and HIV-1 5'LTR (Fig.1b, Fig. 5b). In addition, we further demonstrated that addition of the OCR suppressed HIV production and persistence whereas HTLV-1 with mutated OCR remarkably increased HTLV-1 production (Fig. 3e and f, Fig. 5e and 5g).

Based on these findings, we are proposing the OCR has silencer function not only HTLV-1 LTR but also HIV-1 LTR.

2. Figure 1e suggests that the OCR can suppress the HTLV 5-LTR promoter, however, this is only shown in reporter cell lines. The experiments described in Figure 1e with reporter cell lines are not very well described; specifically, the design of the reporter cell assay should be described more specifically, and additional data showing the validation of this assay should be shown. Experiments showing a direct effect of the OCR on 5-LTR promoter activity in primary cells should be considered.

Our reply: We apologize for insufficient description about the method. We added more information in the revised manuscript. We designed reporter assay to mimic the situation with HTLV-1 provirus. Since the OCR located down stream of 5'-LTR promoter, we made a reporter plasmid DNA containing 5'LTR, reporter gene, and OCR with the same order and orientation as HTLV-1 provirus. Regarding the assay in primary cells, Extended Data Fig. 4b shows a direct effect of the OCR on the 5-LTR promoter activity in primary CD4+ T cells. Primary monocyte is difficult to transfect reporter plasmid; therefore, we alternatively

used monocytes derived from iPS cells. We did not see suppressive the effect in the iPS-derived macrophage. We show the result in Extended Data Fig. 4b.

3. Figure 3c shows that binding of ETS1 to the OCR can stimulate HTLV transcription; therefore, it appears that the OCR/RSR can activate HTLV transcription in specific settings, specifically when ETS1 is highly expressed. Can the authors provide expression intensities of ETS1 and RUNX1 in HTLV-infected PBMC from study participants? How do ETS and RUNX1 expression change during the course of infection?

Our reply: We appreciate this valuable comment. As shown below, RUNX1 expression was slightly decreased in a5 and a9, but increased in a6. EST1 expression was slightly elevated in all cases. These changes may be due to HTLV-1 infection or differences in CD4 subsets between uninfected and infected CD4+ T cells. It would be an interesting question to explore how HTLV-1 infection affects the function of silencer-related molecules, and we are eager to investigate this in our next experiment.

4. In Figure 3, the authors show that a RUNX1 inhibitor can increase HTLV gene expression and enhance the immunological visibility of infected cells. Can HTLV-specific T cells better kill virally-infected cells after treatment with the RUNX1 inhibitor?

Our reply: We appreciate this valuable comment, as it helps us assess the potential translational impact of our findings for therapeutic applications. In response, we conducted an additional experiment to evaluate whether a RUNX1 inhibitor treatment in HTLV-1-infected T cells enhances the cytotoxicity of antiviral CTLs. Our results show a clear enhancement of cytotoxicity in the presence of the RUNX1 inhibitor, using two tax-specific CTLs. These data suggest that the RUNX1 inhibitor could serve as a potential

immunotherapeutic approach (Shock & Kill) against HTLV-1 infection. We have included these new data in Extended Data Fig. 6g.

5. In Figure 4i, the authors show that ETS1 expression is reduced in burst HTLV-infected cells; however, ETS1 binding to the OCR enhances viral gene expression per the authors' prior data. How do the authors explain this?

Our reply: There are likely several interpretations for this result. We assume that ETS1 expression is not the cause but rather a consequence of the proviral transcription burst. This has led us to hypothesize that dynamic changes in OCR-related gene expression play a role in the re-establishment of proviral latency (Extended Data Fig. 10).

6. Can the authors be more specific why the OCR seems to mostly affect the 5-LTR promoter and not the 3-LTR promoter in HTLV-1?

Our reply: In the initial version, we only evaluated the effect of the OCR on the 3'-LTR promoter in one orientation, which does not match the orientation of the HTLV-1 provirus. Therefore, we cannot rule out the possibility that the effect of the OCR on the 3'-LTR promoter was underestimated. To address this, we performed a luciferase assay using the 3'-LTR with an inverted OCR sequence. The results still showed that the effect on the 3'-LTR was much weaker than on the 5'-LTR. The 5'-LTR is known as a TATA promoter, whereas the 3'-LTR is TATA-less². Additionally, different factors contribute to transcription from the 5'-LTR and 3'-LTR. These distinct promoter characteristics likely explain the differing susceptibilities to the OCR. We have added this new result in revised Fig. 1g and included a description in the text.

7. In Figure 5, the authors suggest that the OCR integrated at the nef location in HIV can suppress the 5-LTR activity of HIV. It is not clear how this is mechanistically possible, since nef and the 5-LTR are separated by a considerable genetic distance (around 7KB).

Our reply: It has been reported that chromatin looping forms between the 5'- and 3'-LTR during active transcription of the HIV-1 provirus. This could explain why the OCR can suppress the 5'-LTR even from a region inserted near the 3'-LTR. We have added this description and reference in the revised manuscript. (line 317-320)

8. From a higher level perspective, the data suggest that the HTLV genome contains genetic elements that can regulate proviral transcription and latency in a more autonomous fashion, while HIV lacks such features and largely depends on host genomic features to modulate viral gene expression/repression (Einkauf et al, Cell 2022). Consistent with such

a view, selection of HIV proviruses in specific chromosomal regions seems to be more obvious compared to HTLV, although some recent data suggest selective persistence of HTLV in specific chromatin regions as well (Melamed et al, Science Advances 2023). The authors may wish to discuss this in more detail, and cite the relevant literature.

Our reply: This is a very valuable point. We completely agree with the reviewer that HTLV-1 latency is primarily established through proviral intrinsic mechanisms, as demonstrated in our current study. However, the integration site environment also contributes to HTLV-1 latency (Melamed et al., Science Advances, 2023). In contrast, HIV-1 lacks such regulatory elements within its provirus. This suggests two points: first, HIV-1 is less likely to establish latent infection compared to HTLV-1; second, HIV-1 latency depends more heavily on the host's genetic and epigenetic context at the proviral integration site than HTLV-1 latency (Einkauf et al., Cell, 2022). We have added this point in the revised discussion. (line 369-374)

Reviewer #2: see comments of reviewer #5

Reviewer #3 (Remarks to the Author):

The authors identify a novel open chromatin region in the HTLV-1 provirus. Using in vitro experiments, they find that the region functions as a transcriptional silencer for the HTLV-1 5'-LTR promoter. They show that the transcription factor RUNX1 (and some co-factors) binds to this OCR (named RSR = retrovirus silencer region) and plays a role in the silencer function. Mutant viruses that have impaired RUNX1 binding capacity show higher proviral expression, virus production and immunogenicity in vitro. They insert the silencer within a cloned HIV-1 provirus and this decreases proviral expression, virus production and cytopathic effect.

This is a very elegant study that describes the identification of a novel regulatory region within HTLV-1. It is an important discovery particularly in the field of HTLV-1 which causes devastating diseases, ATL and HAM/TSP. The in vitro experiments have been performed with great attention to every single detail, the single-cell RNA-seq/ATAC-seq work is remarkable, however the authors further claim their findings pertain to "retroviruses" in general and that the absence of this silencer alone can explain the productive phenotype of HIV-1. Also, there is little, if none, evidence from in vivo observations.

General comments:

Overall generalization of the authors' findings to "retrovirus" (also in the title) while this was described for HTLV-1, not for any other retrovirus to our knowledge.

OCR is renamed RSR by the authors = retrovirus silencer region which points to a broad mechanism used by (human) retroviruses in general. Our understanding, according to the authors' findings, is that this RSR is specific of HTLV-1. The authors mention in the discussion and elsewhere that the silencer-related mechanism they uncovered is not used by endogenous retroviruses and other exogenous retroviruses (murine leukemia virus?).

Our reply: We agree with the point reviewer suggested. We show the data regarding OCR with HTLV-1 and HIV-1 but does not with other retroviruses. Thus, we may be better to propose "human retroviral silencer (HRS) region" in revised version. We revised the manuscript accordingly.

HIV: There are many similarities between HTLV-1 and HIV-1, however, there are many differences as well, and simplifying the mechanisms by which HTLV-1 causes cancer (latency) and HIV-1 immunodeficiency (expression, viremia, cell killing) to the presence or absence of a silencer is not a reasonable claim. HTLV-1 is a delta retrovirus. HIV-1 is a lentivirus. Although we agree these viruses share a number of structural features, there are also many differences between these viruses which contribute to their distinct biological behavior. Tax and Tat are very different in structure, and HIV-1 has additional ORFs (vif, vpr, nef ...), the LTRs are significantly different, and there is little consensus regarding antisense expression in HIV-1, many features the authors do not comment on. In the paper, both viruses are "simplified" to Tax and Tat.

Our reply: In this study, we focus on the transcriptional regulation of the provirus. We do not address other virological aspects, such as viral counteractors against host restriction factors or immune evasion, but instead our investigation centers on proviral transcriptional regulation. In this context, the 5' LTR and the Tat/Tax proteins are known to play a central role, which is why we conducted the experiments shown in Fig. 1a, b. Ultimately, we demonstrated that the OCR plays a pivotal role in latency in both HTLV-1 and HIV-1 by performing knock-in or knock-out experiments with the OCR using recombinant viruses (as shown in Fig. 3 and Fig. 5). We observed significant differences in viral productivity for both HIV-1 and HTLV-1 depending on the presence or absence of the OCR. We believe there is sufficient evidence to support the title, "Intragenic silencer regulates human retrovirus latency by recruiting RUNX1." In response to the reviewer's suggestions, we carefully revised the manuscript and added the following sentence in the results section: (line 104-107)

“There are distinct regulatory and accessory viral proteins encoded by the HTLV-1 and HIV-1 proviruses, which explains the significant virological differences between these viruses. Among these proteins, the trans-activators—Tat in HIV-1 and Tax in HTLV-1—are known to regulate proviral gene expression in the sense direction.” (line 350-354)

We also miss a discussion regarding the antisense transcripts (RNA and/or protein for HTLV-1, HBZ) and how they fit within the story of the novel described silencer region.

Our reply: This is really valuable suggestion, so we have added more discussion on this point. There is little suppressive effect of the OCR on the 3’LTR (Fig. 1f, g). In addition, the insulator region between the OCR and 3’TR would also minimize the effect of OCR on the 3’LTR. We added this points in the revised discussion. (line 369-374)

The HIV-1 latent reservoir is the major problem in the HIV-1 clinics nowadays. Upon interruption of anti retroviral (tri-)therapy, the virus rebounds. Thus, HIV-1 is a champion in hiding from the immune system in these reservoirs. The authors mention productive infection, viral expression, but they do not address this essential point.

Our reply: This is really valuable suggestion, so we have added the discussion regarding this point. Also, please see comments for ref #1.

The authors could verify their findings for other delta-retroviruses like STLV-1, the type-C HTLV-1 viruses specifically found in Australia, and HTLV-2? Is this silencer found in BLV, another delta-retrovirus?

Our reply: We performed ATAC-seq analysis using cells infected with other delta retroviruses, including HTLV-2, STLV-1 and BLV. As far as cells we analyzed, there was no open chromatin region in their provirus region except for the LTR. HTLV-1c-infected cells was not available, however there is no sequence difference in RUNX-binding site in the OCR between HTLV-1a and HTLV-1c. We assume that there should be the OCR in HTLV-1c infected cells. We show this new data as Extended Data Fig 1f and 3g.

Conservation of the OCR region between strains? This is shown in extended data Fig 2 f. Were these samples obtained from individuals living in Japan or did the authors look in other strains, originating from US/South America/Caribbean, Australian?

Our reply: This is valuable suggestion. We showed the result from Japanese infected individuals, where HTLV-1a is endemic. To investigate sequence conservation of the OCR region, especially RUNX-binding sites, among various HTLV-1 strains, we analyzed several

subtypes from various endemic areas. The result demonstrated the three RUNX-binding sites in the OCR are well conserved. We show that as a new figure (Extended Data Fig 3g).

The authors claim that defective HTLV-1 proviruses are an exception. There is evidence from the literature about the frequent occurrence of 5'LTR or 5'LTR-extended deletion HTLV-1 proviruses (reported in several publications, significant proportions). Is this region systematically retained in the absence of a 5'LTR or is it part of the deletion(s)? The same question may be asked for tax-mutated HTLV-1 proviruses. Is this silencer still required when Tax is dysfunctional?

Our reply: We appreciate this valuable suggestion and agree with the reviewer's point. We analyzed our previous data of ATL samples. There are ten ATL cases with 5' deletion among 45 cases. Seven ATL cases show monoclonal expansion of infected cells. Two of them contain deletion in the OCR. Even though two ATL cells lost the OCR, they should lose sense transcription due to the 5'-LTR deletion. We show that as a new figure (Extended Data Fig 3h, i).

The in vitro work is outstanding (work in cell lines, transfection experiments). When primary cells are used (ATLs AI5, AI9, smoldering ATLs, ACs), there is little or no information about these samples, both in terms of PVL ("high PVL" in all samples is indicated in the additional information file, Is this the case in ACs, smoldering ATL cases used for single cell work and ATLs earlier in the paper), the clonal situation of the population and proviral structure, etc. This should be clarified for improved understanding.

Our reply: We apologize missing fundamental information in initial version. We presented key information, such as PVL, clonality and pattern of flowcytometry analysis to show the clinical status of each patient (Extended Data Fig. 7).

Fig 1 a: RNA copies in plasma, this is OK, but what about the number of infected T-cells: copies proviral DNA/T-cell, is this correlated with RNA?

Our reply: PVL is available for HTLV-1 infected individuals, whereas HIV-1 PVL is extremely low. For the information for the readers, we presented raw data about plasma viral RNA and HTLV-1 proviral load as table (Extended Data table 3). We also showed association between plasma RNA level and PVL in HTLV-1 infection. There was no positive correlation between them (Extended Data Fig 1a).

For HIV-1 individuals (58), how were they selected?

Our reply: We selected available clinical samples before cART in our stock without any bias.

For HTLV-1 individuals (17), the methods say “from ACs”, there is a need to have an idea of the number of infected T-cells in these individuals which may or may not influence viral RNA levels

Our reply: To provide information for readers, we presented correlation data between plasma viral RNA and HTLV-1 proviral load as Extended Data Fig 1a.

Fig 1 c: w39 and w51 are Jurkat (JET) cells infected and cloned. Is there a single provirus or multiple? Why were these clones specifically selected? Any information regarding the genomic location of the provirus or is this feature not relevant?

Our reply: We established several infected clones with Jurkat T cells in previous study¹. In this study we selected the clones with one copy of the intact provirus. We also characterized viral integration site and the site is the intronic region of the host gene, *CWC27* and *CAMK2D*, respectively. We show these data as a new figure (Extended Data Fig 1b-e).

Fig 1c: cloned HTLV-1 infected JET cells + two ATLs (ex vivo), for HIV-1 the authors use cell lines with “latent” HIV-1. How is this latency explained, in vitro? The authors may want to add primary cells of HIV-1 infected people for improvement.

Our reply: Latency mechanism of ACH-2 is partially explained by a point mutation in the HIV-1 Tat responsive element (Emiliani S et al PNAS 1996, PMID: 8692823). It is also reported the latency of ACH2 and J1.1 is not stable but reversible, suggesting epigenetic mechanisms also involved in the silencing (C Van Lint et al EMBO J 1996, PMID: 8605881). Regarding primary cells of HIV-1 infected people, we found public data from a previous study (Wei Y et al Immunity 2023, PMID: 37922905). We analyzed the scATAC-seq data and found there was no obvious OCR in the HIV provirus. We show this new data as Fig 1e.

Prediction

Prediction of transcription factor binding to the RSR in methods, requires more detailed information (i.e. in supplemental methods section). This refers to SNPs in HTLV-1 infected individuals described by Katsuya (2019). Can the authors include HTLV-1 proviral sequences from different origins, not just a single geographic region and a single paper.

Our reply: We apologize for missing information about the details of the Prediction of transcription factor binding. We added more detailed information in the revised manuscript. To investigate sequence conservation of the OCR region, especially RUNX-binding sites, among various HTLV-1 strains, we analyzed several subtypes from various endemic areas (Extended Data Fig. 3g).

Validation of transcription factor binding prediction with ATAC-seq motif enrichment analysis: elusive description of the method, please clarify or improve the sentences (Peak calling was performed from bam file of ATAC-seq using MACS2 where first bam file converted to BEDPE and then callpeak (options: -f BEDPE --nomodel --shift -37 --extsize 73 -g hs -B --keep-dup all)).

Our reply: We apologize for missing information about the details of the analysis. We added more detailed information in the revised manuscript.

Sometimes RUNX is used, at other places RUNX1, be consistent in denominations or RUNX family if all

Our reply: We appreciate this comment and have addressed this point in the revised manuscript. We used -RUNX to refer binding site for RUNX family and -RUNX1 to refer knock-down, overexpression, western blot, inhibitor etc.

About RUNX1: can the authors more extensively comment on the host-related context in which RUNX1 DNA binding is involved. Where are such motifs located in the host?

Our reply: RUNX proteins are known as DNA-binding proteins located in promoter, enhancer, insulator, and silencer regions in the human genome. Its tissue-specific functions are thought to be governed by the specific expression patterns of RUNX proteins and its cofactors. (Kristie L Durst et al, Oncogene 2004, PMID: 15156176). We have added a description of the role of the RUNX family in host cells. (line 159-162)

Figure 2:

Since these TFs localized to the 5'-LTR as well (Fig. 2a, b), we also analyzed the effect of these TFs on the 5'-LTR itself. These TFs affected promoter activity of the 5'-LTR; however, ...

Luc assays: test the effect on the 3'-LTR promoter activity, IGV shows peaks on both 5' and 3' LTR, given the sequence is similar, and we agree this doesn't mean the 3'LTR is

involved. But can the authors exclude a 3'LTR origin?

Our reply: Since LTR sequence is identical between 5'- and 3'-LTR, we cannot distinguish them. Thus, we are going to mention this limitation in the revised manuscript (line 174-176) .

Order of sub-figures a, b, ... f, g, ... is confusing

Our reply: We got the point, but we cannot rearrange them due to limitation of the space.

“defective” 5'-LTR suppression: do the authors mean “impaired”? g) viral gene expression is “suppressed”, it is decreased, not suppressed

Our reply: We are going to use proper description in revised paper.

Figure 3:

“(b) Experimental workflow illustrates the establishment of stable reporter cells (JET cells) infected with wt- and s-mut-HTLV-1. HTLV-1-infected JET cells were sorted using tdTomato driven by Tax responsive elements as infection marker:” Any idea of the pattern of these cell lines in terms of provirus, is it integrated, are there episomic forms, many integrations or just a single?

Our reply: We performed quantification of cell-associated viral DNA or 2-LTR DNA, a form of episomic viral DNA, by ddPCR. There were a very little proportion of the episomic form in total cell-associated viral DNA, suggesting that proviral load was much higher in cells infected with s-mut-HTLV-1 than those with wt-HTLV-1. (Extended Data Fig. 5a)

Also important for figure 3g (RUNX inhibitor)

Our reply: wt39 and wt51 are Jurkat T cell clones infected with wt-HTLV-1. We established them in a previous study. We show the details in Extended data fig 1 b-e.

(f) s-mut-HTLV-1 JET cells show expansion of PVL during long-term culture after tdTomato sorting: did the authors check this for both cell lines (wt39 and wt51) and was this done in replicate?

Our reply: Due to the limited information provided, some misunderstandings arose. wt39 and wt51 are single-infected clones obtained through limiting dilution in a previous study. We confirmed that there was no increase in PVL after long-term culture (data not shown). Fig. 3f shows data from bulk JET cells infected with either wt-HTLV-1 or s-mut-HTLV-1. We

conducted two independent experiments and obtained similar results. We have modified the figure legend accordingly.

Single-cell data/figure 4 + extended data figures.

Transcriptional burst (general): “This reactivation can cause infectious spread, allowing the virus to establish de novo infection from infected to uninfected individuals”. Has this been effectively demonstrated in vivo?

Our reply: This is theoretical idea. We modified this description according to the reviewer’s suggestion. (line 270-272)

Is there a consensus in the field agreeing on the fact that besides clonal expansion there is also de novo infection at later stages?

Our reply: There is a previous report to describe and discuss thoroughly about this point (PLoS Comput Biol. 2020)².

“To this end, we cultivated primary CD4+ T cells from three indolent type of ATL cases ex vivo overnight to induce transcriptional burst and performed single cell (sc) multiome ...”: Why have smoldering ATLs been chosen for this experiment (a5, a6, a9)? What is the rationale for using ATLs and not infected non-transformed cell populations, and why specifically these three patients, how were they selected?

Our reply: In our experience, indolent ATL cells tend to retain reactivation capacity of Tax protein when we cultivate them *ex vivo*. In contrast, aggressive ATL cells tends to lose the reactivation capacity due to defective provirus or DNA methylation as reported previously. To address our aim, we thought smoldering ATL contain high proviral load with reactivation capacity; therefore, they are suitable for the analysis. To provide more information for the readers, we present the details of these clinical samples in the revised manuscript (Ext Data Fig 7).

Information regarding PVL is not available unless we missed it. Is this consistent with the single cell based detection of infected cells? Do all cells belong to the same malignant clone, defined as carrying a single integration site as expected in ATL samples? If this is the case, will this bias the findings and conclusions?

Our reply: Additional data in Ext Data Fig 7 would address this question.

Could the “burst” cluster and the “latent” cluster be determined by the genomic environment?

Our reply: This is very interesting point and thus is one of the questions we would like to address in future study.

“Infected cells were identified by their transcriptional characteristics (Fig. 4c, Extended Data Fig. 4c and 5c; see Method for more in details), and we further refined cell clustering using only the infected cells.”

“Method: a total of 15,567 cells before ex vivo cultivation (8,486 a5, 3,682 a6, 3,399 a9) and 22577 after ex vivo cultivation (8,375 a5, 10,160 a6, 4,042 a9) were retained for downstream analysis. Then: A total of 6,344 cells before ex vivo cultivation (3,764 a5, 620 a6, 1,960 a9) and 9275 cells after ex vivo cultivation (4,172 a5, 2,464 a6, 2,639 a9) were defined as infected cells. “

Is this consistent, what is the relative contribution to the detection of virus-infected cells by each of the markers used: sense expression, antisense expression, and ATAC-seq peak calling (virus) before cultivation? Do they all three match?

Our reply: In general, scData is sparse, so there are very few cells positive for all three markers. We have provided more details about the method. To estimate the proportion of infected cells in CD4+ T cells, we used the PVL and FACS data. We found proportions of 43.6%, 14.8%, and 37.2% in the a5, a6, and a9 cases, respectively. The proportion of infected cells based on scData was 44.4%, 12.0%, and 44.2% in the a5, a6, and a9 cases, respectively. The values matched closely between the two methods.

And how does each of them relate to the same markers after cultivation?

Our reply: In general, scData is sparse, but after cultivation burst cells are easy to detect, because they express Tax abundantly. But at the same time, some marker gene expressions were changed by Tax induction. However, we did not find any difficulty to identify infected cells because abundant proviral expression helped us to identify infected cells in burst cluster.

In latent cells, the antisense is barely found (Fig 4 h, and extended) while in burst cells it is expressed. Is antisense “silencing” expected in latent cells and how does this relate to 3’LTR activity?

Our reply: Gene expression level is generally regulated by the promoter activity and its openness. As shown Fig. 4g, chromatin structure is much more open in burst cells than latent cells. That would partially explain why antisense transcription level was also enhanced in the burst cell population. In addition, separation between sense and antisense transcript is not so strict in sc-multiome analysis

(<https://www.10xgenomics.com/support/single-cell-gene-expression/documentation/steps/sequencing/interpreting-intronic-and-antisense-reads-in-10-x-genomics-single-cell-gene-expression-data>). That also cause overestimation of antisense transcript in this assay. We describe this limitation in the revised Method.

To what extent is the host transcriptome of a cell modified upon cell culture, in both uninfected and infected cells?

Our reply: In latent cell population, there is similar distribution in cell clustering analysis between before and after cultivated cells, suggesting that there was little cellular transcriptome difference. On the other hand, burst population is quite far from the other cluster. This would be explained by significant change of host transcriptome due to Tax burst. We performed DEG analysis and show the result here. This may not the focus of this study, we would like to propose these results for the reviewer only.

Is it related to burst or latency, and do sense and antisense transcripts (and their corresponding proteins + which proteins amongst all virus-produced) have a direct effect?
 Our reply: It has been reported that Tax is a trans activator of 5'-LTR whereas HBZ suppresses activity of the 5'-LTR. We added this point in the revised manuscript. (line 106-107, 360-362)

Could the authors better explain and provide a visual representation of how they use ATAC-seq marks in single cell ATAC-seq to identify HTLV-1 infected cells?
 Our reply: If two or more reads were detected from HTLV-1 proviral DNA, the cells were defined as infected. We added this description in the revised Method.

What does it look like if PCA and UMPAs are generated separately for ATAC-seq and scRNAseq?

Our reply: We generated the UMAP cell clustering figure by considering either scATAC-seq or scRNA-seq data, and show below as information for the reviewer.

CTL response:

“We performed ELISpot assay and found that infected cells with the s-mut virus showed higher immunogenicity than those with the wild-type virus (Fig. 3i and Extended Data Fig. 4g, h). The result establishes the immunological significance of the RSR by controlling viral antigen Tax.”

The in vitro experiment is elegant and does effectively suggest RSR control of Tax expression, which was shown by the experiments described earlier in the manuscript. That the increase in Tax expression has an effect on the viral immune response is known since many years. Tax is a dominant CTL-inducing viral product. A positive ELISPOT assay is thus expected in any system that enhances Tax expression. It does not confirm that RSR contributes to mechanisms underlying the modulation of immune response in vivo.

Our reply: Please let us repeat reply for the reviewer #1.

We appreciate this valuable comment because the investigation would highlight whether the findings would be translatable for therapeutic approaches. We performed additional experiment to evaluate whether RUNX inhibitor enhances the cytotoxicity of antiviral CTL or not. There was clear enhancing effect in the presence of RUNX inhibitor on cytotoxicity using two tax-specific CTLs. These data indicated that RUNX inhibitor has a possible approach for immune therapy against HTLV-1 infection. We show this new data as Extended data fig 6g.

Other comments:

(Introduction) "HTLV-1 uniquely undergoes a self-induced latency, leading to a spontaneous reduction of viremia, obviating the need of anti-retroviral drugs."

Antiretroviral therapy (ZDV IFN-alpha) is used to treat ATL patients and this treatment is increasingly considered worldwide. Antiretroviral drugs are in the pipeline as candidates for preemptive intervention.

Our reply: We will change the sentence as below.

"HTLV-1 uniquely undergoes a self-induced latency, leading to a spontaneous reduction of viremia, obviating the need of anti-retroviral drugs except for therapeutic use for ATL."

Methods, recombinant HIV: To measure integrated rHIV-1 after 24 hours, genomic DNA in the Jurkat cells was extracted ... Is it possible to discriminate between integrated and episomal forms (double stranded unintegrated provirus, and do the authors think integration is fully completed after 24 h.

Our reply: We thank the reviewer for this valuable comment. We have updated the terminology from 'integrated rHIV-1' to 'cell-associated HIV DNA' to provide a more accurate description.

Language: could be improved, some sentences are hard to understand and several errors can be easily fixed (examples below):

Line 236 a maker of HTLV-1

Line 293 aligned to HITL-1

Line 461 as shown the above section

Our reply: We improved these points.

Reviewer #4 (Remarks to the Author):

In this manuscript, the authors identified a key regulatory element that represses sense transcription of the HTLV-1 provirus, demonstrating that RUNX1 is located at the RSR and plays a pivotal role in silencer function. They also found that insertion of the RSR into recombinant HIV-1 significantly reduced viral expression, virus production, and cytopathic effects. The authors' group recently showed that HTLV-1 antisense transcription is constitutively activated through the role of viral enhancers. This study further extends that previous work, demonstrating that HTLV-1 sense transcription is repressed through RSR and RUNX1. Overall, the new findings about RSR and RUNX1 have potential implications for future research and antiretroviral drug development. However, here are some suggestions to review and address for publication.

Major issues:

1) The author demonstrated RSR repress the transcriptional activity of 5'LTR through the RUNX1 and co-factors. However, the molecular mechanism How the distal RSR affect the transcriptional activity of the 5'LTR is lack. I recommend using 3C-based experiments to determine the topological association between the 5'LTR and RSR in JET cells infected with HTLV-1-wt and -s-mut-RSR using 3C-based experiments.

Our reply: We performed 3C assay according to this comment. The close proximity between the 5' LTR and OCR in wt-HTLV-1-infected cells was greater than that in s-mut-HTLV-1-infected cells. We show this result as new figure (Extended Data Fig. 5c, d).

In 2016, Satou et al. reported the epigenetic properties of the HTLV-1 provirus. Is the distal regulation of RSR in 5'LTR activation associated with epigenetic properties? We recommend identifying epigenetic features such as histone modifications and DNA methylation in JET cells infected with HTLV-1-wt and -s-mut-RSR to elucidate the mechanism of distal regulation of RSR.

Our reply: We performed CHIP assay according to the reviewer's comment. There was decrease of active histone modifications, H3K4me3 and H3K9ac, in the 5'side of provirus in wt-HTLV-1-infected cells as reported previously. In contrast, the decrease was canceled in s-mut-HTLV-1-infected cells (Extended Data Fig. 5b).

2) RSR(4349-4531) appears to be binding site for multiple transcription factors, including repressor RUNX1 and activator ETS1. The authors focus primarily on the role of RUNX1, as RSR appears to play a repressive role in the activation of 5'-LTR, whereas ETS1 activates the 5'LTR in a dose-dependent manner. Experiments exploring the interplay

between RUNX1 and ETS1 is recommended, as RUNX1 and ETS1 have opposing effects on the same RSR.

Our reply: We will perform luc assay to analyze interplay between RUNX1 and ETS1 by overexpressing RUNX1 and ETS1 in 293T cells. ETS1 enhanced reporter activity of 5'-LTR and the OCR expression unit in the presence of RUNX1 in a dose dependent manner. We show the result as a new figure (Fig 2d).

Minor issues:

1) In figure 1e: Why does OCR only affect transcription of 5'LTRs and not 3'LTRs? The author should mention or discuss this in the text.

Our reply: Please let us repeat reply to Reviewer #1. In the initial version, we only evaluated the effect of the OCR on the 3'-LTR promoter in one orientation, which does not match the orientation of the HTLV-1 provirus. Therefore, we cannot rule out the possibility that the effect of the OCR on the 3'-LTR promoter was underestimated. To address this, we performed a luciferase assay using the 3'-LTR with an inverted OCR sequence. The results still showed that the effect on the 3'-LTR was much weaker than on the 5'-LTR. The 5'-LTR is known as a TATA promoter, whereas the 3'-LTR is TATA-less. Additionally, different factors contribute to transcription from the 5'-LTR and 3'-LTR. These distinct promoter characteristics likely explain the differing susceptibilities to the OCR. We have added this new result in revised Figure 1g and included a description in the text.

2) Line #145: Where are the data for following statement "293T cells that have no endogenous expression of these TFs"?

Our reply: We referred a public database, The Human Protein Atlas:

(<https://www.proteinatlas.org>) and found the following information;

RUNX1: 293T (1.0 TPM), Jurkat (241.1 TPM)

ETS1: 293T (3.6 TPM), Jurkat (235.8 TPM)

GATA3: 293T (9.6 TPM), Jurkat (61.3 TPM)

Their expression level is low but not zero, so we change from "no endogenous expression" to "much lower endogenous expression when compared with those in Jurkat T cells" (line 172-173)

3) In figure 2c and 2e: In figure 2c, the addition of GATA3 did not significantly change the relative luciferase activity, but in figure 2e, the relative luciferase activity was clearly reduced in the mutations in the GATA3 binding sites. How can the author explain this discrepancy?

Our reply: Figure 2e shows that mutation in the GATA3-binding site slightly increased the luciferase signal but still retained silencer activity. These data indicate that RUNX1 plays a major role in binding the silencer complex to the OCR. We also confirmed this by performing a ChIP-qPCR assay using plasmids with mutations in the RUNX1-binding, GATA3-binding, or ETS1-binding sites. The results below demonstrated that mutations in the RUNX1-binding site significantly decreased the binding activity of all three related proteins, while mutations in the GATA3-binding or ETS1-binding sites did not significantly change the binding activity of any of the three related proteins.

4) Line #152: A period is missing from the sentence.

Our reply: After 48 hours of the transfection, luciferase assay was performed. We added the information to the figure legend.

5) In figure 2e and 3a: In figure 2e, detailed sequence information for all mutations is required. In figure 3a, the data of luciferase activity assay is redundant with figure 2e.

Our reply: We have added detailed sequence information regarding figure 2e as Extended

Data Table 4. Figure 3a demonstrates whether the mutations used for silencer mutant virus lose silencer activity or not. We would like to keep Figure 3a.

6) The lack of explanation of the data can be misleading to readers.

- Figure 2h: What is the \square NGFR in figure 2h?

-Figure 4a-f: The explanation of data in the main text and figure legends are so implicated. Also, the extended data Figure 4C does not link to transcriptional characteristics as mentioned in line #229.

- Figure 4j and Line #249: What is the TF motif activity? The description of the schematic model in figure 4k is also missing in the main text and figure legend.

Our reply: We apologize for not providing enough information in the initial manuscript. We have corrected all these points in the revised manuscript.

Reviewer #5 (Remarks to the Author):

Satou and coworkers have compared HIV and HTLV-1, the two pathogenic retroviruses in humans. They focused their attention on understanding why HTLV-1 is not active in replicating itself, causing long years of latency whereas HIV is very active in replication and the onset and development of the consequent disease is quick. They identified a formerly unknown element in the HTLV-1 that may account for this difference. This work is novel solid and likely opens up a new area of research, with special emphasis on the involvement of RUNX-1 in the pathophysiology of HTLV-1.

While the study is very intriguing, there are a few points that would make the statements/conclusion of this paper not convincing/relevant enough.

Major issues;

1. Are there any mutations of HTLV-1 in the RSR region known?

Our reply: To investigate sequence conservation of the OCR region, especially RUNX-binding sites, we analyzed several HTLV-1 subtypes from various endemic areas. The result demonstrated the three RUNX-binding sites in the OCR are well conserved. There were some ATL cells with the deletion. We show these results in new figures (Extended Data Fig 3f-i).

2. If such mutation exists, the mutated HTLV-1 might cause higher viremia, and moreover, a juvenile or fulminant form of T-cell leukemia. There have been some reports of early

onsets of ATL/HAM (Oliveira et al., J Trop Pediatr 2018 64: 151). Any insights on the connection?

Our reply: As far as we checked, there are no mutation in the OCR except for deletion of the provirus. We mentioned this possibility by adding the reference (Oliveira et al., J Trop Pediatr 2018 64: 151).

3. The virological nature of the s-mut HTLV-1 would be interesting. Whether its capacity to create T-cell leukemia is augmented or not can be perhaps tested infecting this virus in vivo to normal human T-cells in humanized mice.

Our reply: In this study, we focus on virological impact of the OCR. We do agree with the reviewer that one of next key question is what is the impact of the OCR on the viral pathogenesis, including oncogenesis and inflammation. This is one of the experiments we are keen to perform as next study.

4. In addition to HTLV-1 mutations in the RSR region (noted above), there are individuals with RUNX1 mutations. Have there been known cases in which such individuals have contracted HTLV-1? Do they manifest fulminant T-cell leukemia?

Our reply: We performed promoter assay with RUNX1 mutants reported in ATL patients. As shown below, RUNX1 mutants found in ATL patients in previous report³ retained suppressive effect on the 5'LTR+OCR expression unit (Extended Data Fig 4f).

Minor comments.

5. In the abstract (line 43) and introduction, repeatedly it is mentioned that the HTLV-1 infection leads to the expansion of infected cells over the uninfected cells and causes leukemia. However, the disease manifestation of HTLV-1 is not limited to malignancy. In fact, the combined percentages of HAM/TSP, uveitis and other immunological in which “inflammatory” nature of the infected cells would account for pathogenic involvements.

Thus, this sentence is too simplified and misleading. It needs to be modified. For example, the potential increased proliferation would be seen in asymptomatic carriers as well – have there been studies indicating if proliferative propensity would indicate an increase in the later leukemic occurrence?

Our reply: We have modified introduction according to the reviewer's suggestion. (line 71-77)

6. The authors chose RUNX1, ETS1 and GATA3 because their expression levels are higher than other family members in normal CD4 T-cells. Why? If HTLV-1 infection would elevate the levels of any TFs, would those not be more plausible candidates?

Our reply: We appreciate for this valuable suggestion. We addressed this point by conducting further analyses. Each RUNX molecule showed a suppressive effect on the HTLV-1 5'LTR, with RUNX1 exhibiting the strongest effect. These data suggests that RUNX1 plays a major role in the silencer function. Based on these results, we are focusing on RUNX1, ETS1, GATA3 in this study. We agree with the reviewer that further experiment would be required to elucidate molecular function underlying silencing function of the HRS region and whether HTLV-1 infection modifies expression level of these transcription factors. We are keen to address these points as next study. (line 146-153)

7. In line 205, it should read “freshly isolated PBMCs from HTLV-1 infected individuals”, otherwise misleading.

Our reply: We totally agree with this point and corrected in revised manuscript. (line 249)

8. How specific is Ro5-3335 inhibitor? Are other cellular factors/pathways known to be inhibited by this compound? If the specificity is not guaranteed, the effects observed here can be because of an inhibition of unknown pathways, not by the RUNX1/CBF inhibition.

Our reply: We cannot exclude the possibility that the drug may affect targets other than RUNX1 inhibition; however, we also performed knockdown experiments, which showed a similar tendency. We added the following sentence to the revised text. “Although we cannot exclude the possibility that off-target effects of the RUNX inhibitor may contribute to the experimental outcome, these results establish the immunological significance of OCR in controlling the viral antigen Tax (line 265-267).

9. Perhaps in humans (in vivo), mutations that would negate the latency (and/or expose highly immunogenic antigens) would be detrimental for the survival of the virus, but the cell culture system has no such restrictions. It would be interesting to examine any HTLV-1

infected cell lines for similar mutations. Perhaps these mutations may have more positive impacts on the host cell biology?

Our reply: This is also very interesting suggestions. We'd like to try these points as near future study.

- 1 Matsuo, M. *et al.* Identification and characterization of a novel enhancer in the HTLV-1 proviral genome. *Nat Commun* **13**, 2405, doi:10.1038/s41467-022-30029-9 (2022).
- 2 Laydon, D. J., Sunkara, V., Boelen, L., Bangham, C. R. M. & Asquith, B. The relative contributions of infectious and mitotic spread to HTLV-1 persistence. *PLoS Comput Biol* **16**, e1007470, doi:10.1371/journal.pcbi.1007470 (2020).
- 3 Kogure, Y. *et al.* Whole-genome landscape of adult T-cell leukemia/lymphoma. *Blood* **139**, 967-982, doi:10.1182/blood.2021013568 (2022).

Reviewer #1 (Remarks to the Author):

In this manuscript the authors identified an open chromatin region (OCR) within HTLV1 genome, responsible for transcriptional silencing of the provirus. OCR transcriptional silencing is mediated by direct recruitment of RUNX1/CBFb/HDAC3. Remarkably, mutation of the RUNX1 binding site within HTLV-1 OCR region results in enhanced viral replication and spreading in vitro. The data is convincing, and support the conclusions. The work is of great importance and will represent a forward step towards our understanding of HTLV-1 latency and associated pathology.

The authors' response to reviewer 1's comment regarding HIV-1 is accurate and satisfying. However, showing that the OCR function as transcriptional silencer of HIV-1 LTR in reporter assay and HIV-1 in replication assay does not make it a general human retrovirus silencer (HRS) region. If the authors want to keep such statement, additional experiments are required. Notably, extending the experiment shown in figure 1f-g and figure 2e to non-retroviral promoters (such as CMV and SV40) and to cellular promoters (TATAA containing and TATAA-less). In my opinion, the manuscript is strong enough to merit publication in Nature microbiology even without such statement.

Our reply: We understand this point. New title is "Discovery of a viral silencer element regulating HTLV-1 latency via RUNX complex recruitment". We will be happy to discuss with editors and reviewers for more modification if necessary.

Reviewer #2 (Remarks to the Author):

- This compelling report from Sugata et al. details an insightful analysis of the control mechanisms that dictate the intriguing dominance of latent or non-expressive nature of the transmissible virus particles from the HTLV-1 from the positive sense DNA of the integrated provirus while maintaining with a low pulse of a negative sense gene products that ensures ongoing proliferation of infected cells until a moment when a burst the Tax protein is expressed that greatly enhances an outpouring of infectious virion. The authors contrast this with the other better known human retrovirus, HIV-1, that dominantly maintains a high-level cytopathic expression of transmissible virion and only rarely establishes a latent state of infection. The work presented characterises a region of HTLV-1 proviral DNA that interacts with cellular transcription factors of the RUNX1 family and associating co-factors in a far-ranging study detailing the molecular control elements of HTLV-1 transcriptional latency. They present a very well designed and executed study and deliver an important

finding that strongly reveals new insights into the control of both the HTLV-1 and HIV-1 infections.

The authors have generally addressed most reviewers' concerns, and the most significant shortfall is not sufficiently addressing the following concern: "In latent cells, the antisense is barely found (Fig 4 h, and extended) while in burst cells it is expressed. Is antisense "silencing" expected in latent cells and how does this relate to 3'LTR activity?" Authors have referred to 10X documentation, which suggests antisense reads can be overestimated. If strandedness can't be accurately assigned by scRNA-seq, it should not be presented on the figures. This is not a limitation as mentioned in the methods, but rather it is misleading. As pointed out, the near equal levels of antisense and sense transcription have implications for viral persistence, and if this is a technical artefact, it shouldn't be presented. Instead, it should be shown as viral expression (without quantifying sense and antisense). This is further supported by the qPCR results in Fig. 3d, which are inconsistent with scRNA-seq data presented, showing that changes to tax and hbz expression aren't equivalent when the OCR is disrupted.

Our reply: We agree with the point. To avoid any misleading to the readers, we modified the figure to show total proviral transcript rather than sense and anti-sense separately (Fig. 4h).

However, there are a number of additional less direct concerns from the original reviewers that I recommend for further attention and clarification by the authors.

- The opening paragraph should be reframed. It is a false equivalence to compare regulatory mechanisms for endogenous (ERV) and exogenous retroviruses in this section, given the distinct evolutionary histories (resulting in the so-called "arms race" between ERVs and KZFPs), developmental regulatory pathway (endogenous retroviruses are subject to silencing that is established upon epigenetic reprogramming in early development, which can be stably maintained, whereas exogenous silencing needs de novo establishment), and genetic diversity. Authors should highlight these differences, which underpin why very different mechanisms are employed to control endogenous and exogenous retroviral expression. It also highlights the importance of the work presented in this manuscript, as authors describe a novel specific pathway for the control of exogenous retroviral expression.

Our reply: We totally agree with this comment. We have added new sentences in the introduction (Line 61-63).

- Given antiretroviral drugs for HTLV-1 are in the pipeline as candidates for pre-emptive intervention for infection, the statement “Obviating the need for anti-retroviral drugs” should be removed from the introduction.

Our reply: We have removed the statement.

- Can the authors clarify what the difference is between the 5' and 3' LTRs in the reporter assays apart from their orientation for the direction of transcription? Authors state that in NGS the 3' and 5' LTRs cannot be distinguished due to their identical sequences, so how was this resolved for a reporter assay? As the data unfolds it becomes unclear how the position and orientation of the OCR impacts on the flexibility for transplanting the ascribed functions into other settings.

Our reply: The sequence of 5'- and 3'LTR is identical; however, their promoter activities are clearly different. 5'LTR is a TATA-promoter whereas 3'LTR is TATA-less one. We added more on this point in revised manuscript (Line 126-128).

- RUNX1 binds to a suite of co-factors to effect transcription, either positively or negatively. Why specifically were SIN3A and HDAC3 selected for further characterisation, over other cofactors such as TLE, HDAC1, etc.. for example? This should be stated in text.

Our reply: We analyzed cofactors of RUNX1-binding proteins and showed proteins that we obtained evidence of the localization to the OCR by ChIP assay. However, other factors may be involved in the silencer complex. Thus, we added description below in the revised text (Line 149-150). “Additional studies are needed to fully elucidate the molecular mechanisms underlying the silencer complex.”

- Can the authors comment on difference observed between Jurkat cell lines and ATL samples for binding of repressive complexes at the OCR in Fig. 2a? Specifically, how this relates to transcription from the provirus in each context? Can they explain the difference observed? This should be discussed in text.

Our reply: Generally speaking, HTLV-1 provirus in cell lines tends to form heterochromatin. One reason for that is DNA hypermethylation. Thus, HTLV-1 latency in cell line depends less on the silencer. We added the description in the revised text (Line146-149).

- RUNX1 behaviour as repressor or activator is regulated by PTMs, which isn't captured by

correlation of repressive factor expression (Fig. 4i). Given the ubiquitous roles of cellular TFs and epigenetic modifiers, it's not clear what the relevance of correlating their expression would be to this model. This should be discussed in text, as it is likely more relevant than changes in expression of the transcriptional complex. Further, a much more informative representation of the data would be to create UMAP projections of all cells with a heatmap of expression levels for RUNX1, ETS1, GATA3, HDAC3, and SIN3A (as in Fig 4c) for all cells. This should be included as extended data.

Our reply: Thank you for this valuable comments. We showed the data in the revised figures and add the discussion about possible role of PTMs in silencer function in the text (Line 141-143). Also, we change the title from RUNX1 to RUNX complex, because RUNX not as itself but as complex exerts silencer function.

- The model presented extended Fig. 10 is lacking a figure legend to explain authors' hypothetical dynamic model. Can the authors please provide one, detailing their model and caveats?

Our reply: We have added the legend for the figure (Line 956-960).

- RUNX1 is unique in that it has both activating and repressive functions, depending on co-factor binding and PTMs, making it an intriguing factor that could act as a molecular switch. We agree with authors, that there are many mechanistic insights that will be gained from future studies, which will be important. However, the following points should be addressed in the model and/or in text in the discussion.

1. Data show ETS1 seems to either compete with or modify RUNX1 repressive behaviour (Fig. 2C). ETS1 usually co-binds with RUNX1 to activate transcription. Changing stoichiometry of ETS1 binding is likely an important molecular switch for RUNX1 function. This is not reflected in the discussion.

Our reply: We appreciate for valuable and thoughtful comments. We described this point in the result (Line 251-252).

2. Binding of RUNX1 and co-factors are inferred from population data from a very heterogenous population of cells, with regard to sense/antisense expression from the provirus. Binding dynamics and stoichiometry have not been resolved in cells with defined sense and antisense expression. This is an important consideration that should be mentioned in text.

Our reply: We do agree with this valuable comment. We described this point in the discussion (Line 307-310).

3. RUNX1 behaviour as repressor or activator is regulated by PTMs, often catalysed by chromatin modifying complexes themselves. This should be mentioned in the model or discussion for future studies, particularly as this is likely to be a more important regulator than expression levels of co-factors.

Our reply: We have added sentence to describe this point and related references (Line 141-143).

- The manuscript would benefit from copy-editing before publication. For example:

- Line 45:why HTLV-1 prefers... is an anthropomorphic term and should state..... how HTLV-1 evolved....., or perhaps “it remains unknown how HTLV-1 natural selection has led to such a latency phenotype in contrast to HIV-1”
- Line 77:.....tropical, not tropic
- Line 83:not “an equivalent”, but an analogue
- Line 126....”not being reported”, but perhaps really....not known as...
- Line 269...”reactive” should be “reactivate”

Our reply: We described this point in the result. We have corrected all these points accordingly.

Reviewer #3 (Remarks to the Author):

I am generally satisfied with the authors' responses to my questions. They have adequately addressed most of my concerns and incorporated relevant changes in the revised manuscript, including the addition of new data in Figures (Fig. 1) and Extended Data Figures (several).

However, I have one remaining concern regarding the generalization in the title and the naming of the silencer region. The region should be specifically referred to as the HTLV-1 silencer region rather than the broader term human retroviral silencer region. Additionally, the title should explicitly mention HTLV-1, such as: 'Intragenic silencer regulates human T-cell leukemia virus-1 latency by recruiting RUNX1.

Our reply: We corrected this. New title is Discovery of a viral silencer element regulating

HTLV-1 latency via RUNX complex recruitment.

Reviewer #4 (Remarks to the Author):

In the revised manuscript, the author addressed many of the reviewers' comments. The novel finding regarding OCR is particularly intriguing, as it is linked to the repression of the 5'-LTR.

Ext Fig 5c,d

In response to the reviewer's recommendation, the author conducted 3C analysis to investigate the association between OCR and the 5'-LTR. The results revealed that the wild-type OCR associates with the 5'-LTR, and this association is disrupted by mutations in the OCR sequence.

I wonder whether the association between OCR and the 3'-LTR is blocked by CTCF. It would be beneficial to examine this association and evaluate the role of CTCF in the process. Such insights would greatly enhance our understanding of the underlying mechanism of OCR-mediated gene regulation.

Our reply: We totally agree with this point. We have added more description about this point. Also, we are keen to perform experiment as we describe in the text as below (Line 257-258).

Further studies are needed to elucidate how intragenic regulatory elements—silencer, insulator, and enhancer—coordinate proviral expression and retroviral latency in the host.

Ext Fig 5b.

Active histone markers are more broadly distributed in s-mut-HTLV-1 compared to wt-HTLV-1. I recommend checking the enrichment of histone markers specifically at the OCR position.

The enrichment of histone markers near the 3'-LTR may be higher in wt-HTLV-1 than in s-mut-HTLV-1. How can this result be explained?

Our reply: Thank you very much for this thoughtful comment. We have modified the figure and add value for ChIP-qPCR at OCR in the revised Ext Fig 5b.

Reviewer #2:

Remarks to the Author:

The revisions in reply to the reviewer comments generally address the concerns well and improve the quality and impact of the paper. There are three small things to further consider:

1) The change to the title of the paper to address concerns from reviewer #1 and #3 in focusing the findings to HTLV-1 is welcome. In addition, the last sentence of the opening bolded paragraph should change to "...offering new insights into retroviral evolution..." to complete the focusing of this work to HTLV-1. However, the new version of the title loses the impact of the identification of the intragenic positional location. A further refinement to the title might reinstate 'intragenic' into the new title that specifies the HTLV-1 retrovirus.

Our reply: We would like to suggest a new title "Intragenic viral silencer element regulates HTLV-1 latency via RUNX complex recruitment".

2) The explanation of the differences between the 5' and 3' LTRs that share identical sequence would be further improved at line 128 by adding one additional word (antisense)... "the 3' -LTR lacks an antisense TATA box..."

Our reply: Thank you for the useful comment. We have modified the sentence accordingly.

3) The correction to Extended Figure 5b is welcomed, but I believe the label on the extreme left column of each panel be "3' LTR", and not "5' LTR".

Our reply: Thank you for the useful comment. We have modified the figure accordingly.

Otherwise, all the revisions made this excellent paper have improved the clarity and quality of this important study.

Reviewer #6:

Remarks to the Author:

The authors have spoken to the reviewer concerns and addressed them, significantly. The new additions to Figure 4i are low resolution, and should be

replaced prior to publication.

Our reply: Thank you for the valuable comment. We have modified the figure accordingly.

With the additional clarifications and analyses, this work provides an important contribution towards understanding HTLV-1 regulation and latency, and addresses gaps in knowledge regarding host-viral interactions, and transcriptional regulation of the HTLV-1 provirus, and retroviral regulation more broadly.

Our reply: We sincerely appreciate the reviewers for dedicating their valuable time to providing a critical and insightful peer review. Their feedback and suggestions have been immensely helpful in enhancing the clarity and quality of our manuscript.